# Revisiting Mixture Policies in Entropy-Regularized Actor-Critic

## Abstract

Mixture policies in reinforcement learning offer greater flexibility compared to their base component policies. We demonstrate that this flexibility, in theory, enhances solution quality and improves robustness to the entropy scale. Despite these advantages, mixtures are rarely used in algorithms like Soft Actor-Critic, and the few empirical studies that are available do not show their effectiveness. One possible explanation is that base policies, like Gaussian policies, admit a reparameterization that enables low-variance gradient updates, whereas mixtures do not. To address this, we introduce a marginalized reparameterization (MRP) estimator for mixture policies that has provably lower variance than the standard likelihood-ratio (LR) estimator. We conduct extensive experiments across a large suite of synthetic bandits and environments from classic control, Gym MuJoCo, DeepMind Control Suite, MetaWorld, and MyoSuite. Our results show, for the first time, that mixture policies trained with our MRP estimator are more stable than the LR variant and are competitive compared to Gaussian policies across many benchmarks. In addition, our approach shows benefits when the critic surface is multimodal and in tasks with unshaped rewards.

## 1 Introduction

Policy gradient methods are widely used in online reinforcement learning (RL), particularly for continuous action spaces, and yet, there are many design decisions in these methods that remain underexplored. One of these is the choice of policy parameterization. Gaussian policies are by far the most common parameterization (Williams, 1992; Lillicrap et al., 2015; Schulman et al., 2017), or its bounded variants like squashed Gaussian policies (Haarnoja et al., 2018a). There are a handful of works exploring other distributions, including beta policies (Chou et al., 2017) and the family of heavy-tailed policies (Kobayashi, 2019; Bedi et al., 2024). Using mixture policies, such as a conditional Gaussian mixture model for the policy, however, has been largely unexplored.

Yet there are reasons that this increased flexibility from mixture policies could be beneficial. A more flexible policy class may contain better optimal policies. For example, when the environment is partially observable, the optimal policy could be stochastic and multimodal (Sutton & Barto, 2018). Even in fully observable settings, it is common to use entropy-regularized objectives, which prefer stochastic policies; multimodal rather than unimodal policies may be better in this regime. The increased flexibility may also facilitate exploration. Several works have shown that heavy-tailed policies may improve learning through better exploration compared to Gaussian policies (Kobayashi, 2019; Bedi et al., 2024). Similarly, mixture policies offer a complementary approach by enabling mode-directed exploration, maintaining high probability for multiple promising actions.

Though underexplored, some work has looked at more flexible policy classes. Implicit policies use deep generative models (e.g., energy-based models (Haarnoja et al., 2017; Messaoud et al., 2024), normalizing flows (Tang & Agrawal, 2018; Mazoure et al., 2020) and diffusion models (Wang et al., 2023)). Compared to these more complex implicit policies, policies using parametric distributions like mixture models have two benefits: they are simpler to train, and the explicit densities are useful in entropy-regularized RL. Otherwise, several unpublished works briefly touch on mixture policies. The first version of Soft Actor-Critic (SAC; Haarnoja et al., 2018b) did use mixture policies, but Haarnoja et al. did not pursue this further nor provide insights on this choice. We hypothesize that the lack of reparameterization (RP) gradient estimators for mixture policies may explain why later

versions of SAC switched to a single Gaussian, as the RP estimator was found to perform better (see Footnote 3, p. 67 of Haarnoja, 2018). Later, Hou et al. (2020) tried to avoid reparameterization of the whole mixture policy by using a separate objective for the weighting policy, but they found little improvement from their approach. Finally, Baram et al. (2021) explore the utility of the upper and lower bounds of the mixture model's entropy, without considering a learnable weighting policy. We provide a more in-depth discussion on related work in Section A.

In this work, we provide the first RP estimator for mixture policies that does not compromise flexibility and is empirically viable. We first prove that mixture policies provide improved solution quality: they have comparable or better objective values and are more robust to larger entropy regularization, in that stationary points may not exist for Gaussian policies but do for Gaussian mixture policies. We then derive the Marginalized RP estimator (MRP) for mixture policies and prove that it has lower variance than the standard likelihood-ratio estimator. RP estimators have been critical for the practical success of SAC, and we provide a similarly effective RP estimator for mixture policies.

We empirically study mixture policies and the proposed MRP estimator with SAC across a broad suite of benchmarks. In synthetic bandit experiments designed to mimic multimodal critic functions, mixture policies more often find the maximal peak than base policies. On larger benchmarks—including 7 Gym MuJoCo, 10 DeepMind Control Suite, 30 MetaWorld, 10 MyoSuite, and 6 classic control environments—mixture policies perform on par with base policies and significantly outperforms other estimators from the literature. MRP does improve performance in environments with unshaped rewards because the critic is less smooth and exhibits more peaks, allowing the policy to exploit multiple modes more effectively. Taken together, we are the first to provide a practical and performant actor-critic algorithm with mixture policies with theoretically-sound reparameterization.

## 2 PROBLEM FORMULATION

We consider the standard Markov decision process (MDP) problem setting. An MDP is defined by $\langle \mathcal{S}, \mathcal{A}, p, d_0, r, \gamma \rangle$, where $\mathcal{S}$ is the state space, $\mathcal{A}$ is the action space, $p$ is the transition function, $d_0$ is the initial state distribution, $r$ is the reward function, and $\gamma$ is the discount factor. In this paper, we consider $\mathcal{A}$ to be continuous, and $r$ to be deterministic and bounded by $[-r_{\max}, r_{\max}]$. The agent's goal is to find a policy $\pi$ that maximizes the *expected return* from the start states:

$$J_0(\pi) \doteq \mathbb{E}_\pi \left[ \sum_{t=0}^\infty \gamma^t r(S_t, A_t) \right], \tag{1}$$

where the expectation is with respect to the initial state distribution, transition function, and policy.

Oftentimes, the agent optimizes the *entropy-regularized objective* that promotes stochastic policies:

$$J(\pi) \doteq \mathbb{E}_\pi \left[ \sum_{t=0}^\infty \gamma^t \left( r(S_t, A_t) + \alpha \mathcal{H}(\pi(\cdot|S_t)) \right) \right] = \mathbb{E}_{s \sim d_0, a \sim \pi(\cdot|s)} \left[ Q_\pi(s, a) - \alpha \log \pi(a|s) \right], \tag{2}$$

where $\alpha$ is the entropy scale, $\mathcal{H}(q) \doteq - \int q(x) \log q(x) \, dx$ is the differential entropy for distribution $q(x)$, and $Q_\pi(s, a) \doteq \mathbb{E}_\pi [\sum_{t=0}^\infty \gamma^t (r(S_t, A_t) + \alpha \gamma \mathcal{H}(\pi(\cdot|S_{t+1})))]$ is the soft action-value function.

Soft Actor-Critic (SAC; Haarnoja et al., 2018a) learns $\pi$ by maximizing a surrogate of Equation (2):

$$\hat{J}(\pi_{\boldsymbol{\theta}}) = \mathbb{E}_{S_t \sim \mathcal{B}, A_t \sim \pi_{\boldsymbol{\theta}}(\cdot|S_t)} \left[ Q_{\mathbf{w}}(S_t, A_t) - \alpha \log \pi_{\boldsymbol{\theta}}(A_t|S_t) \right], \tag{3}$$

where $\mathcal{B}$ is a buffer of collected data and $Q_{\mathbf{w}}$ is an estimate of $Q_{\pi_{\boldsymbol{\theta}}}$. In this work, we focus on the role of policy parameterization; we refer the reader to the original paper for other details on SAC.

We can obtain an unbiased sample of the gradient of Equation (3) in two ways. One is the *likelihood-ratio* (LR) gradient estimator (Williams, 1992):

$$\hat{\nabla}_{\boldsymbol{\theta}} \hat{J}(\pi_{\boldsymbol{\theta}}) = \nabla_{\boldsymbol{\theta}} \log \pi_{\boldsymbol{\theta}}(A_t|S_t) \left( Q_{\mathbf{w}}(S_t, A_t) - \alpha \log \pi_{\boldsymbol{\theta}}(A_t|S_t) \right). \tag{4}$$

The LR estimator often suffers from high variance, and a baseline is often used to reduce variance. When the action is reparameterizable as $A_t = f_{\boldsymbol{\theta}}(\epsilon_t; S_t)$ with $\epsilon_t$ sampled from a prior distribution $p(\cdot)$, an alternative is to use the *reparameterization* (RP) gradient estimator (Heess et al., 2015):

$$\hat{\nabla}_{\boldsymbol{\theta}} \hat{J}(\pi_{\boldsymbol{\theta}}) = \nabla_{\boldsymbol{\theta}} \left( Q_{\mathbf{w}}(S_t, f_{\boldsymbol{\theta}}(\epsilon_t; S_t)) \right) - \alpha \log \pi_{\boldsymbol{\theta}}(f_{\boldsymbol{\theta}}(\epsilon_t; S_t)|S_t). \tag{5}$$

*Gaussian policies* are a common choice when the action space is continuous:

$$\pi_{\boldsymbol{\theta}}(a|s) = \mathcal{N}(a; \mu_{\boldsymbol{\theta}}(s), \sigma_{\boldsymbol{\theta}}(s)^2), \tag{6}$$

where $\mu_{\boldsymbol{\theta}}(s)$ is the mean and $\sigma_{\boldsymbol{\theta}}(s)$ is the standard deviation. Gaussian policies have infinite support, but the action space is typically bounded in practice. To address the bias of clipping actions, *squashed Gaussian policies* use $\tanh$ to transform the unbounded support to a bounded interval:

$$\pi_{\boldsymbol{\theta}}(a|s) = \mathcal{N}(\tanh^{-1}(a); \mu_{\boldsymbol{\theta}}(s), \sigma_{\boldsymbol{\theta}}(s)^2) \quad \text{for } a \in [0, 1]. \tag{7}$$

In this paper, we study *mixture policies* with $N \in \mathbb{N}^+$ components:

$$\pi_{\boldsymbol{\theta}}^m(a|s) = \sum_{k=1}^N \pi_{\boldsymbol{\theta}}^w(k|s)\pi_{\boldsymbol{\theta}}^b(a|s, k),$$

where $\pi_{\boldsymbol{\theta}}^w$ is the *weighting policy* and $\pi_{\boldsymbol{\theta}}^b$ with different $k$ are the *component policies*. When needed, we also explicitly write $\pi_{\boldsymbol{\theta}^m}^m(a|s) = \sum_{k=1}^N \pi_{\boldsymbol{\theta}^w}^w(k|s)\pi_{\boldsymbol{\theta}_k^b}^b(a|s)$, where $\boldsymbol{\theta}^m = \left[\boldsymbol{\theta}_1^{b\top}, \cdots, \boldsymbol{\theta}_N^{b\top}, \boldsymbol{\theta}^{w\top}\right]^\top$. The weighting policy is usually parameterized as a softmax policy, while the component policies may be any continuous policies. When the component policies are Gaussian (Equation (6)), we refer to the resulting policy as the *Gaussian mixture (GM) policy*. In this case, we call the Gaussian policy the *base policy*. Similarly, when the base policy is a squashed Gaussian (Equation (7)), the resulting mixture policy is the *squashed Gaussian mixture (SGM) policy*. While we focus on Gaussian-based policies for certain analysis and empirical evaluation in our papers, many of our results also apply to other policy classes.

## 3 ROBUSTNESS OF MIXTURE POLICIES TO ENTROPY REGULARIZATION

Policy parameterization influences the set of stationary points of the entropy-regularized objective in Equation (3), which is non-concave (Agarwal et al., 2019). We first show that mixture policies improve solution quality, in terms of achieving higher regularized and unregularized objective values under entropy-constrained optimization. Then we show that mixtures are robust to higher levels of entropy, both theoretically and empirically. Section C contains proofs for all theoretical results.

### 3.1 OPTIMALITY OF STATIONARY POINTS

We first show that the optimal stationary points, namely $\boldsymbol{\theta}^* \doteq \arg\max_{\boldsymbol{\theta} \in \{\boldsymbol{\theta}|\nabla_{\boldsymbol{\theta}}J(\pi_{\boldsymbol{\theta}})=\mathbf{0}\}} J(\pi_{\boldsymbol{\theta}})$, of the mixture policy is at least as good as or better than the base policy in Proposition 3.1.

**Proposition 3.1.** *When both $\pi_{\boldsymbol{\theta}^{b,*}}^b$ and $\pi_{\boldsymbol{\theta}^{m,*}}^m$ exist, then $J(\pi_{\boldsymbol{\theta}^{m,*}}^m) \geq J(\pi_{\boldsymbol{\theta}^{b,*}}^b)$.*

The inequality is likely strict when the return landscape is multimodal, as the mixture policy can maintain high returns while increasing its entropy by splitting its density into different modes.

The next natural question is how their optimal stationary points compare regarding the unregularized objective (the expected return) $J_0(\pi_{\boldsymbol{\theta}})$. In general, it is difficult to guarantee $J_0(\pi_{\boldsymbol{\theta}^{m,*}}^m) \geq J_0(\pi_{\boldsymbol{\theta}^{b,*}}^b)$. However, when the entropy is imposed as a constraint instead of regularization, the mixture policy is guaranteed to be at least as good as the base policy, as shown in Proposition 3.2.

**Proposition 3.2.** *Consider entropy-constrained policy optimization $\max_{\boldsymbol{\theta}} J_0(\pi_{\boldsymbol{\theta}})$ subject to $\mathcal{H}(\pi_{\boldsymbol{\theta}}) \geq H$ for some $H > 0$ and define the optimal solution as $\boldsymbol{\theta}'$, then $J_0(\pi_{\boldsymbol{\theta}^{m,\prime}}^m) \geq J_0(\pi_{\boldsymbol{\theta}^{b,\prime}}^b)$.*

### 3.2 NON-EXISTENCE OF STATIONARY POINTS UNDER STRONG ENTROPY REGULARIZATION

This section shows that the mixture policy may have stationary points in scenarios where the base policy does not. We focus on the bandit setting, in which case, the objective degenerates to

$$J(\pi_{\boldsymbol{\theta}}) = \mathbb{E}_{a \sim \pi_{\boldsymbol{\theta}}}[r(a) - \alpha \log \pi_{\boldsymbol{\theta}}(a)]. \tag{8}$$

Proposition 3.3 shows that for sufficiently large entropy scales, there are no stationary points for Gaussian policies.

**Proposition 3.3.** *Assume $r : \mathcal{A} \to \mathbb{R}$ is an integrable function on $\mathcal{A} = \mathbb{R}$. For all $\alpha > \frac{3}{2}r_{\max}$, $J(\pi_{\mu,\sigma}) = \mathbb{E}_{a \sim \mathcal{N}(\mu,\sigma)}[r(a) - \alpha \log \mathcal{N}(a; \mu, \sigma)]$ does not have any stationary point.*

On the other hand, Gaussian mixture (GM) policies are less sensitive to entropy regularization. Specifically, we show that for every stationary point of the regularized objective with a base policy, there exists a corresponding set of stationary points for the mixture policy.

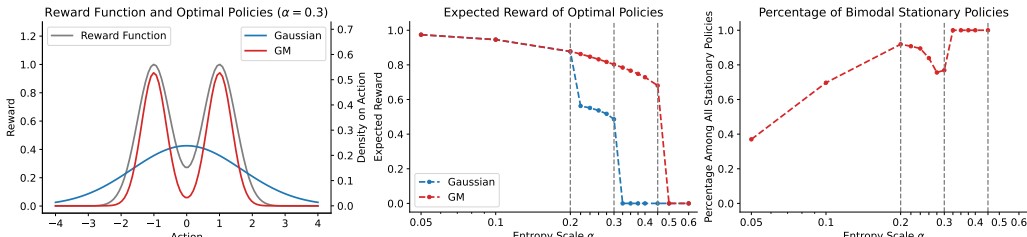

Figure 1: Stationary points of Gaussian and two-component Gaussian Mixture (GM) policies in a bimodal bandit. We use local gradient-based optimization on Equation (8) (100 trials; details in Appendix D). **Left:** Bandit reward function and optimal policies at $\alpha = 0.3$. **Middle:** Expected reward of optimal policies at different $\alpha$; zero means no stationary points found. **Right:** Percentage of bimodal policies among stationary GM policies. Dashed vertical lines mark different levels of entropy regularization, each producing qualitatively distinct effects on Gaussian and GM policies.

**Proposition 3.4.** *The minimum $\alpha$ after which the mixture policy no longer has a stationary point is at least as large as that of the base policy, i.e., $\alpha_{\min}^{\pi^m} \geq \alpha_{\min}^{\pi^b}$, where $\alpha_{\min}^{\pi} = \inf\{\alpha \mid \nabla_{\boldsymbol{\theta}} J(\pi_{\boldsymbol{\theta}}) \neq \mathbf{0}, \forall \boldsymbol{\theta}\}$ for policy $\pi_{\boldsymbol{\theta}}$.*

*Remark* 3.5. While we only compare the mixture policy with the base policy in Propositions 3.1, 3.2 and 3.4, the results can be generalized to compared mixture policies with different $N$.

In fact, instances where the inequality in Proposition 3.4 is strict are not hard to construct. The bimodal bandit in Figure 1 (Left) is one such example.

**Robustness to entropy regularization.** Figure 1 (Middle) plots the expected reward $J_0(\pi_{\boldsymbol{\theta}^*})$ of the optimal stationary points for the two policies. When $\alpha \geq 0.325$, the Gaussian policy admits no stationary points, while the GM policy retains stationary points up to $\alpha \leq 0.45$. Moreover, for $\alpha \in [0.225, 0.3]$, the optimal stationary GM policy outperforms its Gaussian counterpart. In this regime, the GM policy captures both modes of the reward function with two separate components, while the Gaussian policy collapses to a single mode centered between them (Figure 1, Left).

**Preference for multimodality increases with larger entropy regularization.** Figure 1 (Right) shows a clear trend: as $\alpha$ increases, the frequency of bimodal policies also increases. This suggests that a larger entropy scale helps prevent mode collapse in mixture policies, i.e., the loss of multimodality when the policy becomes effectively unimodal.

## 4 THE MARGINALIZED REPARAMETERIZATION (MRP) ESTIMATOR

While there is no general guarantee that the RP estimator is better than the LR estimator, it is shown to have lower variance under certain assumptions (Xu et al., 2019), be more robust in high dimensional settings (Mohamed et al., 2020), and perform better empirically (Mohamed et al., 2020; Zhang et al., 2024). Further, the LR estimator often requires a baseline to reduce variance for it to work well. In RL, this baseline is typically either a learned state-value function or the average of sampled estimated action values. The former requires an additional function approximator, adding complexity and making the algorithm more sensitive, while the latter is prone to instability in high-dimensional settings, as we demonstrate in Section F.1. Thus, the RP estimator is often preferred in high-dimensional continuous control (Haarnoja et al., 2018a; Fujimoto et al., 2018).

Despite this advantage, reparameterization for mixture policies remains underexplored, as their soft-max weighting policies are not directly reparameterizable. Prior work circumvents this issue by either using a fixed weighting policy (Baram et al., 2021) or learning the weighting policy with a separate objective (Hou et al., 2020). As a result, little is known about how to reparameterize mixture policies without sacrificing flexibility or altering the algorithm—a gap we address next.

### 4.1 DERIVING THE MRP ESTIMATOR

We first extend the reparameterization policy gradient theorem (Lan et al., 2022) to the case of mixture policies where we reparameterize only the components in Theorem 4.3. Here, we assume

the component policies are reparameterized as $\pi_{\boldsymbol{\theta}}^b(a|s, k) = p(\epsilon)$, where $a = f_{\boldsymbol{\theta}}(\epsilon; s, k)$. We also define the (discounted) occupancy measure under $\pi$ as $d_\pi(s) \doteq \mathbb{E}_\pi[\sum_{t=0}^\infty \gamma^t \mathbb{I}(S_t = s)]$, which quantifies the expected discounted state visitation frequency under $\pi$.

**Assumption 4.1.** $\mathcal{S}$ and $\mathcal{A}$ are compact.

**Assumption 4.2.** $p(s'|s, a)$, $d_0(s)$, $r(s, a)$, $f_{\boldsymbol{\theta}}(\epsilon; s, k)$, $f_{\boldsymbol{\theta}}^{-1}(a; s, k)$, $\pi_{\boldsymbol{\theta}}^w(k|s)$, $\pi_{\boldsymbol{\theta}}^b(a|s, k)$, $p(\epsilon)$, and their derivatives are continuous in variables $s$, $a$, $s'$, $\boldsymbol{\theta}$, and $\epsilon$.

**Theorem 4.3** (Entropy-Regularized Half-Reparameterization Policy Gradient Theorem). *Under Assumptions 4.1 and 4.2, we have*

$$\nabla_{\boldsymbol{\theta}} J(\pi_{\boldsymbol{\theta}}^m) = \mathbb{E}_{s \sim d_{\pi_{\boldsymbol{\theta}}^m}, k \sim \pi_{\boldsymbol{\theta}}^w(\cdot|s), \epsilon \sim p}\Big[\nabla_{\boldsymbol{\theta}} \log \pi_{\boldsymbol{\theta}}^w(k|s)\big(Q_{\pi_{\boldsymbol{\theta}}^m}(s, f_{\boldsymbol{\theta}}(\epsilon; s, k)) - \alpha \log \pi_{\boldsymbol{\theta}}^m(f_{\boldsymbol{\theta}}(\epsilon; s, k)|s)\big)$$
$$+ \nabla_{\boldsymbol{\theta}} f_{\boldsymbol{\theta}}(\epsilon; s, k)\nabla_a\big(Q_{\pi_{\boldsymbol{\theta}}^m}(s, a) - \alpha \log \pi_{\boldsymbol{\theta}}^m(a|s)\big)|_{a = f_{\boldsymbol{\theta}}(\epsilon; s, k)}\Big].$$

Similarly, we can obtain the *half-reparameterization* (HalfRP) estimator for SAC's objective:

$$\hat{\nabla}_{\boldsymbol{\theta}} \hat{J}(\pi_{\boldsymbol{\theta}}^m) = \nabla_{\boldsymbol{\theta}} \log \pi_{\boldsymbol{\theta}}^w(K_t|S_t)\big(Q_{\mathbf{w}}(S_t, f_{\boldsymbol{\theta}}(\epsilon_t; S_t, K_t)) - \alpha \log \pi_{\boldsymbol{\theta}}^m(f_{\boldsymbol{\theta}}(\epsilon_t; S_t, K_t)|S_t)\big)$$
$$+ \nabla_{\boldsymbol{\theta}}\big(Q_{\mathbf{w}}(S_t, f_{\boldsymbol{\theta}}(\epsilon_t; S_t, K_t)) - \alpha \log \pi_{\boldsymbol{\theta}}^m(f_{\boldsymbol{\theta}}(\epsilon_t; S_t, K_t)|S_t)\big). \tag{9}$$

While the HalfRP estimator still suffers from high variance (see Section 6), we propose a better alternative by marginalizing over the mixing weights in the HalfRP estimator–in other words, we marginalize the random variable $K_t$. This gives rise to a new policy gradient theorem and estimator.

**Theorem 4.4** (Entropy-Regularized Marginalized-Reparameterization Policy Gradient Theorem). *Under Assumptions 4.1 and 4.2, we have*

$$\nabla_{\boldsymbol{\theta}} J(\pi_{\boldsymbol{\theta}}^m) = \mathbb{E}_{s \sim d_{\pi_{\boldsymbol{\theta}}^m}, \epsilon \sim p}\left[\nabla_{\boldsymbol{\theta}} \sum_{k=1}^N \pi_{\boldsymbol{\theta}}^w(k|s)\big(Q_{\pi_{\boldsymbol{\theta}}^m}(s, f_{\boldsymbol{\theta}}(\epsilon; s, k)) - \alpha \log \pi_{\boldsymbol{\theta}}^m(f_{\boldsymbol{\theta}}(\epsilon; s, k)|s)\big)\right].$$

The corresponding *marginalized-reparameterization* (MRP) estimator for SAC's objective is

$$\hat{\nabla}_{\boldsymbol{\theta}} \hat{J}(\pi_{\boldsymbol{\theta}}^m) = \nabla_{\boldsymbol{\theta}} \sum_{k=1}^N \pi_{\boldsymbol{\theta}}^w(k|S_t)\big(Q_{\mathbf{w}}(S_t, f_{\boldsymbol{\theta}}(\epsilon_t; S_t, k)) - \alpha \log \pi_{\boldsymbol{\theta}}^m(f_{\boldsymbol{\theta}}(\epsilon_t; S_t, k)|S_t)\big). \tag{10}$$

To our knowledge, Equations (9) and (10) provide the first two unbiased RP-based gradient estimators for mixture policies. In the next section, we establish conditions for a provably lower variance of the MRP estimator compared to the LR estimator.

## 4.2 Variance Reduction Properties of the MRP Estimator

For clarity, we focus on the bandit setting without regularization: $J(\pi_{\boldsymbol{\theta}}) = \mathbb{E}_{a \sim \pi_{\boldsymbol{\theta}}}[r(a)]$. This simple setting allows us to isolate the effect of mixture reparameterization and provide clean, interpretable variance comparisons. The entropy term can be included by redefining $r(a)$ to be the sum of the reward and sample entropy, while the extension to the MDP setting is straightforward by considering state-conditioned variance.

We analyze the gradient of the MRP and LR estimators with respect to the distribution parameters and the weighting probabilities of Gaussian mixture policies. In this case, the mixture policy can be expressed as $\pi_{\boldsymbol{\theta}}^m(a) = \sum_{k=1}^N \pi_{\boldsymbol{\theta}^w}^w(k)\pi_{\boldsymbol{\theta}^{b_k}}^b(a) = \sum_{k=1}^N w_k \mathcal{N}(a; \mu_k, \sigma_k^2)$ with $\boldsymbol{\theta} = \big[\boldsymbol{\theta}^{b_1\top}, \cdots, \boldsymbol{\theta}^{b_N\top}, \boldsymbol{\theta}^{w\top}\big]^\top = \big[[\mu_1, \sigma_1]^\top, \cdots, [\mu_N, \sigma_N]^\top, [w_1, \cdots, w_N]^\top\big]^\top \in \mathbb{R}^{3N}$, where $\mu_k$, $\sigma_k$, and $w_k$ may depend on shared parameters implicitly. The corresponding estimators are

$$\text{LR: } \hat{\nabla}_{\boldsymbol{\theta}}^{\text{LR}} J(\pi_{\boldsymbol{\theta}}^m) = \nabla_{\boldsymbol{\theta}} \log \pi_{\boldsymbol{\theta}}^m(A)r(A), \ \text{MRP: } \hat{\nabla}_{\boldsymbol{\theta}}^{\text{MRP}} J(\pi_{\boldsymbol{\theta}}^m) = \nabla_{\boldsymbol{\theta}} \sum_{k=1}^N \pi_{\boldsymbol{\theta}^w}^w(k)r\big(f_{\boldsymbol{\theta}^{b_k}}(\epsilon)\big),$$

where $A \sim \pi_{\boldsymbol{\theta}}^m(\cdot)$, $\epsilon \sim \mathcal{N}(0, 1)$, $\pi_{\boldsymbol{\theta}^w}^w(k) = w_k$, and $f_{\boldsymbol{\theta}^{b_k}}(\epsilon) = \mu_k + \epsilon\sigma_k$. Define $\pi^m \doteq \pi_{\boldsymbol{\theta}}^m$, $\pi^{b_k} \doteq \pi_{\boldsymbol{\theta}^{b_k}}^b$, $\hat{\partial}_{\theta_i^{b_k}} \cdot \doteq \big[\hat{\nabla}_{\boldsymbol{\theta}^{b_k}} \cdot\big]_i$, and $\hat{\partial}_{w_k} \cdot \doteq \big[\hat{\nabla}_{\boldsymbol{\theta}^w} \cdot\big]_k$, where $\theta_i^{b_k}$ and $w_k$ are the $i$-th distribution parameter and the weight of the $k$-th component policy, respectively. For Gaussian component policies, $\theta_i^{b_k} \in \{\mu_k, \sigma_k\}$. We present MRP's variance reduction properties in Proposition 4.7.

**Assumption 4.5.** $r : \mathbb{R} \to \mathbb{R}$ is twice differentiable with first and second derivatives $r'$ and $r''$. For all $k \in \{1, \cdots, N\}$ and $\varphi(a) \in \{\phi_1(a) = (a - \mu_k)r(a), \psi_1(a) = r'(a), \phi_2(a) = ((a - \mu_k)^2 - \sigma_k^2)r(a), \psi_2(a) = (a - \mu_k)r'(a)\}$, $\varphi$ has finite variance and its first derivative $\varphi'$ is absolutely integrable under $\pi^{b_k}$: $\mathbb{V}_{\pi^{b_k}(A)}(\varphi(A)) < \infty$ and $\mathbb{E}_{\pi^{b_k}(A)}[|\varphi'(A)|] < \infty$, with $\pi^{b_k}(a) = \mathcal{N}(a; \mu_k, \sigma_k)$. Further, it holds for $i \in \{1, 2\}$ that

$$\sum_{k=1}^{N} \left( \mathbb{E}_{\pi^{b_k}(A)}[\phi_i'(A)]^2 - \sigma_k^4 \mathbb{E}_{\pi^{b_k}(A)}[\psi_i'(A)^2] \right) \geq 0.$$

**Assumption 4.6.** Define $\rho_m^{b_k}(A) = \frac{\pi^{b_k}(A)}{\pi^m(A)}$, the sum of the variance of the importance-sampling LR estimators over all components is larger than that of the on-policy LR estimators:

$$\sum_{k=1}^{N} \left( \mathbb{V}_{\pi^m(A)}\left(\rho_m^{b_k}(A)\hat{\partial}_{\theta_i^{b_k}}^{\mathrm{LR}} J(\pi_{\boldsymbol{\theta}^{b_k}}^b)\right) - \mathbb{V}_{\pi^{b_k}(A)}\left(\hat{\partial}_{\theta_i^{b_k}}^{\mathrm{LR}} J(\pi_{\boldsymbol{\theta}^{b_k}}^b)\right) \right) \geq 0 \quad \text{for} \quad \theta_i^{b_k} \in \{\mu_k, \sigma_k\},$$

$$\sum_{k=1}^{N} \left( \mathbb{V}_{\pi^m(A)}\left(\rho_m^{b_k}(A)r(A)\right) - \mathbb{V}_{\pi^{b_k}(A)}\left(r(A)\right) \right) \geq 0.$$

**Proposition 4.7.** *Under Assumptions 4.5 and 4.6, the trace of the covariance matrix of the MRP estimator is smaller than that of the LR estimator:*

$$\mathrm{Tr}\left(\mathbb{C}_{\mathcal{N}(\epsilon; 0, 1)}(\hat{\nabla}_{\boldsymbol{\theta}}^{MRP} J(\pi_{\boldsymbol{\theta}}^m))\right) \leq \mathrm{Tr}\left(\mathbb{C}_{\pi^m(A)}(\hat{\nabla}_{\boldsymbol{\theta}}^{LR} J(\pi_{\boldsymbol{\theta}}^m))\right).$$

*Remark* 4.8. Proposition 4.7 builds on the marginal variance bound for the mean derivative estimator of Gaussians in Gal (2016). Here, we extend it to the standard deviation derivative estimator and Gaussian mixtures. Instead of deriving a bound for marginal variance as in Gal (2016), we derive a bound for the traces of the estimators' covariance matrices for two reasons. First, the trace provides a single scalar value that summarizes the multivariate variability of the gradient, which is common in the literature (Miller et al., 2017; Xu et al., 2019). Second, by directly targeting the trace, we can relax the assumptions so that the required conditions do not need to hold for each individual component. For example, in Assumption 4.6, it is possible that sampling from the mixture policy $\pi^m$ might reduce variance for some of its components but the total variance across all components is still higher when some components induce dominating high-variance.

*Remark* 4.9. Proposition 4.7 is under the univariate setting, but can be extended to the multivariate case. For example, we can extend the analysis of the multivariate case for Gaussians in Xu et al. (2019) by using Lemmas C.6 and C.7 in Appendix C.

While Proposition 4.7 is derived for the bandit setting with univariate actions for simplicity, the practical effect of reduced gradient variance and improved stability holds empirically in high-dimensional RL tasks, as demonstrated in seven MuJoCo tasks in Appendix F.1.

## 5 EXPERIMENTS ON THE UTILITY OF MIXTURE POLICIES

We empirically compare mixture and base policies in this section, while investigating different gradient estimators for mixture policies in Section 6. We use SAC as the base learning algorithm and the squashed Gaussian policy as the base policy (see Appendix F.4 for results using a heavy-tailed base policy). We denote different SAC instances using X-Y, where X represents the policy's parameterization, and Y represents the gradient estimator. For squashed Gaussian (SG) policies, we consider the RP estimator (SG-RP). For squashed Gaussian mixture (SGM) policies, we consider the MRP estimator (SGM-MRP). We use 5 components for mixture policies in all our experiments (see Appendix F.3 for an ablation study). Unless otherwise noted, error bars and shaded regions indicate 95% bootstrap confidence intervals.

### 5.1 DO MIXTURE POLICIES HELP FIND HIGHER PEAKS IN THE CRITIC?

We first investigate the following question: *When the critic landscape is multimodal, do mixture policies help find better peaks?* To avoid confounding factors, such as nonstationarity of the critic and the impact of the actor on the critic, we focus on the bandit setting, where the critic (i.e., the reward function) is stationary and given to the agent. Specifically, we create 100 continuous bandits using density functions of randomly generated Gaussians. We sweep the initial actor step size and the entropy scale, running each setting on each bandit for 10 runs. We report results of the best

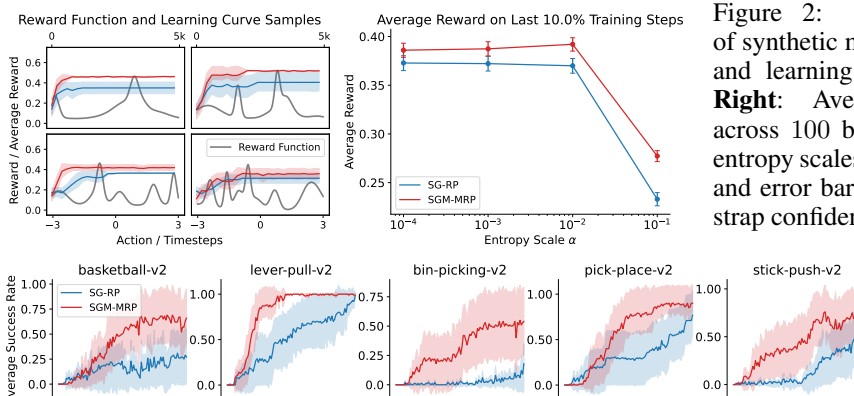

Figure 2: **Left**: Examples of synthetic multimodal bandits and learning curves on them. **Right**: Average performance across 100 bandits at different entropy scales. All shaded areas and error bars show 95% bootstrap confidence intervals (CIs).

Figure 3: Learning curves in selected MetaWorld tasks where the performance gap between mixture policies and base policies is most pronounced. Shaded areas show 95% bootstrap CIs over 10 runs.

setting that has the highest average reward over the last $10.0\%$ training steps, as we are mostly interested in how well the agent explores. Figure 2 (Left) shows a few examples of the generated bandits and the learning curves corresponding to them. More details can be found in Appendix E.1.

**Mixture policies explore more efficiently.** Figure 2 (Right) shows the aggregated final performance of base policies (SG-RP) and mixture policies (SGM-MRP). We can see that mixture policies are better than base policies across different $\alpha$. From Figure 2 (Left), we can see that mixture policies can indeed find higher peaks in the critic compared to base policies.

**Mixture policies also improve robustness.** Another observation from Figure 2 (Right) is that the mixture policy is more robust to the entropy scale, which is consistent with our results presented in Section 3. Specifically, other than the consistent improvement of the mixture policy over the base policy, we can see that the gap between them increase as $\alpha$ increases. This result suggests that mixture policies are possibly more robust to larger entropy scales and explore more efficiently with a moderate large entropy scale. This is not the case for the base policy in this experiment.

## 5.2 Do Mixture Policies Help in Continuous Control Benchmarks?

We next move on to the full RL setting: *Do mixture policies improve performance in common continuous control benchmarks?* Specifically, we investigate the performance of mixture policies in 57 environments from four common continuous control benchmarks: 7 from OpenAI Gym MuJoCo (Brockman et al., 2016), 10 from DeepMind Control (DMC) Suite (Tassa et al., 2018), 30 from MetaWorld (Yu et al., 2020), and 10 from MyoSuite (Caggiano et al., 2022). Performance is measured using the average (normalized) return in Gym MuJoCo and DM Control or success rate in MetaWorld and MyoSuite. We use SAC with *automatic entropy tuning* and the hyperparameters reported in the second SAC paper (Haarnoja et al., 2018c), which are tuned based on SG-RP. We use 10 random seeds for each environment. Please refer to Appendix E.1 for more details.

**Mixture policies consistently match or surpass base policies without hyperparameter tuning.** Figure 4 reports the performance gap between SGM-MRP and SG-RP across four benchmarks. Overall, mixture policies perform comparably to base policies and often achieve slightly higher average performance. The gains are pronounced in MetaWorld, where mixture policies more often outperform base policies by a large margin. Figure 3 shows the learning curves in MetaWorld environments where the performance gap is larger than $20\%$. Learning curves on all tasks are in Figures 13 and 14.

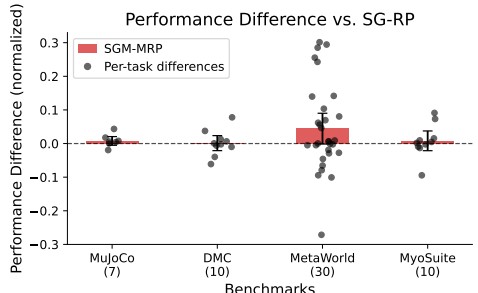

Figure 4: Performance difference between SGM-MRP and SG-RP across common benchmarks.

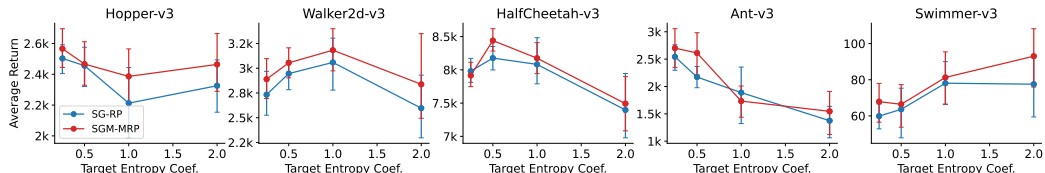

Figure 5: Performance with different target entropy coefficients for mixture policies and base policies in five MuJoCo environments. Results are averaged over 10 runs.

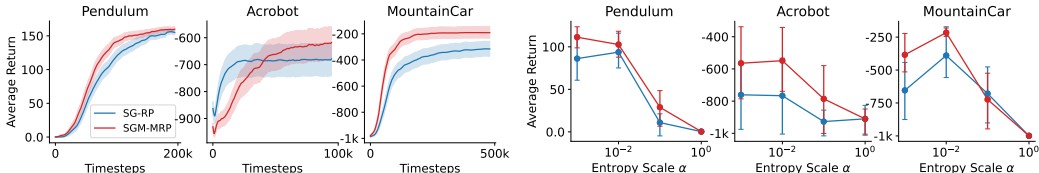

Figure 6: Learning and sensitivity curves for classic control environments with unshaped rewards. Learning curves are averaged over 200 runs, whereas sensitivity curves are over 10 runs.

We also evaluate additional target-entropy values in MuJoCo environments, and the results consistently show that mixture policies outperform base policies (see Figures 5 and 16).

### 5.3 Do Mixture Policies Help With Unshaped Rewards?

While mixture policies do not appear to be generally beneficial on common benchmarks, we want to understand when mixture policies may be more helpful. Since most tasks in those benchmarks have shaped rewards, we hypothesize that *mixture policies are more helpful in environments with unshaped rewards compared to those with shaped rewards*. We next test this hypothesis.

To perform a more extensive empirical investigation with proper hyperparameter tuning, we use three classic control environments as they require less computation resource: Pendulum (Degris et al., 2012), Acrobot (Sutton & Barto, 2018), and MountainCar (Sutton & Barto, 2018). Specifically, we use two different variants for each of them: one with shaped rewards and the other one with unshaped rewards. We refer the reader to Section E.5 about their versions and specific reward functions. We use SAC with a fixed entropy scale as it is reported that SAC with automatic entropy performs worse in this domain (Neumann et al., 2022). We sweep the entropy scale, the initial critic step size, and the initial actor step size, running each setting for 10 runs. We report another 30 reruns for environments with shaped rewards and 200 reruns for those with unshaped rewards of the best setting that has the largest area under the learning curve. See Appendix E.1 for more details.

**Mixture policies improve exploration when rewards are unshaped.** Figure 6 shows results in environments with unshaped rewards, where mixture policies substantially outperform base policies and consistently achieve higher returns across entropy scales. In contrast, performance in shaped-reward environments is more stable, and the gap between the two is much smaller (see Figure 17).

**Visualization of state visitation.** To quantitatively demonstrate the exploration efficiency of mixture policies, we measure the state visitation during early training in `MountainCar`, where the performance gain is significant. In Figure 7, the difference in visitation on states far away from the starting state at $(-0.5, 0.0)$ demonstrates the better exploration efficiency of mixture policies. See Appendix E.6 for details.

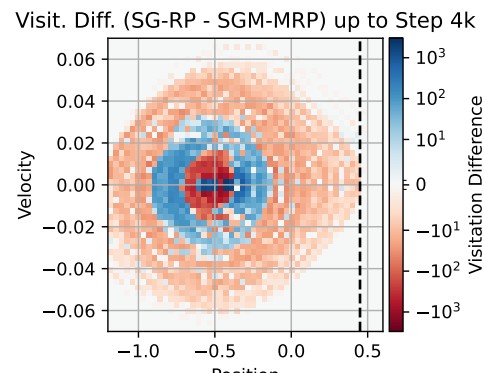

Figure 7: Difference in state visitation between SG-RP and SGM-MRP in `MountainCar`. Red indicates higher visitation by SGM-MRP, whereas blue reflects higher visitation by SG-RP.

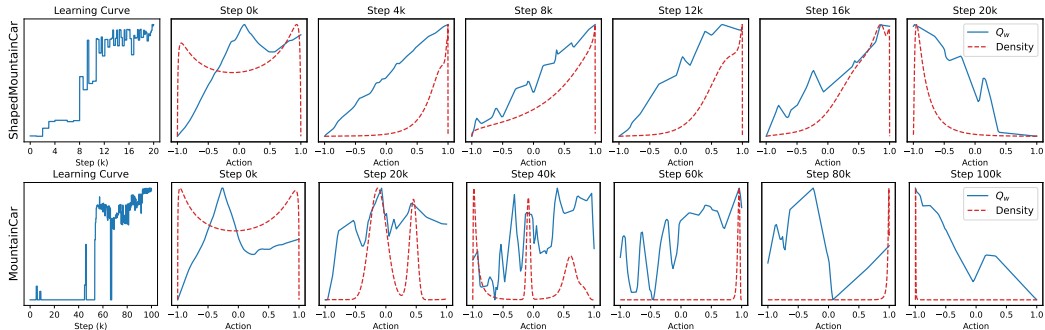

Figure 8: Learning curves, action-value estimates, and policy densities at a starting state from a sample run of SGM-MRP in two MountainCar variants. The y-axes differ across plots and are not shown with ticks to highlight the shape of the curves rather than the values. The learning curves show faster convergence and smoother performance in `ShapedMountainCar` with shaped rewards, whereas unshaped rewards in `MountainCar` lead to more erratic and slower learning. The action-value estimates are smoother and more stable in `ShapedMountainCar`. In contrast, they are more unstable and multimodal in `MountainCar`. Correspondingly, the density quickly becomes unimodal and concentrates on one of the boundaries in the former, while multimodal density has more occurrence in the latter, reflecting continued exploration of the mixture policy.

**Visualization of action-value estimates and policy density.** To understand why mixture policies deviate more from base policies in unshaped-reward settings, we examine an SGM-MRP agent in `MountainCar` and `ShapedMountainCar` (Figure 8). In both environments, the agent must decide which direction to accelerate to gain momentum (Figure 20). In `MountainCar`, where rewards are unshaped, the critic is less smooth and mixture policies exhibit greater multimodality, preserving multiple action modes that support bidirectional exploration. In contrast, shaped rewards in `ShapedMountainCar` yield smoother critics and policies with fewer modes, reducing the need for exploration. This helps explain why mixture policies, which enhance exploration (Section 5.1), show limited benefits in common benchmark environments with shaped rewards (Section 5.2).

## 6 EXPERIMENTS ON GRADIENT ESTIMATORS FOR MIXTURE POLICIES

In this section, we compare our proposed MRP estimator for mixture policies against a prior approach that fixes the mixing weights to avoid reparameterizing the weighting policy (Baram et al., 2021). Following Baram et al., we use uniform mixing weights and denote this baseline as USGM-RP. In addition to the MRP estimator, we also evaluate the HalfRP estimator (Equation (9)) and the GumbelRP estimator. The GumbelRP estimator uses the Gumbel-Softmax trick (Jang et al., 2016) to obtain biased reparameterization samples for the discrete weighting policy. Please refer to Appendix B for more details. We compare them in synthetic multimodal bandits, classic control, and three common continuous control benchmarks. For each setting, we follow the same experimental setup as

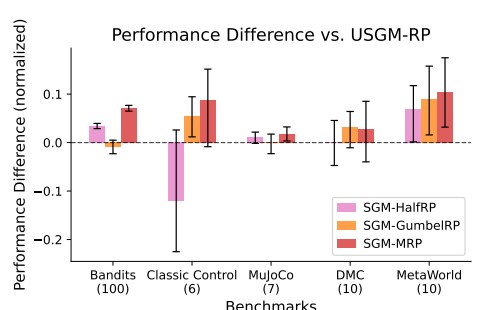

Figure 9: Summarized performance over Gym MuJoCo, DeepMind Control, and MetaWorld.

Section 5. Figure 9 shows the summarized results. Learning curves and more plots can be found in Appendix E.

**The MRP estimator is the most reliable overall.** Across all settings, SGM-MRP consistently performs best, in some cases significantly outperforming both the baseline and other estimators. SGM-GumbelRP is competitive in classic control and continuous control benchmarks but performs worst in multimodal bandits, likely because the bias in its gradient is more pronounced in this simpler

setting. Conversely, the unbiased SGM-HalfRP does well in multimodal bandits but fails badly in some classic control environments, suggesting that its high variance hinders stable learning.

## 7 CONCLUSIONS

Mixture policies are a simple way to increase the flexibility of the policy parameterization, but very little has been documented about their efficacy, or lack of efficacy. Our aim was to start to fill this gap, to make this a more accessible tool when using entropy-regularized actor-critic algorithms with continuous actions, like Soft-Actor Critic (SAC). The clear outcome from the study is that mixture policies are comparable, and sometimes better than, a base unimodal policy. Through a few basic theoretical results and experiments in bandits, we highlighted that mixture policies are more robust to entropy scale, with 1) a preference for multimodality increasing with higher entropy, 2) divergence (lack of stationary points) for the base Gaussian policy, unlike the mixture, 3) better balance between entropy and the expected reward objective, resulting in higher unregularized values in addition to higher regularized values and 4) higher likelihood of finding maxima on a multimodal surface. This behavior seemed to manifest in better exploration in environments with unshaped, or uninformative rewards; in such environments, without shaped rewards, exploration is critical and the mixture policies performed better than the unimodal base policy. In particular, we found the base policy had more failed runs where it was unable to find the goal at all.

To leverage the utility of mixture policies, however, we needed a small algorithmic improvement: reparameterization gradients. We proposed a new reparameterization gradient estimator for mixture policies that reduces variance effectively, filling in a gap in the literature. To derive the new estimator, we first derive a half-reparameterization, only reparameterizing the component policies and not the softmax weighting policy. By further marginalizing the mixing weights, we obtained the desired estimator, which is proven to have lower variance than the LR estimator and shown to perform better than all alternatives.

Finally, while mixture policies provide robust or superior performance across domains, they introduce a modest additional computational cost and require choosing the number of mixture components. We provide a concrete analysis of the computational overhead in Appendix E.2 and discuss selection of the number of components in Appendix F.3.

### REPRODUCIBILITY STATEMENT

We will release all code, including implementations of the investigated estimators (HalfRP, GumbelRP, and MRP). Hyperparameter settings and environment details are provided in Appendix E.1. Training curves and aggregated results are reported over multiple seeds with $95\%$ bootstrap confidence intervals. Instructions for reproducing the main results will be included in the code repository.

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

## A  RELATED WORKS

In this section, we provide a more in-depth discussion of related work.

**Implicit policies.**  Existing works on modeling continuous distributions on continuous action spaces can be put into two categories: Policies using parametric distributions and implicit policies. We have discussed the former and touched on the latter in the introduction of our paper. Here, we further discuss implicit policies. Implicit policies utilize deep generative models (e.g., energy-based models; Haarnoja et al., 2017; Messaoud et al., 2024; normalizing flows; Tang & Agrawal, 2018; Mazoure et al., 2020; diffusion models; Wang et al., 2023) to model the policy. These models can model complex distributions and have improved learning efficiency, but they usually have more parameters and complex training pipelines. Compared to these more complex policies, policies using parametric distributions have several benefits. Firstly, they are simpler and more efficient to train. Secondly, they have simple explicit probability density, which is useful in various ways in entropy-regularized RL. Note that some implicit policies do not hold such a property. Further, mixture policies, which we consider in our paper, also have modeling power to model arbitrary distribution given a sufficiently large number of components. Given these benefits, we think it is important to improve our understanding of mixture policies. Nevertheless, we agree that it is also important to investigate more complex but powerful implicit policies, and the benefits of mixture policies demonstrated in our paper can potentially be generalized to them.

**Other uses of mixture policies.**  Since mixture distributions are a widely known model, mixture policies have been investigated in various ways in the literature. Daniel et al. (2012) and Celik et al. (2022) focus on exploiting the hierarchy in mixture policies for problems with hierarchical structures. In their work, they design algorithms specific to mixture policies. Sharing the same motivation to use mixture policies to model diverse behaviors, Nematollahi et al. (2022) learn a prior GMM using imitation learning and then use SAC to learn the changes to the prior GMM for adaptation. Here, the action space of SAC is the changes to GMM's parameters. With a different motivation, Seyde et al. (2022) use mixture policies to select from a diverse set of sub-policies to reduce the hyperparameter sensitivity of the algorithm.

**Mixture policies as the policy parameterization for SAC.**  Different from the above previous works, our motivation for using mixture policies is to treat them as a more complex policy class and understand the effect of doing so under the entropy-regularization setting. Thus, our treatment does not include designing specific objective functions for mixture policies but using the standard regularized objective that is agnostic to policy parameterizations. In this regard, the closest related works are Haarnoja et al. (2018b), Hou et al. (2020), and Baram et al. (2021). In the first version of SAC, Haarnoja et al. did indeed test mixture policies but then did not pursue this further nor provide insights on this choice. We hypothesize the lack of reparameterization (RP) gradient estimators for mixture policies might be the reason that later versions of SAC switched to a single Gaussian as they found the RP estimator works better (see Footnote 3 on Page 67 of Haarnoja, 2018). Later, Hou et al. try to avoid reparameterization of the whole mixture policy in SAC by using a separate objective for the weighting policy. However, their evaluation on a restrictive set of MuJoCo environments does not show a significant improvement from their approach. Further, Baram et al. also revisit SAC with a mixture policy. However, their focus is to explore the utility of the upper and lower bounds of mixture models' entropy, without considering a learnable weighting policy.

It is not highly novel to use mixture policies, but rather to understand the effect of doing so. In our work: 1) We provide new insights into the effect of a more flexible policy class on the stationary points of the entropy regularized objective; 2) we propose a new RP gradient estimator for mixture policies with provable variance reduction properties and without compromising flexibility or altering the algorithm, filling in a gap in the literature; and 3) we explore the benefits of and provide insights into using mixture policies in environments without shaped rewards, complementing existing works on using mixture policies in entropy-regularized actor-critic.

# B  THE GUMBEL REPARAMETERIZATION (GUMBELRP) ESTIMATOR

Since the output of the weighting policy $\pi_\theta^w(\cdot|S_t)$ is a discrete random variable $K_t$, we can not directly reparameterize it. In Section 4.1, we avoid using the LR estimator for the weighting policy by marginalizing the discrete random variable $K_t$. Here, we introduce an alternative to the MRP estimator through *biased* reparameterizations of discrete random variables (Bengio et al., 2013; Maddison et al., 2016; Jang et al., 2016). Here, we investigate the straight-through Gumbel-Softmax reparameterization (Jang et al., 2016).

Given a weighting distribution $\pi_\theta^w(\cdot|s)$ and i.i.d samples from Gumbel$(0, 1)$, $g_1, \cdots, g_N$, we can obtain a sample from the corresponding Gumbel-Softmax distribution:

$$y_\theta(\mathbf{g}; s, k) = \frac{\exp((\log \pi_\theta^w(k|s) + g_k)/\tau)}{\sum_{k'=1}^N \exp((\log \pi_\theta^w(k'|s) + g_{k'})/\tau)} \quad \text{for } k = 1, \ldots, N,$$

where we define $\mathbf{g} = [g_1, \cdots, g_N]$, and $\tau$ is a temperature parameter, controlling a bias-variance trade-off. Using the Gumbel-Max trick, we can obtain a sample from $\pi_\theta^w(\cdot|s)$ using $y_\theta(\mathbf{g}; s, k)$:

$$\hat{\mathbf{z}} = \text{one\_hot}\left(\arg\max_k\left(y_\theta(\mathbf{g}; s, k)\right)\right) = \text{one\_hot}\left(\arg\max_k\left(\log \pi_\theta^w(k|s) + g_k\right)\right).$$

We can further use the straight-through trick to obtain a differentiable one-hot sample $\mathbf{z} = [z_\theta(\mathbf{g}; s, 1), \cdots, z_\theta(\mathbf{g}; s, N)]$, where $z_\theta(\mathbf{g}; s, k)$ is defined as follows:

$$z_\theta(\mathbf{g}; s, k) = \hat{z}_k + y_\theta(\mathbf{g}; s, k) - y_\phi(\mathbf{g}; s, k)|_{\phi=\theta} \quad \text{for } k = 1, \ldots, N.$$

Finally, using the differential one-hot sample $\mathbf{z}$ from the weighting policy $\pi_\theta^w$ and reparameterized samples from the component policies, we can obtain a differentiable action sample $a$:

$$a = \sum_{k=1}^N z_\theta(\mathbf{g}; s, k) f_\theta(\epsilon; s, k). \tag{11}$$

Plugging (11) back to (5), we can obtain a full RP estimator, which we call the *Gumbel-reparameterization* (GumbelRP) estimator:

$$\hat{\nabla}_\theta \hat{J}(\pi_\theta^m) = \nabla_\theta\left(Q_\mathbf{w}(S_t, A_t) - \alpha \log \pi_\theta^m(A_t|S_t)\right) \text{ with } A_t = \sum_{k=1}^N z_\theta(\mathbf{g}_t; S_t, k) f_\theta(\epsilon_t; S_t, k). \tag{12}$$

**The temperature parameter $\tau$ controls a bias-variance trade-off.** When $\tau$ approaches 0, the soft sample $\mathbf{y} = [y_\theta(\mathbf{g}; s, 1), \cdots, y_\theta(\mathbf{g}; s, N)]$ will converge to a one-hot vector and recover the categorical sample. However, the variance of the gradients with respect to $\pi_\theta^w(\cdot|s)$ will increase. Conversely, when $\tau$ becomes larger, the variance of the gradients will decrease, but the soft sample $\mathbf{y}$ will converge to a uniform vector. See Jang et al. (2016) for detailed discussions. In our study, we find that using a fixed temperature $\tau = 1$ works well (see Section F.5 for a sensitivity study).

In Table 1, we compare different estimators in terms of their flexibility and bias-variance trade-off.

Table 1: Comparison of gradient estimators for mixture policies. Checkmarks indicate whether the method supports learnable mixture weights, has low variance, and is unbiased. HalfRP has intermediate variance between the LR and MRP estimators. U-RP denotes mixture policies with uniform weightings and the RP estimator.

| Estimator | Learnable weights | Low variance | Unbiased |
|---|---|---|---|
| U-RP (Baram et al., 2021) | ✗ | ✓ | ✓ |
| LR (Williams, 1992) | ✓ | ✗ | ✓ |
| HalfRP (Equation (9)) | ✓ | ± | ✓ |
| GumbelRP (Equation (12)) | ✓ | ✓ | ✗ |
| MRP (Equation (10)) | ✓ | ✓ | ✓ |

## C  PROOFS

### C.1  PROOFS FOR RESULTS IN SECTION 3

**Proposition 3.1.** *When both $\pi_{\boldsymbol{\theta}^{b},*}^{b}$ and $\pi_{\boldsymbol{\theta}^{m},*}^{m}$ exist, then $J(\pi_{\boldsymbol{\theta}^{m},*}^{m}) \geq J(\pi_{\boldsymbol{\theta}^{b},*}^{b})$.*

*Proof.* Since the policy class of $\pi_{\boldsymbol{\theta}^{m}}^{m}$ is a super set of the policy class of $\pi_{\boldsymbol{\theta}^{b}}^{b}$, it is apparent that the optimal value of the policy class of $\pi_{\boldsymbol{\theta}^{m}}^{m}$ is better than that of $\pi_{\boldsymbol{\theta}^{b}}^{b}$. $\qquad\square$

**Proposition 3.2.** *Consider entropy-constrained policy optimization $\max_{\boldsymbol{\theta}} J_0(\pi_{\boldsymbol{\theta}})$ subject to $\mathcal{H}(\pi_{\boldsymbol{\theta}}) \geq H$ for some $H > 0$ and define the optimal solution $\boldsymbol{\theta}'$, then $J_0(\pi_{\boldsymbol{\theta}^{m},\prime}^{m}) \geq J_0(\pi_{\boldsymbol{\theta}^{b},\prime}^{b})$.*

*Proof.* Define $\tilde{\boldsymbol{\theta}}^{m} = [\boldsymbol{\theta}^{b,\prime\top}, \cdots, \boldsymbol{\theta}^{b,\prime\top}, \boldsymbol{\theta}^{w\top}]^{\top}$ for any $\boldsymbol{\theta}^{w}$. Apparently, $\pi_{\tilde{\boldsymbol{\theta}}^{m}}^{m}(a) = \pi_{\boldsymbol{\theta}^{b},\prime}^{b}(a)$. Then,

$$J_0(\pi_{\boldsymbol{\theta}^{m},\prime}^{m}) \geq J_0(\pi_{\tilde{\boldsymbol{\theta}}^{m}}^{m}) = J_0(\pi_{\boldsymbol{\theta}^{b},\prime}^{b}).$$

$\qquad\square$

**Proposition 3.3.** *Assume $r : \mathcal{A} \to \mathbb{R}$ is an integrable function on $\mathcal{A} = \mathbb{R}$. For all $\alpha > \frac{3}{2} r_{\max}$, $J(\pi_{\mu,\sigma}) = \mathbb{E}_{a \sim \mathcal{N}(\mu,\sigma)}[r(a) - \alpha \log \mathcal{N}(a; \mu, \sigma)]$ does not have any stationary point.*

*Proof.* To show that $J(\pi_{\mu,\sigma})$ does not have any stationary point, it is sufficient to show that its partial derivative with respect to $\sigma$ is lower bounded by zero:

$$\frac{\partial J(\pi_{\mu,\sigma})}{\partial \sigma} > 0, \quad \forall \mu \in \mathbb{R}, \sigma > 0. \tag{13}$$

We first simplify the entropy term $H(\mathcal{N}(\cdot; \mu, \sigma))$ for the Gaussian policy:

$$
\begin{aligned}
H(\mathcal{N}(\cdot; \mu, \sigma)) &= -\mathbb{E}_{a \sim \mathcal{N}(\mu,\sigma)}[\log \mathcal{N}(a; \mu, \sigma)] \\
&= -\mathbb{E}_{a \sim \mathcal{N}(\mu,\sigma)}\left[ \log \left( \frac{1}{\sqrt{2\pi\sigma^2}} \exp \left( -\frac{(a-\mu)^2}{2\sigma^2} \right) \right) \right] \\
&= \frac{1}{2}\log(2\pi\sigma^2) + \frac{1}{2\sigma^2}\mathbb{E}_{a \sim \mathcal{N}(\mu,\sigma)}[(a-\mu)^2] \\
&= \frac{1}{2}\log(2\pi\sigma^2) + \frac{1}{2}.
\end{aligned}
$$

Further, we can derive its partial derivative with respect to $\sigma$: $\frac{\partial H(\mathcal{N}(\cdot;\mu,\sigma))}{\partial \sigma} = \frac{1}{\sigma}$.

Define $r(\mu, \sigma) = \mathbb{E}_{a \sim \mathcal{N}(\mu,\sigma)}[r(a)] = \int_a r(a)\mathcal{N}(a|\mu, \sigma)\,da$. Then,

$$
\begin{aligned}
\frac{\partial J(\pi_{\mu,\sigma})}{\partial \sigma} &= \frac{\partial}{\partial \sigma}\mathbb{E}_{a \sim \mathcal{N}(\mu,\sigma)}[r(a) - \alpha \log \mathcal{N}(a; \mu, \sigma)] \\
&= \frac{\partial}{\partial \sigma}\left( r(\mu, \sigma) + \alpha H(\mathcal{N}(\cdot; \mu, \sigma)) \right) \\
&= \frac{\partial r(\mu, \sigma)}{\partial \sigma} + \frac{\alpha}{\sigma}.
\end{aligned}
$$

To show equation 13, we just need to show

$$\sigma \frac{\partial r(\mu, \sigma)}{\partial \sigma} > -\alpha. \tag{14}$$

We first analyze the left hand side of equation 14:

$$\sigma \frac{\partial r(\mu, \sigma)}{\partial \sigma} = \sigma \frac{\partial}{\partial \sigma} \int_a r(a) \mathcal{N}(a|\mu, \sigma) \, da$$

$$= \sigma \int_a r(a) \frac{\partial}{\partial \sigma} \mathcal{N}(a|\mu, \sigma) \, da$$

$$= \sigma \int_a r(a) \frac{\partial}{\partial \sigma} \left( \frac{1}{\sqrt{2\pi\sigma^2}} \exp\left( -\frac{1}{2\sigma^2}(a-\mu)^2 \right) \right) da$$

$$= \sigma \int_a r(a) \left( -\frac{1}{\sqrt{2\pi\sigma^4}} + \frac{(a-\mu)^2}{\sqrt{2\pi\sigma^8}} \right) \exp\left( -\frac{1}{2\sigma^2}(a-\mu)^2 \right) da$$

$$= \int_a r(a) \frac{1}{\sqrt{2\pi\sigma^2}} \exp\left( -\frac{(a-\mu)^2}{2\sigma^2} \right) \left( \frac{(a-\mu)^2}{\sigma^2} - 1 \right) da$$

$$\overset{b=\frac{a-\mu}{\sigma}}{=} \int_b r(\sigma b + \mu) \frac{1}{\sqrt{2\pi}} \exp\left( -\frac{b^2}{2} \right) \left( \frac{b^2}{2} - 1 \right) db,$$

which is bounded:

$$\left| \sigma \frac{\partial r(\mu, \sigma)}{\partial \sigma} \right| = \left| \int_b r(\sigma b + \mu) \frac{1}{\sqrt{2\pi}} \exp\left( -\frac{b^2}{2} \right) \left( \frac{b^2}{2} - 1 \right) db \right|$$

$$\leq \int_b |r(\sigma b + \mu)| \frac{1}{\sqrt{2\pi}} \exp\left( -\frac{b^2}{2} \right) \left| \frac{b^2}{2} - 1 \right| db$$

$$\leq \int_b r_{\max} \frac{1}{\sqrt{2\pi}} \exp\left( -\frac{b^2}{2} \right) \left| \frac{b^2}{2} - 1 \right| db$$

$$\leq r_{\max} \int_b \frac{1}{\sqrt{2\pi}} \exp\left( -\frac{b^2}{2} \right) \left( \frac{b^2}{2} + 1 \right) db$$

$$\leq r_{\max} \mathbb{E}_{b \sim \mathcal{N}(0,1)} \left[ \frac{b^2}{2} + 1 \right]$$

$$= \frac{3}{2} r_{\max}.$$

Then for any $\alpha > \frac{3}{2} r_{\max}$, we have $\sigma \frac{\partial r(\mu,\sigma)}{\partial \sigma} \geq -\frac{3}{2} r_{\max} > -\alpha$. $\qquad \square$

**Lemma C.1.** *For arbitrary* $\boldsymbol{\theta}^w$ *and any* $\tilde{\boldsymbol{\theta}}^b$ *such that* $\nabla_{\boldsymbol{\theta}^b} J(\pi_{\tilde{\boldsymbol{\theta}}^b}^b) = 0$, *we have* $\nabla_{\boldsymbol{\theta}^m} J(\pi_{\tilde{\boldsymbol{\theta}}^m}^m) = 0$, *where* $\tilde{\boldsymbol{\theta}}^m = \left[ \tilde{\boldsymbol{\theta}}^{b\top}, \cdots, \tilde{\boldsymbol{\theta}}^{b\top}, \boldsymbol{\theta}^{w\top} \right]^\top$.

*Proof.* By assumption,

$$\nabla_{\boldsymbol{\theta}^b} J(\pi_{\tilde{\boldsymbol{\theta}}^b}^b) = \nabla_{\boldsymbol{\theta}^b} \mathbb{E}_{a \sim \pi_{\tilde{\boldsymbol{\theta}}^b}^b(a)} \left[ r(a) - \alpha \log \pi_{\tilde{\boldsymbol{\theta}}^b}^b(a) \right]$$

$$= \nabla_{\boldsymbol{\theta}^b} \int_a \pi_{\tilde{\boldsymbol{\theta}}^b}^b(a) \left( r(a) - \alpha \log \pi_{\tilde{\boldsymbol{\theta}}^b}^b(a) \right) da$$

$$= \int_a \left( r(a) - \alpha \log \pi_{\tilde{\boldsymbol{\theta}}^b}^b(a) - \alpha \right) \nabla_{\boldsymbol{\theta}^b} \pi_{\tilde{\boldsymbol{\theta}}^b}^b(a) \, da$$

$$= 0.$$

For any $\boldsymbol{\theta}^w$, to show $\nabla_{\boldsymbol{\theta}^m} J(\pi_{\tilde{\boldsymbol{\theta}}^m}^m) = 0$, we can show $\nabla_{\boldsymbol{\theta}_k^b} J(\pi_{\tilde{\boldsymbol{\theta}}^m}^m) = 0$ and $\nabla_{\boldsymbol{\theta}^w} J(\pi_{\tilde{\boldsymbol{\theta}}^m}^m) = 0$.

We first derive the gradient of $\pi_{\boldsymbol{\theta}^m}^m$ with respect to $\boldsymbol{\theta}_k^b$ and $\boldsymbol{\theta}^w$:

$$\nabla_{\boldsymbol{\theta}_k^b} \pi_{\boldsymbol{\theta}^m}^m(a) = \nabla_{\boldsymbol{\theta}_k^b} \sum_{k=1}^N \pi_{\boldsymbol{\theta}^w}^w(k) \pi_{\boldsymbol{\theta}_k^b}^b(a) = \pi_{\boldsymbol{\theta}^w}^w(k) \nabla_{\boldsymbol{\theta}_k^b} \pi_{\boldsymbol{\theta}_k^b}^b(a),$$

$$\nabla_{\boldsymbol{\theta}^w} \pi_{\boldsymbol{\theta}^m}^m(a) = \nabla_{\boldsymbol{\theta}^w} \sum_{k=1}^N \pi_{\boldsymbol{\theta}^w}^w(k) \pi_{\boldsymbol{\theta}_k^b}^b(a) = \sum_{k=1}^N \nabla_{\boldsymbol{\theta}^w} \pi_{\boldsymbol{\theta}^w}^w(k) \pi_{\boldsymbol{\theta}_k^b}^b(a).$$

Then, we can derive the gradient $J(\pi_{\boldsymbol{\theta}^m}^m)$ with respect to $\boldsymbol{\theta}_k^b$:

$$\nabla_{\boldsymbol{\theta}_k^b} J(\pi_{\boldsymbol{\theta}^m}^m) = \nabla_{\boldsymbol{\theta}_k^b} \mathbb{E}_{a \sim \pi_{\boldsymbol{\theta}^m}^m(a)} \left[ r(a) - \alpha \log \pi_{\boldsymbol{\theta}^m}^m(a) \right]$$

$$= \nabla_{\boldsymbol{\theta}_k^b} \int_a \left( r(a) - \alpha \pi_{\boldsymbol{\theta}^m}^m(a) \log \pi_{\boldsymbol{\theta}^m}^m(a) \right) da$$

$$= \int_a \left( r(a) - \alpha \log \pi_{\boldsymbol{\theta}^m}^m(a) - \alpha \right) \nabla_{\boldsymbol{\theta}_k^b} \pi_{\boldsymbol{\theta}^m}^m(a) \, da$$

$$= \int_a \left( r(a) - \alpha \log \pi_{\boldsymbol{\theta}^m}^m(a) - \alpha \right) \pi_{\boldsymbol{\theta}^w}^w(k) \nabla_{\boldsymbol{\theta}_k^b} \pi_{\boldsymbol{\theta}_k^b}^b(a) \, da$$

$$= \pi_{\boldsymbol{\theta}^w}^w(k) \int_a \left( r(a) - \alpha \log \pi_{\boldsymbol{\theta}^m}^m(a) - \alpha \right) \nabla_{\boldsymbol{\theta}_k^b} \pi_{\boldsymbol{\theta}_k^b}^b(a) \, da.$$

Plugging $\tilde{\boldsymbol{\theta}}^m = \left[ \tilde{\boldsymbol{\theta}}^{b\top}, \cdots, \tilde{\boldsymbol{\theta}}^{b\top}, \boldsymbol{\theta}^{w\top} \right]^\top$ and $\pi_{\tilde{\boldsymbol{\theta}}^m}^m(a) = \pi_{\tilde{\boldsymbol{\theta}}^b}^b(a)$ in, we have

$$\nabla_{\boldsymbol{\theta}_k^b} J(\pi_{\tilde{\boldsymbol{\theta}}^m}^m) = \pi_{\boldsymbol{\theta}^w}^w(k) \int_a \left( r(a) - \alpha \log \pi_{\tilde{\boldsymbol{\theta}}^b}^b(a) - \alpha \right) \nabla_{\boldsymbol{\theta}_k^b} \pi_{\tilde{\boldsymbol{\theta}}^b}^b(a) \, da = 0.$$

Next, we derive the gradient $J(\pi_{\boldsymbol{\theta}^m}^m)$ with respect to $\boldsymbol{\theta}^w$:

$$\nabla_{\boldsymbol{\theta}^w} J(\pi_{\boldsymbol{\theta}^m}^m) = \nabla_{\boldsymbol{\theta}^w} \mathbb{E}_{a \sim \pi_{\boldsymbol{\theta}^m}^m(a)} \left[ r(a) - \alpha \log \pi_{\boldsymbol{\theta}^m}^m(a) \right]$$

$$= \nabla_{\boldsymbol{\theta}^w} \int_a \left( r(a) - \alpha \pi_{\boldsymbol{\theta}^m}^m(a) \log \pi_{\boldsymbol{\theta}^m}^m(a) \right) da$$

$$= \int_a \left( r(a) - \alpha \log \pi_{\boldsymbol{\theta}^m}^m(a) - \alpha \right) \nabla_{\boldsymbol{\theta}^w} \pi_{\boldsymbol{\theta}^m}^m(a) \, da$$

$$= \int_a \left( r(a) - \alpha \log \pi_{\boldsymbol{\theta}^m}^m(a) - \alpha \right) \sum_{k=1}^N \nabla_{\boldsymbol{\theta}^w} \pi_{\boldsymbol{\theta}^w}^w(k) \pi_{\boldsymbol{\theta}_k^b}^b(a) \, da.$$

Again, plugging $\tilde{\boldsymbol{\theta}}^m = \left[ \tilde{\boldsymbol{\theta}}^{b\top}, \cdots, \tilde{\boldsymbol{\theta}}^{b\top}, \boldsymbol{\theta}^{w\top} \right]^\top$ and $\pi_{\tilde{\boldsymbol{\theta}}^m}^m(a) = \pi_{\tilde{\boldsymbol{\theta}}^b}^b(a)$ in, we have

$$\nabla_{\boldsymbol{\theta}^w} J(\pi_{\tilde{\boldsymbol{\theta}}^m}^m) = \int_a \left( r(a) - \alpha \log \pi_{\tilde{\boldsymbol{\theta}}^b}^b(a) - \alpha \right) \sum_{k=1}^N \nabla_{\boldsymbol{\theta}^w} \pi_{\boldsymbol{\theta}^w}^w(k) \pi_{\tilde{\boldsymbol{\theta}}^b}^b(a) \, da$$

$$= \int_a \left( r(a) - \alpha \log \pi_{\tilde{\boldsymbol{\theta}}^b}^b(a) - \alpha \right) \pi_{\tilde{\boldsymbol{\theta}}^b}^b(a) \, da \nabla_{\boldsymbol{\theta}^w} \sum_{k=1}^N \pi_{\boldsymbol{\theta}^w}^w(k)$$

$$= \int_a \left( r(a) - \alpha \log \pi_{\tilde{\boldsymbol{\theta}}^b}^b(a) - \alpha \right) \pi_{\tilde{\boldsymbol{\theta}}^b}^b(a) \, da \nabla_{\boldsymbol{\theta}^w} 1$$

$$= 0.$$

Thus, $\nabla_{\tilde{\boldsymbol{\theta}}} J(\pi_{\tilde{\boldsymbol{\theta}}^m}^m) = [\nabla_{\boldsymbol{\theta}_1^b} J(\pi_{\tilde{\boldsymbol{\theta}}^m}^m)^\top, \cdots, \nabla_{\boldsymbol{\theta}_N^b} J(\pi_{\tilde{\boldsymbol{\theta}}^m}^m)^\top, \nabla_{\boldsymbol{\theta}^w} J(\pi_{\tilde{\boldsymbol{\theta}}^m}^m)^\top]^\top = \boldsymbol{0}.$ $\qquad \square$

**Proposition 3.4** *The minimum $\alpha$ after which the mixture policy no longer has a stationary point is at least as large as that of the base policy, i.e., $\alpha_{\min}^{\pi^m} \geq \alpha_{\min}^{\pi^b}$, where $\alpha_{\min}^\pi = \inf\{\alpha \mid \nabla_{\boldsymbol{\theta}} J(\pi_{\boldsymbol{\theta}}) \neq \boldsymbol{0}, \forall \boldsymbol{\theta}\}$ for policy $\pi_{\boldsymbol{\theta}}$.*

*Proof.* This is a direct consequence of Lemma C.1. $\qquad \square$

### C.2 PROOFS FOR RESULTS IN SECTION 4

Define the (discounted) occupancy measure under $\pi_{\boldsymbol{\theta}}^m$ as $d_{\pi_{\boldsymbol{\theta}}^m}(s) \doteq \sum_{t=0}^\infty \mathbb{E}_{\pi_{\boldsymbol{\theta}}^m} [\gamma^t \mathbb{I}(S_t = s)]$. We first prove the half-reparameterization policy gradient theorem, which is a special case of Theorem 4.3 with $\alpha = 0$.

**Assumption 4.1.** $\mathcal{S}$ and $\mathcal{A}$ are compact.

**Assumption 4.2.** $p(s'|s, a)$, $d_0(s)$, $r(s, a)$ $f_{\boldsymbol{\theta}}(\epsilon; s, k)$, $f_{\boldsymbol{\theta}}^{-1}(a; s, k)$, $\pi_{\boldsymbol{\theta}}^w(k|s)$, $\pi_{\boldsymbol{\theta}}^b(a|s, k)$, $p(\epsilon)$, and their derivatives are continuous in variables $s$, $a$, $s'$, $\boldsymbol{\theta}$, and $\epsilon$.

**Theorem C.2** (Half-Reparameterization Policy Gradient Theorem). *Under Assumptions 4.1 and 4.2, we have*

$$\nabla_{\boldsymbol{\theta}} J_0(\pi_{\boldsymbol{\theta}}^m) = \mathbb{E}_{s \sim d_{\pi_{\boldsymbol{\theta}}^m}, k \sim \pi_{\boldsymbol{\theta}}^w(\cdot|s), \epsilon \sim p} \Big[ Q_{\pi_{\boldsymbol{\theta}}^m}(s, f_{\boldsymbol{\theta}}(\epsilon; s, k)) \nabla_{\boldsymbol{\theta}} \log \pi_{\boldsymbol{\theta}}^w(k|s)$$
$$+ \nabla_{\boldsymbol{\theta}} f_{\boldsymbol{\theta}}(\epsilon; s, k) \nabla_a Q_{\pi_{\boldsymbol{\theta}}^m}(s, a)|_{a=f_{\boldsymbol{\theta}}(\epsilon; s, k)} \Big].$$

*Proof.* We start with the policy gradient theorem (Sutton et al., 1999), which shows

$$\nabla_{\boldsymbol{\theta}} J_0(\pi_{\boldsymbol{\theta}}^m) = \int_{s,a} d_{\pi_{\boldsymbol{\theta}}^m}(s) \pi_{\boldsymbol{\theta}}^m(a|s) Q_{\pi_{\boldsymbol{\theta}}^m}(s, a) \nabla_{\boldsymbol{\theta}} \log \pi_{\boldsymbol{\theta}}^m(a|s) \, da \, ds.$$

Then

$$\nabla_{\boldsymbol{\theta}} J_0(\pi_{\boldsymbol{\theta}}^m) = \int_{s,a} d_{\pi_{\boldsymbol{\theta}}^m}(s) \pi_{\boldsymbol{\theta}}^m(a|s) Q_{\pi_{\boldsymbol{\theta}}^m}(s, a) \nabla_{\boldsymbol{\theta}} \log \pi_{\boldsymbol{\theta}}^m(a|s) \, da \, ds$$

$$= \int_s d_{\pi_{\boldsymbol{\theta}}^m}(s) \left( \int_a Q_{\pi_{\boldsymbol{\theta}}^m}(s, a) \nabla_{\boldsymbol{\theta}} \pi_{\boldsymbol{\theta}}^m(a|s) \, da \right) ds \qquad (15)$$

$$= \int_s d_{\pi_{\boldsymbol{\theta}}^m}(s) \left( \int_a Q_{\pi_{\boldsymbol{\theta}}^m}(s, a) \nabla_{\boldsymbol{\theta}} \left( \sum_{k=1}^N \pi_{\boldsymbol{\theta}}^w(k|s) \pi_{\boldsymbol{\theta}}^b(a|s, k) \right) da \right) ds$$

$$= \int_s d_{\pi_{\boldsymbol{\theta}}^m}(s) \sum_{k=1}^N \left( \int_a Q_{\pi_{\boldsymbol{\theta}}^m}(s, a) \nabla_{\boldsymbol{\theta}} \left( \pi_{\boldsymbol{\theta}}^w(k|s) \pi_{\boldsymbol{\theta}}^b(a|s, k) \right) da \right) ds$$

$$= \int_s d_{\pi_{\boldsymbol{\theta}}^m}(s) \sum_{k=1}^N \left( \int_a Q_{\pi_{\boldsymbol{\theta}}^m}(s, a) \nabla_{\boldsymbol{\theta}} \pi_{\boldsymbol{\theta}}^w(k|s) \pi_{\boldsymbol{\theta}}^b(a|s, k) \, da \right.$$

$$\left. + \int_a Q_{\pi_{\boldsymbol{\theta}}^m}(s, a) \pi_{\boldsymbol{\theta}}^w(k|s) \nabla_{\boldsymbol{\theta}} \pi_{\boldsymbol{\theta}}^b(a|s, k) \, da \right) ds$$

$$= \int_s d_{\pi_{\boldsymbol{\theta}}^m}(s) \sum_{k=1}^N \left( \int_a Q_{\pi_{\boldsymbol{\theta}}^m}(s, a) \nabla_{\boldsymbol{\theta}} \log \pi_{\boldsymbol{\theta}}^w(k|s) \pi_{\boldsymbol{\theta}}^w(k|s) \pi_{\boldsymbol{\theta}}^b(a|s, k) \, da \right.$$

$$\left. + \pi_{\boldsymbol{\theta}}^w(k|s) \left( \int_a \nabla_{\boldsymbol{\theta}} \left( Q_{\pi_{\boldsymbol{\theta}}^m}(s, a) \pi_{\boldsymbol{\theta}}^b(a|s, k) \right) da - \int_a \pi_{\boldsymbol{\theta}}^b(a|s, k) \nabla_{\boldsymbol{\theta}} Q_{\pi_{\boldsymbol{\theta}}^m}(s, a) \, da \right) \right) ds$$

$$= \int_s d_{\pi_{\boldsymbol{\theta}}^m}(s) \sum_{k=1}^N \pi_{\boldsymbol{\theta}}^w(k|s) \left( \int_a Q_{\pi_{\boldsymbol{\theta}}^m}(s, a) \nabla_{\boldsymbol{\theta}} \log \pi_{\boldsymbol{\theta}}^w(k|s) \pi_{\boldsymbol{\theta}}^b(a|s, k) \, da \right.$$

$$\left. + \nabla_{\boldsymbol{\theta}} \int_a Q_{\pi_{\boldsymbol{\theta}}^m}(s, a) \pi_{\boldsymbol{\theta}}^b(a|s, k) \, da - \int_a \pi_{\boldsymbol{\theta}}^b(a|s, k) \nabla_{\boldsymbol{\theta}} Q_{\pi_{\boldsymbol{\theta}}^m}(s, a) \, da \right) ds$$

$$\overset{a=f_{\boldsymbol{\theta}}(\epsilon; s, k)}{=} \int_s d_{\pi_{\boldsymbol{\theta}}^m}(s) \sum_{k=1}^N \pi_{\boldsymbol{\theta}}^w(k|s) \left( \int_\epsilon p(\epsilon) Q_{\pi_{\boldsymbol{\theta}}^m}(s, f_{\boldsymbol{\theta}}(\epsilon; s, k)) \nabla_{\boldsymbol{\theta}} \log \pi_{\boldsymbol{\theta}}^w(k|s) \, d\epsilon \right.$$

$$\left. + \nabla_{\boldsymbol{\theta}} \int_\epsilon p(\epsilon) Q_{\pi_{\boldsymbol{\theta}}^m}(s, f_{\boldsymbol{\theta}}(\epsilon; s, k)) \, d\epsilon - \int_\epsilon p(\epsilon) \nabla_{\boldsymbol{\theta}} Q_{\pi_{\boldsymbol{\theta}}^m}(s, a)|_{a=f_{\boldsymbol{\theta}}(\epsilon; s, k)} \, d\epsilon \right) ds$$

$$= \int_s d_{\pi_{\boldsymbol{\theta}}^m}(s) \sum_{k=1}^N \pi_{\boldsymbol{\theta}}^w(k|s) \left( \int_\epsilon p(\epsilon) Q_{\pi_{\boldsymbol{\theta}}^m}(s, f_{\boldsymbol{\theta}}(\epsilon; s, k)) \nabla_{\boldsymbol{\theta}} \log \pi_{\boldsymbol{\theta}}^w(k|s) \, d\epsilon \right.$$

$$\left. + \int_\epsilon p(\epsilon) \nabla_{\boldsymbol{\theta}} f_{\boldsymbol{\theta}}(\epsilon; s, k) \nabla_a Q_{\pi_{\boldsymbol{\theta}}^m}(s, a)|_{a=f_{\boldsymbol{\theta}}(\epsilon; s, k)} \, d\epsilon \right) ds$$

$$= \mathbb{E}_{s \sim d_{\pi_{\boldsymbol{\theta}}^m}, k \sim \pi_{\boldsymbol{\theta}}^w(\cdot|s), \epsilon \sim p} \Big[ Q_{\pi_{\boldsymbol{\theta}}^m}(s, f_{\boldsymbol{\theta}}(\epsilon; s, k)) \nabla_{\boldsymbol{\theta}} \log \pi_{\boldsymbol{\theta}}^w(k|s)$$

$$+ \nabla_{\boldsymbol{\theta}} f_{\boldsymbol{\theta}}(\epsilon; s, k) \nabla_a Q_{\pi_{\boldsymbol{\theta}}^m}(s, a)|_{a=f_{\boldsymbol{\theta}}(\epsilon; s, k)} \Big],$$

where the second last equality is due to

$$\nabla_{\boldsymbol{\theta}} Q_{\pi_{\boldsymbol{\theta}}^m}(s, f_{\boldsymbol{\theta}}(\epsilon; s, k)) - \nabla_{\boldsymbol{\theta}} Q_{\pi_{\boldsymbol{\theta}}^m}(s, a)|_{a=f_{\boldsymbol{\theta}}(\epsilon; s, k)}$$

$$=\nabla_{\boldsymbol{\theta}} f_{\boldsymbol{\theta}}(\epsilon; s, k) \nabla_a Q_{\pi_{\boldsymbol{\theta}}^m}(s, a)|_{a=f_{\boldsymbol{\theta}}(\epsilon; s, k)} + \nabla_{\boldsymbol{\theta}} Q_{\pi_{\boldsymbol{\theta}}^m}(s, a)|_{a=f_{\boldsymbol{\theta}}(\epsilon; s, k)} - \nabla_{\boldsymbol{\theta}} Q_{\pi_{\boldsymbol{\theta}}^m}(s, a)|_{a=f_{\boldsymbol{\theta}}(\epsilon; s, k)}$$

$$=\nabla_{\boldsymbol{\theta}} f_{\boldsymbol{\theta}}(\epsilon; s, k) \nabla_a Q_{\pi_{\boldsymbol{\theta}}^m}(s, a)|_{a=f_{\boldsymbol{\theta}}(\epsilon; s, k)}.$$

$$\square$$

*Remark* C.3. The key contribution of this proof is the decoupling of the gradient of the weighting policy $\pi_{\boldsymbol{\theta}}^w$ and the gradient of the component policies $\pi_{\boldsymbol{\theta}}^b$. The former, $\nabla_{\boldsymbol{\theta}} \pi_{\boldsymbol{\theta}}^w$, is converted back to the likelihood-ratio gradient, while the latter, $\nabla_{\boldsymbol{\theta}} \pi_{\boldsymbol{\theta}}^b$, is handled in the same way as in the proof the reparameterization policy gradient theorem (Lan et al., 2022).

Combining the insight from Remark C.3 with the proof of the entropy-regularized reparameterization policy gradient theorem in Lan et al. (2022), we can obtain Theorem 4.3.

**Theorem 4.3** (Entropy-Regularized Half-Reparameterization Policy Gradient Theorem). *Under Assumptions 4.1 and 4.2, we have*

$$\nabla_{\boldsymbol{\theta}} J(\pi_{\boldsymbol{\theta}}^m) = \mathbb{E}_{s \sim d_{\pi_{\boldsymbol{\theta}}^m}, k \sim \pi_{\boldsymbol{\theta}}^w(\cdot|s), \epsilon \sim p} \Big[ \nabla_{\boldsymbol{\theta}} \log \pi_{\boldsymbol{\theta}}^w(k|s) \big( Q_{\pi_{\boldsymbol{\theta}}^m}(s, f_{\boldsymbol{\theta}}(\epsilon; s, k)) - \alpha \log \pi_{\boldsymbol{\theta}}^m(f_{\boldsymbol{\theta}}(\epsilon; s, k)|s) \big)$$

$$+ \nabla_{\boldsymbol{\theta}} f_{\boldsymbol{\theta}}(\epsilon; s, k) \nabla_a \big( Q_{\pi_{\boldsymbol{\theta}}^m}(s, a) - \alpha \log \pi_{\boldsymbol{\theta}}^m(a|s) \big)|_{a=f_{\boldsymbol{\theta}}(\epsilon; s, k)} \Big].$$

*Proof.* From (3) of Ahmed et al. (2019), we have the entropy-regularized policy gradient for the regularized objective:

$$\nabla_{\boldsymbol{\theta}} J(\pi_{\boldsymbol{\theta}}^m) = \int_{s,a} d_{\pi_{\boldsymbol{\theta}}^m}(s) \pi_{\boldsymbol{\theta}}^m(a|s) \big( Q_{\pi_{\boldsymbol{\theta}}^m}(s, a) \nabla_{\boldsymbol{\theta}} \log \pi_{\boldsymbol{\theta}}^m(a|s) + \alpha \nabla_{\boldsymbol{\theta}} \mathcal{H}(\pi_{\boldsymbol{\theta}}^m(\cdot|s)) \big) \, da \, ds.$$

The first term, $Q_{\pi_{\boldsymbol{\theta}}^m}(s, a) \nabla_{\boldsymbol{\theta}} \log \pi_{\boldsymbol{\theta}}^m(a|s)$, can be directly handled by Theorem C.2. Here, we analyze the second term, $\alpha \nabla_{\boldsymbol{\theta}} \mathcal{H}(\pi_{\boldsymbol{\theta}}^m(\cdot|s))$. Notice that

$$\nabla_{\boldsymbol{\theta}} \mathcal{H}(\pi_{\boldsymbol{\theta}}^m(\cdot|s)) = -\nabla_{\boldsymbol{\theta}} \int_a \pi_{\boldsymbol{\theta}}^m(a|s) \log \pi_{\boldsymbol{\theta}}^m(a|s) \, da$$

$$= -\int_a \big( \nabla_{\boldsymbol{\theta}} \pi_{\boldsymbol{\theta}}^m(a|s) \log \pi_{\boldsymbol{\theta}}^m(a|s) + \pi_{\boldsymbol{\theta}}^m(a|s) \nabla_{\boldsymbol{\theta}} \log \pi_{\boldsymbol{\theta}}^m(a|s) \big) \, da$$

$$= -\int_a \big( \nabla_{\boldsymbol{\theta}} \pi_{\boldsymbol{\theta}}^m(a|s) \log \pi_{\boldsymbol{\theta}}^m(a|s) + \nabla_{\boldsymbol{\theta}} \pi_{\boldsymbol{\theta}}^m(a|s) \big) \, da$$

$$\overset{\int_a \nabla_{\boldsymbol{\theta}} \pi_{\boldsymbol{\theta}}^m(a|s) \, da = 0}{=} -\int_a \nabla_{\boldsymbol{\theta}} \pi_{\boldsymbol{\theta}}^m(a|s) \log \pi_{\boldsymbol{\theta}}^m(a|s) \, da,$$

then we have

$$\int_{s,a} d_{\pi_{\boldsymbol{\theta}}^m}(s) \pi_{\boldsymbol{\theta}}^m(a|s) \alpha \nabla_{\boldsymbol{\theta}} \mathcal{H}(\pi_{\boldsymbol{\theta}}^m(\cdot|s)) \, da \, ds$$

$$= \alpha \int_s d_{\pi_{\boldsymbol{\theta}}^m}(s) \nabla_{\boldsymbol{\theta}} \mathcal{H}(\pi_{\boldsymbol{\theta}}^m(\cdot|s)) \, ds$$

$$= -\alpha \int_s d_{\pi_{\boldsymbol{\theta}}^m}(s) \int_a \nabla_{\boldsymbol{\theta}} \pi_{\boldsymbol{\theta}}^m(a|s) \log \pi_{\boldsymbol{\theta}}^m(a|s) \, da \, ds. \tag{16}$$

Since equation 16 resembles equation 15, by following the same steps in the proof of Theorem C.2, we can obtain

$$\int_{s,a} d_{\pi_{\boldsymbol{\theta}}^m}(s) \pi_{\boldsymbol{\theta}}^m(a|s) \alpha \nabla_{\boldsymbol{\theta}} \mathcal{H}(\pi_{\boldsymbol{\theta}}^m(\cdot|s)) \, da \, ds$$

$$= \mathbb{E}_{s \sim d_{\pi_{\boldsymbol{\theta}}^m}, k \sim \pi_{\boldsymbol{\theta}}^w(\cdot|s), \epsilon \sim p} \Big[ -\alpha \log \pi_{\boldsymbol{\theta}}^m(f_{\boldsymbol{\theta}}(\epsilon; s, k)|s) \nabla_{\boldsymbol{\theta}} \log \pi_{\boldsymbol{\theta}}^w(k|s)$$

$$- \alpha \nabla_{\boldsymbol{\theta}} f_{\boldsymbol{\theta}}(\epsilon; s, k) \nabla_a \log \pi_{\boldsymbol{\theta}}^m(a|s)|_{a=f_{\boldsymbol{\theta}}(\epsilon; s, k)} \Big],$$

Combining the above gradient term with the gradient term from Theorem C.2 concludes the proof.

$$\square$$

By using the same technique, we can obtain the half-reparameterization gradient of SAC's objective in equation 3.

**Assumption C.4.** $Q_{\mathbf{w}}(s, a)$ and its derivatives are continuous in variables $s$ and $a$.

**Proposition C.5.** *Under Assumptions 4.1, 4.2, and C.4, we have*

$$\nabla_{\boldsymbol{\theta}} \hat{J}(\pi_{\boldsymbol{\theta}}^w) = \mathbb{E}_{s \sim \mathcal{B}, k \sim \pi_{\boldsymbol{\theta}}^w(\cdot|s), \epsilon \sim p} \Big[ \nabla_{\boldsymbol{\theta}} \log \pi_{\boldsymbol{\theta}}^w(k|s) \big( Q_{\mathbf{w}}(s, f_{\boldsymbol{\theta}}(\epsilon; s, k)) - \alpha \log \pi_{\boldsymbol{\theta}}^m(f_{\boldsymbol{\theta}}(\epsilon; s, k)|s) \big)$$
$$+ \nabla_{\boldsymbol{\theta}} \big( Q_{\mathbf{w}}(s, f_{\boldsymbol{\theta}}(\epsilon; s, k)) - \alpha \log \pi_{\boldsymbol{\theta}}^m(f_{\boldsymbol{\theta}}(\epsilon; s, k)|s) \big) \Big].$$

*Proof.* We first rewrite (3) with reparameterized component policies:

$$\hat{J}(\pi_{\boldsymbol{\theta}}^w) = \mathbb{E}_{S_t \sim \mathcal{B}, A_t \sim \pi_{\boldsymbol{\theta}}^m} [Q_{\mathbf{w}}(S_t, A_t) - \alpha \log \pi_{\boldsymbol{\theta}}^m(A_t|S_t)]$$

$$= \int_s d_{\mathcal{B}}(s) \int_a \pi_{\boldsymbol{\theta}}^m(a|s) \big( Q_{\mathbf{w}}(s, a) - \alpha \log \pi_{\boldsymbol{\theta}}^m(a|s) \big) \, da \, ds$$

$$= \int_s d_{\mathcal{B}}(s) \int_a \sum_{k=1}^N \pi_{\boldsymbol{\theta}}^w(k|s) \pi_{\boldsymbol{\theta}}^b(a|s, k) \big( Q_{\mathbf{w}}(s, a) - \alpha \log \pi_{\boldsymbol{\theta}}^m(a|s) \big) \, da \, ds$$

$$\stackrel{a = f_{\boldsymbol{\theta}}(\epsilon; s, k)}{=} \int_s d_{\mathcal{B}}(s) \int_\epsilon \sum_{k=1}^N \pi_{\boldsymbol{\theta}}^w(k|s) p(\epsilon) \big( Q_{\mathbf{w}}(s, f_{\boldsymbol{\theta}}(\epsilon; s, k)) - \alpha \log \pi_{\boldsymbol{\theta}}^m(f_{\boldsymbol{\theta}}(\epsilon; s, k)|s) \big) \, d\epsilon \, ds$$

$$= \int_s d_{\mathcal{B}}(s) \int_\epsilon p(\epsilon) \sum_{k=1}^N \pi_{\boldsymbol{\theta}}^w(k|s) \big( Q_{\mathbf{w}}(s, f_{\boldsymbol{\theta}}(\epsilon; s, k)) - \alpha \log \pi_{\boldsymbol{\theta}}^m(f_{\boldsymbol{\theta}}(\epsilon; s, k)|s) \big) \, d\epsilon \, ds.$$

We can then derive its gradient:

$$\nabla_{\boldsymbol{\theta}} \hat{J}(\pi_{\boldsymbol{\theta}}^m)$$

$$= \nabla_{\boldsymbol{\theta}} \int_s d_{\mathcal{B}}(s) \int_\epsilon p(\epsilon) \sum_{k=1}^N \pi_{\boldsymbol{\theta}}^w(k|s) \big( Q_{\mathbf{w}}(s, f_{\boldsymbol{\theta}}(\epsilon; s, k)) - \alpha \log \pi_{\boldsymbol{\theta}}^m(f_{\boldsymbol{\theta}}(\epsilon; s, k)|s) \big) \, d\epsilon \, ds$$

$$= \int_s d_{\mathcal{B}}(s) \int_\epsilon p(\epsilon) \sum_{k=1}^N \nabla_{\boldsymbol{\theta}} \Big( \pi_{\boldsymbol{\theta}}^w(k|s) \big( Q_{\mathbf{w}}(s, f_{\boldsymbol{\theta}}(\epsilon; s, k)) - \alpha \log \pi_{\boldsymbol{\theta}}^m(f_{\boldsymbol{\theta}}(\epsilon; s, k)|s) \big) \Big) \, d\epsilon \, ds$$

$$= \int_s d_{\mathcal{B}}(s) \int_\epsilon p(\epsilon) \sum_{k=1}^N \Big( \nabla_{\boldsymbol{\theta}} \pi_{\boldsymbol{\theta}}^w(k|s) \big( Q_{\mathbf{w}}(s, f_{\boldsymbol{\theta}}(\epsilon; s, k)) - \alpha \log \pi_{\boldsymbol{\theta}}^m(f_{\boldsymbol{\theta}}(\epsilon; s, k)|s) \big)$$
$$+ \pi_{\boldsymbol{\theta}}^w(k|s) \nabla_{\boldsymbol{\theta}} \big( Q_{\mathbf{w}}(s, f_{\boldsymbol{\theta}}(\epsilon; s, k)) - \alpha \log \pi_{\boldsymbol{\theta}}^m(f_{\boldsymbol{\theta}}(\epsilon; s, k)|s) \big) \Big) \, d\epsilon \, ds$$

$$= \int_s d_{\mathcal{B}}(s) \int_\epsilon p(\epsilon) \sum_{k=1}^N \Big( \pi_{\boldsymbol{\theta}}^w(k|s) \nabla_{\boldsymbol{\theta}} \log \pi_{\boldsymbol{\theta}}^w(k|s) \big( Q_{\mathbf{w}}(s, f_{\boldsymbol{\theta}}(\epsilon; s, k)) - \alpha \log \pi_{\boldsymbol{\theta}}^m(f_{\boldsymbol{\theta}}(\epsilon; s, k)|s) \big)$$
$$+ \pi_{\boldsymbol{\theta}}^w(k|s) \nabla_{\boldsymbol{\theta}} \big( Q_{\mathbf{w}}(s, f_{\boldsymbol{\theta}}(\epsilon; s, k)) - \alpha \log \pi_{\boldsymbol{\theta}}^m(f_{\boldsymbol{\theta}}(\epsilon; s, k)|s) \big) \Big) \, d\epsilon \, ds$$

$$= \int_s d_{\mathcal{B}}(s) \int_\epsilon p(\epsilon) \sum_{k=1}^N \pi_{\boldsymbol{\theta}}^w(k|s) \Big( \nabla_{\boldsymbol{\theta}} \log \pi_{\boldsymbol{\theta}}^w(k|s) \big( Q_{\mathbf{w}}(s, f_{\boldsymbol{\theta}}(\epsilon; s, k)) - \alpha \log \pi_{\boldsymbol{\theta}}^m(f_{\boldsymbol{\theta}}(\epsilon; s, k)|s) \big)$$
$$+ \nabla_{\boldsymbol{\theta}} \big( Q_{\mathbf{w}}(s, f_{\boldsymbol{\theta}}(\epsilon; s, k)) - \alpha \log \pi_{\boldsymbol{\theta}}^m(f_{\boldsymbol{\theta}}(\epsilon; s, k)|s) \big) \Big) \, d\epsilon \, ds$$

$$= \mathbb{E}_{s \sim \mathcal{B}, k \sim \pi_{\boldsymbol{\theta}}^w(\cdot|s), \epsilon \sim p} \Big[ \nabla_{\boldsymbol{\theta}} \log \pi_{\boldsymbol{\theta}}^w(k|s) \big( Q_{\mathbf{w}}(s, f_{\boldsymbol{\theta}}(\epsilon; s, k)) - \alpha \log \pi_{\boldsymbol{\theta}}^m(f_{\boldsymbol{\theta}}(\epsilon; s, k)|s) \big)$$
$$+ \nabla_{\boldsymbol{\theta}} \big( Q_{\mathbf{w}}(s, f_{\boldsymbol{\theta}}(\epsilon; s, k)) - \alpha \log \pi_{\boldsymbol{\theta}}^m(f_{\boldsymbol{\theta}}(\epsilon; s, k)|s) \big) \Big].$$

$\square$

### C.3 VARIANCE REDUCTION PROPERTIES OF THE MRP ESTIMATOR

In this section, we prove that the MRP estimator has a lower variance than the LR estimator under some smoothness conditions. This variance reduction property of the MRP estimator parallels that

of the RP estimator (Gal, 2016; Xu et al., 2019). Specifically, we show that the marginal variance of MRP can be expressed or controlled by the variance of the corresponding RP estimator for the individual components. Through this reduction, we can focus on analyzing the variance of the individual components, which is much easier to work with.

While the following results can be easily extend to the full reinforcement learning setting with regularization, we consider the bandit setting without regularization for the ease of presentation:

$$J(\pi_{\boldsymbol{\theta}}) = \mathbb{E}_{a \sim \pi_{\boldsymbol{\theta}}}[r(a)]. \tag{17}$$

The entropy term can also be included by redefining $r(a)$ to be the sum of the reward and sample entropy. We focus on analyzing the gradient of different estimators with respect to the distribution parameters and the weighting probabilities of Gaussian mixture policies. In this case, the mixture policy can be expressed as $\pi_{\boldsymbol{\theta}}^m(a) = \sum_{k=1}^N \pi_{\boldsymbol{\theta}^w}^w(k)\pi_{\boldsymbol{\theta}^{b_k}}^b(a) = \sum_{k=1}^N w_k \mathcal{N}(a; \mu_k, \sigma_k^2)$ with $\boldsymbol{\theta} = \left[\boldsymbol{\theta}^{b_1 \top}, \cdots, \boldsymbol{\theta}^{b_N \top}, \boldsymbol{\theta}^{w \top}\right]^\top = \left[[\mu_1, \sigma_1]^\top, \cdots, [\mu_N, \sigma_N]^\top, [w_1, \cdots, w_N]^\top\right]^\top \in \mathbb{R}^{3N}$, where $\mu_k$, $\sigma_k$, and $w_k$ may depend on shared parameters implicitly. The corresponding estimators then are

$$\text{LR:} \quad \hat{\nabla}_{\boldsymbol{\theta}}^{\text{LR}} J(\pi_{\boldsymbol{\theta}}^m) = \nabla_{\boldsymbol{\theta}} \log \pi_{\boldsymbol{\theta}}^m(A) r(A), \tag{18}$$

$$\text{MRP:} \quad \hat{\nabla}_{\boldsymbol{\theta}}^{\text{MRP}} J(\pi_{\boldsymbol{\theta}}^m) = \nabla_{\boldsymbol{\theta}} \sum_{k=1}^N \pi_{\boldsymbol{\theta}^w}^w(k) r\left(f_{\boldsymbol{\theta}^{b_k}}(\epsilon)\right), \tag{19}$$

where $A \sim \pi_{\boldsymbol{\theta}}^m(\cdot)$, $\epsilon \sim \mathcal{N}(0, 1)$, $\pi_{\boldsymbol{\theta}^w}^w(k) = w_k$, and $f_{\boldsymbol{\theta}^{b_k}}(\epsilon) = \mu_k + \epsilon \sigma_k$.

Before we present the variance comparison between different estimators, we first analyze the relationship between the variance for the mixture policy and its individual components. Assuming only one of the component $\pi_{\boldsymbol{\theta}^{b_k}}^b(a) = \mathcal{N}(a; \mu_k, \sigma_k^2)$ is used, the corresponding estimators are

$$\text{LR:} \quad \hat{\nabla}_{\boldsymbol{\theta}^{b_k}}^{\text{LR}} J(\pi_{\boldsymbol{\theta}^{b_k}}^b) = \nabla_{\boldsymbol{\theta}^{b_k}} \log \pi_{\boldsymbol{\theta}^{b_k}}^b(A) r(A), \tag{20}$$

$$\text{RP:} \quad \hat{\nabla}_{\boldsymbol{\theta}^{b_k}}^{\text{RP}} J(\pi_{\boldsymbol{\theta}^{b_k}}^b) = \nabla_{\boldsymbol{\theta}^{b_k}} r\left(f_{\boldsymbol{\theta}^{b_k}}(\epsilon)\right). \tag{21}$$

Define $\pi^m \doteq \pi_{\boldsymbol{\theta}}^m$, $\pi^{b_k} \doteq \pi_{\boldsymbol{\theta}^{b_k}}^b$, $\hat{\partial}_{\theta_i^{b_k}} \cdot \doteq \left[\hat{\nabla}_{\boldsymbol{\theta}^{b_k}} \cdot\right]_i$, $\hat{\partial}_{w_k} \cdot \doteq \left[\hat{\nabla}_{\boldsymbol{\theta}^w} \cdot\right]_k$ and $\rho_m^{b_k}(A) = \frac{\pi^{b_k}(A)}{\pi^m(A)}$, where $\theta_i^{b_k}$ and $w_k$ are the $i$-th distribution parameter and the weight of the $k$-th component policy, respectively. For Gaussian component policies, $\theta_i^{b_k} \in \{\mu_k, \sigma_k\}$. We have the following relationships.

**Lemma C.6.** *The marginal variances of the LR estimator for mixture policies satisfy*

$$\mathbb{V}_{\pi^m(A)} \left(\hat{\partial}_{\theta_i^{b_k}}^{LR} J(\pi_{\boldsymbol{\theta}}^m)\right) = w_k^2 \mathbb{V}_{\pi^m(A)} \left(\rho_m^{b_k}(A) \hat{\partial}_{\theta_i^{b_k}}^{LR} J(\pi_{\boldsymbol{\theta}^{b_k}}^b)\right),$$

$$\mathbb{V}_{\pi^m(A)} \left(\hat{\partial}_{w_k}^{LR} J(\pi_{\boldsymbol{\theta}}^m)\right) = \mathbb{V}_{\pi^m(A)} \left(\rho_m^{b_k}(A) r(A)\right).$$

**Lemma C.7.** *The marginal variances of the MRP estimator for mixture policies satisfy*

$$\mathbb{V}_{\mathcal{N}(\epsilon; 0, 1)} \left(\hat{\partial}_{\theta_i^{b_k}}^{MRP} J(\pi_{\boldsymbol{\theta}}^m)\right) = w_k^2 \mathbb{V}_{\mathcal{N}(\epsilon; 0, 1)} \left(\hat{\partial}_{\theta_i^{b_k}}^{RP} J(\pi_{\boldsymbol{\theta}^{b_k}}^b)\right),$$

$$\mathbb{V}_{\mathcal{N}(\epsilon; 0, 1)} \left(\hat{\partial}_{w_k}^{MRP} J(\pi_{\boldsymbol{\theta}}^m)\right) = \mathbb{V}_{\pi^{b_k}(A)} \left(r(A)\right).$$

Similar to Gal (2016), we assume the reward function $r$ satisfies certain smoothness conditions (see Assumption C.8). Assumption C.8 gives the condition of the reward function under which the MRP estimator has lower variance. Heuristically, it requires the reward function to be smooth relative to the noise scale. For example, $\sin(x)$ satisfies the condition for $\mathcal{N}(0, 1)$ while $\sin(10x)$ does not (see Table 3.3 in Gal for a numerical experiment).

**Assumption C.8** (Expanded form of Assumption 4.5). $r : \mathbb{R} \to \mathbb{R}$ is twice differentiable with first and second derivatives $r'$ and $r''$. For all $k \in \{1, \cdots, N\}$ and $g(a) \in \{(a - \mu_k)r(a), r'(a), ((a - \mu_k)^2 - \sigma_k^2)r(a), (a - \mu_k)r'(a)\}$, $g$ has finite variance and its first derivative $g'$ is absolutely integrable under $\pi^{b_k}$: $\mathbb{V}_{\pi^{b_k}(A)}(g(A)) < \infty$ and $\mathbb{E}_{\pi^{b_k}(A)}[|g'(A)|] < \infty$, with $\pi^{b_k}(a) = \mathcal{N}(a; \mu_k, \sigma_k)$. Further,

it holds that

$$\sum_{k=1}^{N} \left( \mathbb{E}_{\pi^{b_k}(A)} \left[ (A - \mu_k)r'(A) + r(A) \right]^2 - \sigma_k^4 \mathbb{E}_{\pi^{b_k}(A)} \left[ r''(A)^2 \right] \right) \geq 0,$$

$$\sum_{k=1}^{N} \left( \mathbb{E}_{\pi^{b_k}(A)} \left[ ((A - \mu_k)^2 - \sigma_k^2)r'(A) + 2(A - \mu_k)r(A) \right]^2 \right.$$
$$\left. - \sigma_k^4 \mathbb{E}_{\pi^{b_k}(A)} \left[ ((A - \mu_k)r''(A) + r'(A))^2 \right] \right) \geq 0.$$

**Assumption C.9** (Restatement of Assumption 4.6). The sum of the variance of the importance-sampling LR estimator over all components is larger than that of the on-policy LR estimator:

$$\sum_{k=1}^{N} \left( \mathbb{V}_{\pi^m(A)} \left( \rho_m^{b_k}(A) \hat{\partial}_{\theta_i^{b_k}}^{\mathrm{LR}} J(\pi_{\boldsymbol{\theta}^{b_k}}^b) \right) - \mathbb{V}_{\pi^{b_k}(A)} \left( \hat{\partial}_{\theta_i^{b_k}}^{\mathrm{LR}} J(\pi_{\boldsymbol{\theta}^{b_k}}^b) \right) \right) \geq 0 \quad \text{for} \quad \theta_i^{b_k} \in \{\mu_k, \sigma_k\},$$

$$\sum_{k=1}^{N} \left( \mathbb{V}_{\pi^m(A)} \left( \rho_m^{b_k}(A) r(A) \right) - \mathbb{V}_{\pi^{b_k}(A)} \left( r(A) \right) \right) \geq 0.$$

**Proposition C.10** (Restatement of Proposition 4.7). *Under Assumptions C.8 and C.9, the trace of the covariance matrix of the MRP estimator in Equation (19) is smaller than that of the LR estimator in Equation (18):*

$$\mathrm{Tr} \left( \mathbb{C}_{\mathcal{N}(\epsilon;0,1)} \left( \hat{\nabla}_{\boldsymbol{\theta}}^{MRP} J(\pi_{\boldsymbol{\theta}}^m) \right) \right) \leq \mathrm{Tr} \left( \mathbb{C}_{\pi^m(A)} \left( \hat{\nabla}_{\boldsymbol{\theta}}^{LR} J(\pi_{\boldsymbol{\theta}}^m) \right) \right).$$

### C.3.1 PROOFS

*Proof of Lemma C.6.*

For any fixed component index $k$ and any of its distribution parameters $\theta_i^{b_k} \in \{\mu_k, \sigma_k\}$, we begin with inspecting the corresponding partial derivative

$$\hat{\partial}_{\theta_i^{b_k}}^{\mathrm{LR}} J(\pi_{\boldsymbol{\theta}}^m) \;=\; r(A) \, \partial_{\theta_i^{b_k}} \log \pi_{\boldsymbol{\theta}}^m(A).$$

Because only the $k$-th component depends on $\theta_i^{b_k}$,

$$\partial_{\theta_i^{b_k}} \pi_{\boldsymbol{\theta}}^m(A) = w_k \, \partial_{\theta_i^{b_k}} \pi_{\boldsymbol{\theta}^{b_k}}^b(A),$$

$$\partial_{\theta_i^{b_k}} \log \pi_{\boldsymbol{\theta}}^m(A) = w_k \, \frac{\pi_{\boldsymbol{\theta}^{b_k}}^b(A)}{\pi_{\boldsymbol{\theta}}^m(A)} \, \partial_{\theta_i^{b_k}} \log \pi_{\boldsymbol{\theta}^{b_k}}^b(A)$$

$$= w_k \, \rho_m^{b_k}(A) \, \partial_{\theta_i^{b_k}} \log \pi_{\boldsymbol{\theta}^{b_k}}^b(A).$$

Hence

$$\hat{\partial}_{\theta_i^{b_k}}^{\mathrm{LR}} J(\pi_{\boldsymbol{\theta}}^m) = w_k \, \rho_m^{b_k}(A) \, \hat{\partial}_{\theta_i^{b_k}}^{\mathrm{LR}} J(\pi_{\boldsymbol{\theta}^{b_k}}^b).$$

Taking variances under $A \sim \pi_{\boldsymbol{\theta}}^m$ gives

$$\mathbb{V}_{\pi^m(A)} \left( \hat{\partial}_{\theta_i^{b_k}}^{\mathrm{LR}} J(\pi_{\boldsymbol{\theta}}^m) \right) = w_k^2 \, \mathbb{V}_{\pi^m(A)} \left( \rho_m^{b_k}(A) \, \hat{\partial}_{\theta_i^{b_k}}^{\mathrm{LR}} J(\pi_{\boldsymbol{\theta}^{b_k}}^b) \right),$$

which is exactly the first equality in the proposition.

For the mixture weight parameter $w_k$, we note that

$$\partial_{w_k} \log \pi_{\boldsymbol{\theta}}^m(A) = \frac{1}{\pi_{\boldsymbol{\theta}}^m(A)} \, \partial_{w_k} \left( \sum_{\ell=1}^{N} w_\ell \pi_{\boldsymbol{\theta}^{b_\ell}}^b(A) \right) = \frac{\pi_{\boldsymbol{\theta}^{b_k}}^b(A)}{\pi_{\boldsymbol{\theta}}^m(A)} = \rho_m^{b_k}(A),$$

so that $\hat{\partial}_{w_k}^{\mathrm{LR}} J(\pi_{\boldsymbol{\theta}}^m) \;=\; \rho_m^{b_k}(A) \, r(A)$. Taking the variance under $A \sim \pi_{\boldsymbol{\theta}}^m$ immediately yields the second equality:

$$\mathbb{V}_{\pi^m(A)} \left( \hat{\partial}_{w_k}^{\mathrm{LR}} J(\pi_{\boldsymbol{\theta}}^m) \right) = \mathbb{V}_{\pi^m(A)} \left( \rho_m^{b_k}(A) \, r(A) \right).$$

$\square$

*Proof of Lemma C.7.*

For an MRP sample we first draw $\epsilon \sim \mathcal{N}(0,1)$ *once* and deterministically construct the $N$ actions $A_k = f_{\boldsymbol{\theta}^{b_k}}(\epsilon) = \mu_k + \epsilon \sigma_k$. Hence

$$\hat{\partial}^{\text{MRP}}_{\theta_i^{b_k}} J(\pi_{\boldsymbol{\theta}}^m) = \partial_{\theta_i^{b_k}}\left(w_k\, r\big(f_{\boldsymbol{\theta}^{b_k}}(\epsilon)\big)\right) = w_k\, \hat{\partial}^{\text{RP}}_{\theta_i^{b_k}} J(\pi_{\boldsymbol{\theta}^{b_k}}^b),$$

so that

$$\mathbb{V}_{\mathcal{N}(\epsilon;0,1)}\left(\hat{\partial}^{\text{MRP}}_{\theta_i^{b_k}} J(\pi_{\boldsymbol{\theta}}^m)\right) = w_k^2 \mathbb{V}_{\mathcal{N}(\epsilon;0,1)}\left(\hat{\partial}^{\text{RP}}_{\theta_i^{b_k}} J(\pi_{\boldsymbol{\theta}^{b_k}}^b)\right).$$

For the weight parameter, $\hat{\partial}^{\text{MRP}}_{\theta_k^w} J(\pi_{\boldsymbol{\theta}}^m) = r\big(f_{\boldsymbol{\theta}^{b_k}}(\epsilon)\big)$. Because $A = f_{\boldsymbol{\theta}^{b_k}}(\epsilon)$ with $A \sim \pi_{\boldsymbol{\theta}^{b_k}}^b$, the stated identity of variances follows immediately. $\square$

*Proof of Proposition C.10.*

To prove the inequality in the proposition, it is sufficient to show, for $\theta_i^k \in \{\mu_k, \sigma_k, w_k\}$,

$$\sum_{k=1}^{N}\left(\mathbb{V}_{\pi^m(A)}\left(\hat{\partial}^{\text{LR}}_{\theta_i^k} J(\pi_{\boldsymbol{\theta}}^m)\right) - \mathbb{V}_{\mathcal{N}(\epsilon;0,1)}\left(\hat{\partial}^{\text{MRP}}_{\theta_i^k} J(\pi_{\boldsymbol{\theta}}^m)\right)\right) \geq 0. \tag{22}$$

For $\theta_i^k$, this is immediate after applying Propositions C.6 and C.7 under Assumption C.9.

For $\theta_i^k \in \{\mu_k, \sigma_k\}$, by Propositions C.6 and C.7, Equation (22) holds if the following is satisfied

$$\sum_{k=1}^{N}\left(\mathbb{V}_{\pi^m(A)}\left(\rho_m^{b_k}(A)\hat{\partial}^{\text{LR}}_{\theta_i^{b_k}} J(\pi_{\boldsymbol{\theta}^{b_k}}^b)\right) - \mathbb{V}_{\mathcal{N}(\epsilon;0,1)}\left(\hat{\partial}^{\text{RP}}_{\theta_i^{b_k}} J(\pi_{\boldsymbol{\theta}^{b_k}}^b)\right)\right) \geq 0.$$

Under Assumption C.8, it is sufficient to show that the sum of the variance of the LR estimator over all component policies is larger than that of the RP estimator:

$$\sum_{k=1}^{N}\left(\mathbb{V}_{\pi^{b_k}(A)}\left(\hat{\partial}^{\text{LR}}_{\theta_i^{b_k}} J(\pi_{\boldsymbol{\theta}^{b_k}}^b)\right) - \mathbb{V}_{\mathcal{N}(\epsilon;0,1)}\left(\hat{\partial}^{\text{RP}}_{\theta_i^{b_k}} J(\pi_{\boldsymbol{\theta}^{b_k}}^b)\right)\right) \geq 0. \tag{23}$$

Next, we show the above inequality holds for $\theta_i^{b_k} = \mu_k$ under our assumptions, following by the result for $\theta_i^{b_k} = \sigma_k$.

Since $\pi_{\boldsymbol{\theta}^{b_k}}^b$ is Gaussian, we have

$$\hat{\partial}^{\text{LR}}_{\mu_k} J(\pi_{\boldsymbol{\theta}^{b_k}}^b) = r(A)\frac{A - \mu_k}{\sigma_k^2}, \quad \hat{\partial}^{\text{RP}}_{\mu_k} J(\pi_{\boldsymbol{\theta}^{b_k}}^b) = r'(\mu_k + \epsilon \sigma_k).$$

Following Gal (2016), we use Proposition 3.2 in Cacoullos (1982), which states if a real-valued function $g(a)$ has finite variance and its first derivative $g'(a)$ is absolutely integrable under $\mathcal{N}(\mu, \sigma)$, then

$$\sigma^2 \mathbb{E}_{\mathcal{N}(A;\mu,\sigma)}\left[g'(A)\right]^2 \leq \mathbb{V}_{\mathcal{N}(A;\mu,\sigma)}\left[g(A)\right] \leq \sigma^2 \mathbb{E}_{\mathcal{N}(A;\mu,\sigma)}\left[g'(A)^2\right].$$

Invoking the above proposition with $g(a) = (a - \mu_k)r(a)$ and $g(a) = r'(a)$, respectively, we have

$$\sigma_k^2 \mathbb{E}_{\pi^{b_k}(A)}\left[(A - \mu_k)r'(A) + r(A)\right]^2 \leq \mathbb{V}_{\pi^{b_k}(A)}\big((A - \mu_k)r(A)\big),$$
$$\mathbb{V}_{\pi^{b_k}(A)}\big(r'(A)\big) \leq \sigma_k^2 \mathbb{E}_{\pi^{b_k}(A)}\left[r''(A)^2\right].$$

Under Assumption C.8, we then have

$$\sum_{k=1}^{N} \mathbb{V}_{\mathcal{N}(\epsilon;0,1)} \left( \hat{\partial}_{\mu_k}^{\mathrm{RP}} J(\pi_{\boldsymbol{\theta}^{b_k}}^b) \right) = \sum_{k=1}^{N} \mathbb{V}_{\mathcal{N}(\epsilon;0,1)} \left( r'(\mu_k + \epsilon\sigma_k) \right)$$

$$= \sum_{k=1}^{N} \mathbb{V}_{\pi^{b_k}} \left( r'(A) \right)$$

$$\leq \sum_{k=1}^{N} \sigma_k^2 \mathbb{E}_{\pi^{b_k}(A)} \left[ r''(A)^2 \right]$$

$$\leq \sum_{k=1}^{N} \frac{1}{\sigma_k^2} \mathbb{E}_{\pi^{b_k}(A)} \left[ (A - \mu_k) r'(A) + r(A) \right]^2$$

$$\leq \sum_{k=1}^{N} \frac{1}{\sigma_k^4} \mathbb{V}_{\pi^{b_k}(A)} \left( (A - \mu_k) r(A) \right)$$

$$= \sum_{k=1}^{N} \mathbb{V}_{\pi^{b_k}(A)} \left( \hat{\partial}_{\mu_k}^{\mathrm{LR}} J(\pi_{\boldsymbol{\theta}^{b_k}}^b) \right),$$

which proves Equation (23) for $\theta_i^k = \mu_k$.

Similarly, for $\theta_i^k = \sigma_k$, we have

$$\hat{\partial}_{\sigma_k}^{\mathrm{LR}} J(\pi_{\boldsymbol{\theta}^{b_k}}^b) = r(A) \frac{(A - \mu_k)^2 - \sigma_k^2}{\sigma_k^3}, \quad \hat{\partial}_{\sigma_k}^{\mathrm{RP}} J(\pi_{\boldsymbol{\theta}^{b_k}}^b) = \epsilon r'(\mu_k + \epsilon\sigma_k).$$

Invoking Proposition 3.2 in Cacoullos (1982) with $g(a) = \left( (a - \mu_k)^2 - \sigma_k^2 \right) r(a)$ and $g(a) = (a - \mu_k) r'(a)$, respectively, we have

$$\sigma_k^2 \mathbb{E}_{\pi^{b_k}(A)} \left[ \left( (A - \mu_k)^2 - \sigma_k^2 \right) r'(A) + 2(A - \mu_k) r(A) \right]^2 \leq \mathbb{V}_{\pi^{b_k}(A)} \left( \left( (A - \mu_k)^2 - \sigma_k^2 \right) r(A) \right),$$

$$\mathbb{V}_{\pi^{b_k}(A)} \left( (A - \mu_k) r'(A) \right) \leq \sigma_k^2 \mathbb{E}_{\pi^{b_k}(A)} \left[ \left( (A - \mu_k) r''(A) + r'(A) \right)^2 \right].$$

Under Assumption C.8, we have

$$\sum_{k=1}^{N} \mathbb{V}_{\mathcal{N}(\epsilon;0,1)} \left( \hat{\partial}_{\sigma_k}^{\mathrm{RP}} J(\pi_{\boldsymbol{\theta}^{b_k}}^b) \right) = \sum_{k=1}^{N} \mathbb{V}_{\mathcal{N}(\epsilon;0,1)} \left( \epsilon r'(\mu_k + \epsilon\sigma_k) \right)$$

$$= \sum_{k=1}^{N} \mathbb{V}_{\pi^{b_k}} \left( r'(A) \frac{A - \mu_k}{\sigma_k} \right)$$

$$= \sum_{k=1}^{N} \frac{1}{\sigma_k^2} \mathbb{V}_{\pi^{b_k}} \left( (A - \mu_k) r'(A) \right)$$

$$\leq \sum_{k=1}^{N} \mathbb{E}_{\pi^{b_k}(A)} \left[ \left( (A - \mu_k) r''(A) + r'(A) \right)^2 \right]$$

$$\leq \sum_{k=1}^{N} \frac{1}{\sigma_k^4} \mathbb{E}_{\pi^{b_k}(A)} \left[ \left( (A - \mu_k)^2 - \sigma_k^2 \right) r'(A) + 2(A - \mu_k) r(A) \right]^2$$

$$\leq \sum_{k=1}^{N} \frac{1}{\sigma_k^6} \mathbb{V}_{\pi^{b_k}(A)} \left( \left( (A - \mu_k)^2 - \sigma_k^2 \right) r(A) \right)$$

$$= \sum_{k=1}^{N} \mathbb{V}_{\pi^{b_k}(A)} \left( \hat{\partial}_{\sigma_k}^{\mathrm{LR}} J(\pi_{\boldsymbol{\theta}^{b_k}}^b) \right),$$

which proves Equation (23) for $\theta_i^k = \sigma_k$. $\qquad\square$

## D  STATIONARY POINT STUDY DETAILS

In this section, we provide more details on the stationary point study presented in Section 3.2. The reward function of the bimodal bandit is the normalized summation of two Gaussians' density functions whose standard deviations are both $0.5$ and whose means are $-1$ and $1$, respectively. We use the default optimization algorithm for variable with bounds in SciPy (Virtanen et al., 2020) to optimize the entropy regularized objective $J(\pi_{\boldsymbol{\theta}}) = \mathbb{E}_{a \sim \pi_{\boldsymbol{\theta}}}[r(a) - \alpha \log \pi_{\boldsymbol{\theta}}(a)]$, where $r(a)$ is the value of the action depicted in Figure 1. We then sort the obtained stationary points based on their regularized values $J(\pi_{\boldsymbol{\theta}})$ and use the ones that have the highest values as the parameters of optimal policies for Figure 1.

For each policy class, we run the default optimization algorithm for 100 trials, each with a set of randomly sampled initial policy parameters. Specifically, the initial means, log standard deviations, and mixing weights are randomly sampled from $[-2, 2]$, $[-3, 0]$, and $[0, 1]$, respectively. To avoid numerical issues in numerical integral when the standard deviation gets too large, we impose an upper bound of 3 for the log standard deviation. In addition, the mixing weights are defined and bounded within $[0, 1]$. We initially run this optimization procedure for seven different entropy scales: $\alpha \in \{0.05, 0.1, 0.2, 0.3, 0.4, 0.5, 0.6\}$. With difference shown when $\alpha$ is between $0.2$ and $0.5$ (see Figure 1 (Middle)), we additionally run another eight entropy scales: $\alpha \in \{0.22, 0.24, 0.26, 0.28, 0.325, 0.35, 0.375, 0.45\}$ to obtain more insights when $\alpha$ within this range. Note that, we did not obtain any convergent results for the Gaussian policy when $\alpha >= 0.325$ and for the Gaussian mixture (GM) policy when $\alpha > 0.5$.

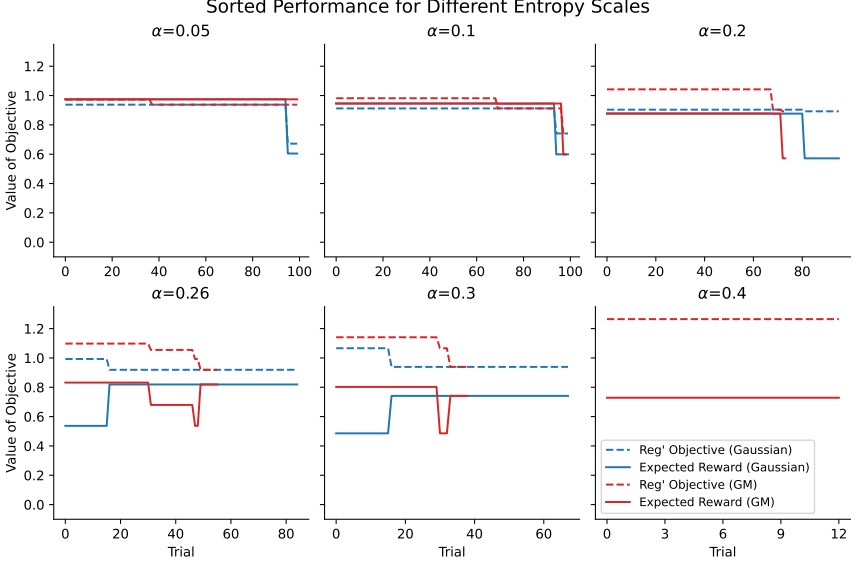

Figure 10: Expected reward and objective value of the regularized objective's stationary points found in 100 trials for the Gaussian and the Gaussian mixture policies. Note that the stationary points are sorted by their regularized objective value for clarity (dashed line). When a line ends before Trial 100, it means that the rest of the trials either diverged or encountered a numerical issue. The difference between the dash line and the solid line represents the differential entropy scaled by $\alpha$.

Figure 10 shows the expected reward and objective value of all the stationary points found for representative $\alpha$. Apart from the observation discussed in Section 3.2, we can see that both the Gaussian and the GM mixture often have different types of stationary points. The optimal Gaussian policy concentrates on one of the reward modes with a high expected reward with small $\alpha$, but it then shifts to the middle of two modes in $\alpha = 0.26$ and $\alpha = 0.3$ with a lower expected reward but a much higher entropy. The optimal GM policy, on the other hand, always has two modes covering two reward modes: it obtains a high reward while maintaining a higher entropy.

# E  EXPERIMENT DETAILS

## E.1  EXPERIMENTAL DETAILS

In this section, we supply the omitted experimental details in Section 5. In addition, we will open source our code for reproducibility after publication.

**Common configurations.**    For SAC with a likelihood-ratio gradient for the actor, we sample $N_b = 30$ actions for the given state and use the average of the corresponding action values as the baseline. We also use such a baseline for the likelihood-ratio part of the half-reparameterization gradient estimator. We use the Adam optimizer (Kingma & Ba, 2014) with $\beta_1 = 0.9$ and $\beta_2 = 0.999$ for all experiments.

**Synthetic multimodal bandits.**    We create 100 continuous bandits with the reward function proportional to the summation of 30 randomly generated Gaussian density functions. The means and standard deviations are uniformly sampled from $[-3, 3]$ and $[0.1, 1.0]$, respectively. We use a two-layer feedforward network with a hidden dimension of 16, a replay buffer size of 5000, and a batch size of 32. Note that we replace the critic with the true value function as these bandits with a deterministic reward function are for illustration purposes. We sweep the initial actor step size $\eta_{p,0} = 10^x$ for $x \in \{-4, -3, -2, -1\}$ and the entropy scale $\alpha = 10^y$ for $y \in \{-4, -3, -2, -1\}$. We report additional plots in Appendix E.3.

**Common continuous control benchmarks.**    We use a two-layer feedforward network with a hidden dimension of 256, a replay buffer size of $1,000,000$, and a batch size of 100. We use the automatic entropy tuning and the same initial step size $3 \times 10^{-4}$ for the actor, critic, and entropy scale. We use a double Q network and a target network for the critic, which is an exponential moving average of the critic with a smoothing factor of 0.005. In the initial $10,000$ steps, the actions are uniformly sampled. We report the state and action dimensions of Gym MuJoCo environments in Table 2 and learning curves in each individual environment in Appendix E.4.

**Classic control environments.**    We use a two-layer feedforward network with a hidden dimension of 64, a replay buffer size of $100,000$, and a batch size of 32. We use a double Q network and a target network for the critic, which is an exponential moving average of the critic with a smoothing factor of 0.01. We sweep the entropy scale $\alpha = 10^y$ for $y \in \{-3, -2, -1, 0\}$. In addition, we sweep the initial critic step size $\eta_{q,0} = 10^x$ for $x \in \{-5, -4, -3, -2\}$ and the initial actor step size $\eta_{p,0} = \kappa \eta_{q,0}$ for $\kappa \in \{10^{-2}, 10^{-1}, 1, 10\}$. Details of the versions and reward functions of these environment is in Section E.5. We report the best hyperparameter setting in Table 3 and additional plots in Section E.6.

## E.2  COMPUTATIONAL OVERHEAD OF USING MIXTURE POLICIES

**Network architecture of mixture policies.**    Compared to the base policy's actor, the mixture policy's actor has additional heads for the additional parameters in the mixture distribution. For example, the last layer of a squashed Gaussian policy's actor has two outputs (one for the mean and one for the standard deviation), while the last layer of a squashed Gaussian policy's actor with five components has 15 outputs (five for the means, five for the standard deviations, and five for the mixing weights). Such an architecture induces only negligible additional memory usage (see Table 4). Similarly, the inference time could also be marginal.

**Computational overhead during training.**    Using the MRP estimator does slightly increase actor training cost as it requires $N$ Q-evaluations per state, compared to 1 for Gaussian policies. However, the additional wall-clock training time is modest, since computation can be batched and parallelized. Further, a lot of SAC's wall-clock time comes from critic updates (each with 4 separate Q-evaluations per transition, a factor of 2 from double-Q and another factor of 2 from two consecutive states).

**Reference training and inference time.**    We provide training and inference time samples for `Pendulum` and `HalfCheetah-v3` in Table 4 for reference. We can see that mixture policies

require modest additional training and inference time. However, we use PyTorch (Paszke et al., 2019) in our experiments and have not optimized our code for more efficient training and inference. We expect the gap between mixture policies and base policies to be much smaller if one switch to a JAX implementation (Bradbury et al., 2018).

For transparency, the training time samples are obtained via an example run when the server is idle and no other active program is running. The CPU of the server is AMD Ryzen 9 5900X 12-Core Processor, and the GPU of the server is NVIDIA Geforce RTX 3080 Ti.

Table 2: State and action dimensions of MuJoCo environments.

| Environment | State Dimension | Action Dimension |
|---|---|---|
| Hopper-v3 | 11 | 3 |
| Walker2d-v3 | 17 | 6 |
| HalfCheetah-v3 | 17 | 6 |
| Ant-v3 | 111 | 8 |
| Swimmer-v3 | 8 | 2 |
| Humanoid-v3 | 376 | 17 |
| HumanoidStandup-v2 | 376 | 17 |

Table 3: Tuned hyperparameters in classic control environments.

| Environment | Algorithm | $\eta_{q,0}$ | $\kappa$ | $\alpha$ |
|---|---|---|---|---|
| ShapedPendulum | SG-RP | $10^{-2}$ | $10^{-1}$ | $10^{-1}$ |
| | USGM-RP | $10^{-2}$ | $10^{-1}$ | $10^{-1}$ |
| | SGM-HalfRP | $10^{-2}$ | $10^{-1}$ | $10^{-1}$ |
| | SGM-MRP | $10^{-2}$ | $10^{-1}$ | $10^{-2}$ |
| | SGM-GumbelRP | $10^{-2}$ | $10^{-1}$ | $10^{-1}$ |
| ShapedAcrobot | SG-RP | $10^{-2}$ | $10^{-2}$ | $10^{-2}$ |
| | USGM-RP | $10^{-2}$ | $10^{-1}$ | $10^{-3}$ |
| | SGM-HalfRP | $10^{-2}$ | $10^{-1}$ | $10^{-2}$ |
| | SGM-MRP | $10^{-2}$ | $10^{-1}$ | $10^{-2}$ |
| | SGM-GumbelRP | $10^{-2}$ | $10^{-1}$ | $10^{-2}$ |
| ShapedMountainCar | SG-RP | $10^{-3}$ | $1$ | $10^{-1}$ |
| | USGM-RP | $10^{-2}$ | $10^{-1}$ | $10^{-1}$ |
| | SGM-HalfRP | $10^{-2}$ | $10^{-1}$ | $10^{-1}$ |
| | SGM-MRP | $10^{-3}$ | $1$ | $10^{-1}$ |
| | SGM-GumbelRP | $10^{-3}$ | $1$ | $10^{-1}$ |
| Pendulum | SG-RP | $10^{-3}$ | $1$ | $10^{-2}$ |
| | USGM-RP | $10^{-3}$ | $1$ | $10^{-2}$ |
| | SGM-HalfRP | $10^{-3}$ | $1$ | $10^{-2}$ |
| | SGM-MRP | $10^{-3}$ | $1$ | $10^{-3}$ |
| | SGM-GumbelRP | $10^{-3}$ | $1$ | $10^{-2}$ |
| Acrobot | SG-RP | $10^{-3}$ | $10^{-2}$ | $10^{-3}$ |
| | USGM-RP | $10^{-3}$ | $1$ | $10^{-3}$ |
| | SGM-HalfRP | $10^{-3}$ | $1$ | $10^{-3}$ |
| | SGM-MRP | $10^{-3}$ | $1$ | $10^{-2}$ |
| | SGM-GumbelRP | $10^{-3}$ | $10^{-1}$ | $10^{-3}$ |
| MountainCar | SG-RP | $10^{-4}$ | $10$ | $10^{-2}$ |
| | USGM-RP | $10^{-4}$ | $10$ | $10^{-3}$ |
| | SGM-HalfRP | $10^{-3}$ | $1$ | $10^{-2}$ |
| | SGM-MRP | $10^{-4}$ | $10$ | $10^{-2}$ |
| | SGM-GumbelRP | $10^{-4}$ | $10$ | $10^{-2}$ |

Table 4: Training time for $100k$ steps, inference time for 110 episodes, and GPU memory usage during training.

|  |  | SG-RP | SGM-MRP | SGM-HalfRP | SGM-GumbelRP |
|---|---|---|---|---|---|
| Pendulum | CPU Training | 317s | 455s | 445s | 435s |
|  | CPU Inference | 2.1s | 3.0s | 3.0s | 3.0s |
| HalfCheetah | GPU Training | 618s | 818s | 797s | 807s |
|  | GPU Inference | 23s | 30s | 30s | 31s |
|  | GPU Memory | 554MB | 556MB | 554MB | 554MB |

### E.3 ADDITIONAL PLOTS FOR SYNTHETIC BANDITS EXPERIMENTS

Figure 11 shows the learning curves of SG-RP and SGM-MRP with their best hyperparameter setting in all 100 synthetic multimodal bandits. We can see that SGM-MRP outperforms SG-RP in many bandits where they are not tied.

## Results on Each Reward Function

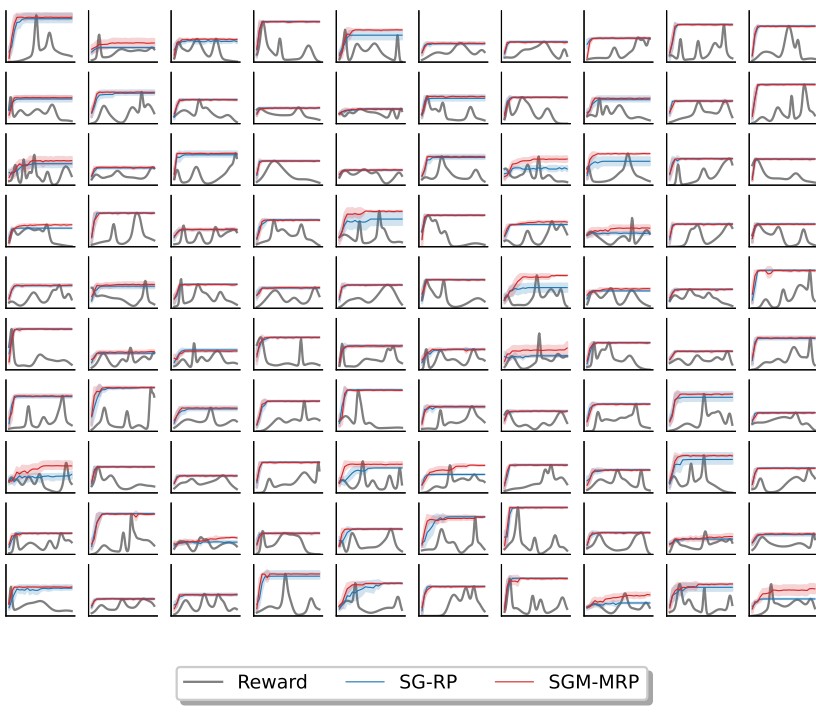

Figure 11: Reward function and the corresponding learning curves for each synthetic bandit. Each bandit is run for 10 seed. The shade area shows the $95\%$ bootstrap CIs.

Figure 12 shows the average reward in the final $10\%$ of training steps for the hyperparameter setting with the best final reward across different $\alpha$ for different estimators. We can see that the SGM-MRP is consistently better than alternatives.

### E.4 ADDITIONAL PLOTS FOR COMMON CONTINUOUS CONTROL BENCHMARKS

Figures 13 and 14 show the learning curves in each of the MuJoCo, DeepMind Control (DMC), MetaWorld, and MyoSuite environments. We can see that the performance of SGM-MRP is quite similar to SG-RP across different environments except for a few environments in MetaWorld.

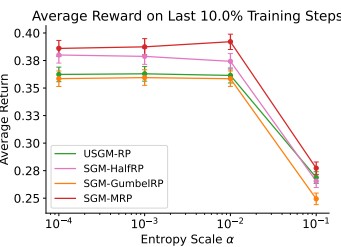

Figure 12: Average performance across 100 bandits at different entropy scales. The shaded areas and error bars show $95\%$ bootstrap CIs across 100 bandits (10 runs each).

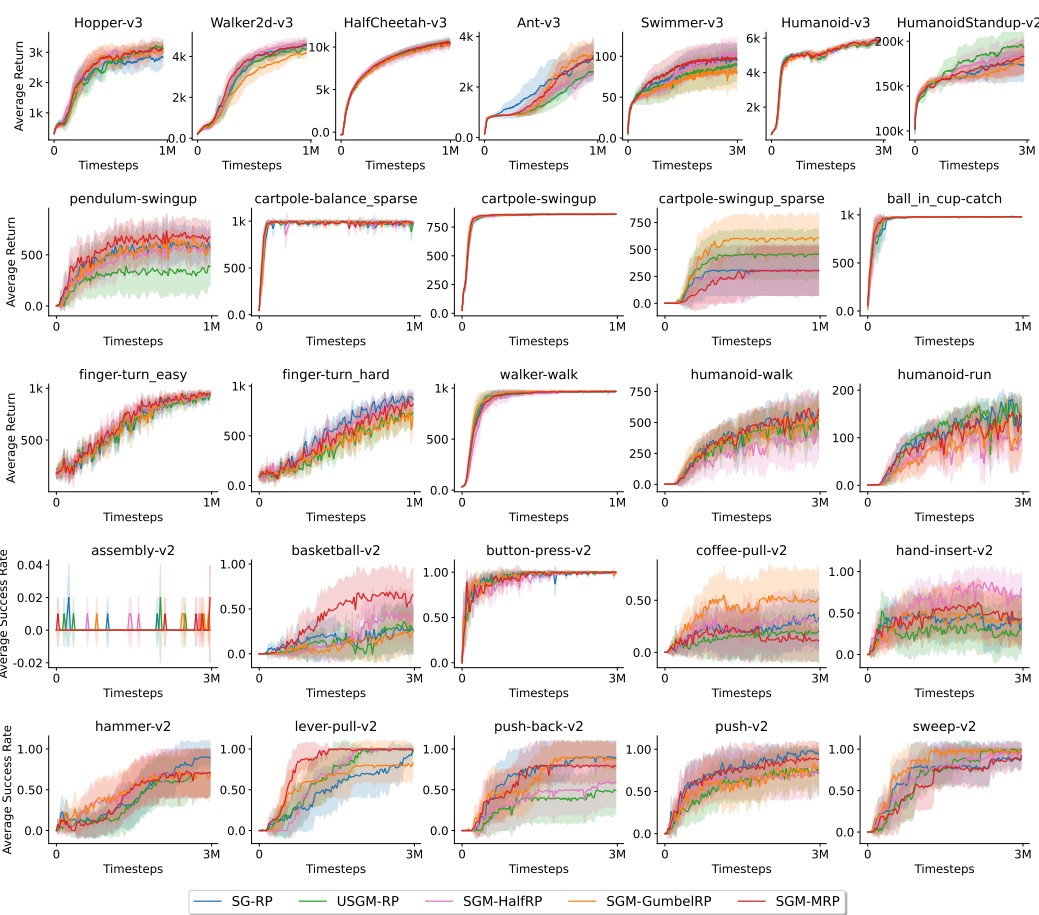

Figure 13: Learning curves in 27 environments from Gym MuJoCo, DMC, and MetaWorld.

Figure 15 shows the learning curves in two MetaWorld environments where mixture policies fail to achieve meaningful successful rate. We can see that SGM-MRP still learns steadily in terms of average return in these cases. The insignificant success rate might be due to SGM-MRP's convergence to a stable but non-successful mode, whereas SG-RP occasionally discovers the success trajectory.

Figure 16 shows the sensitivity of mixture policies and base policies in the remaining two MuJoCo environments, in which the difference is smaller than those in Figure 5.

### E.5 CLASSIC CONTROL ENVIRONMENT DETAILS

We use the v1 version of Pendulum from OpenAI Gym for `ShapedPendulum`. For `Pendulum`, we set the reward to 1 if the angle of the pendulum from the upright position is smaller than 0.25 and

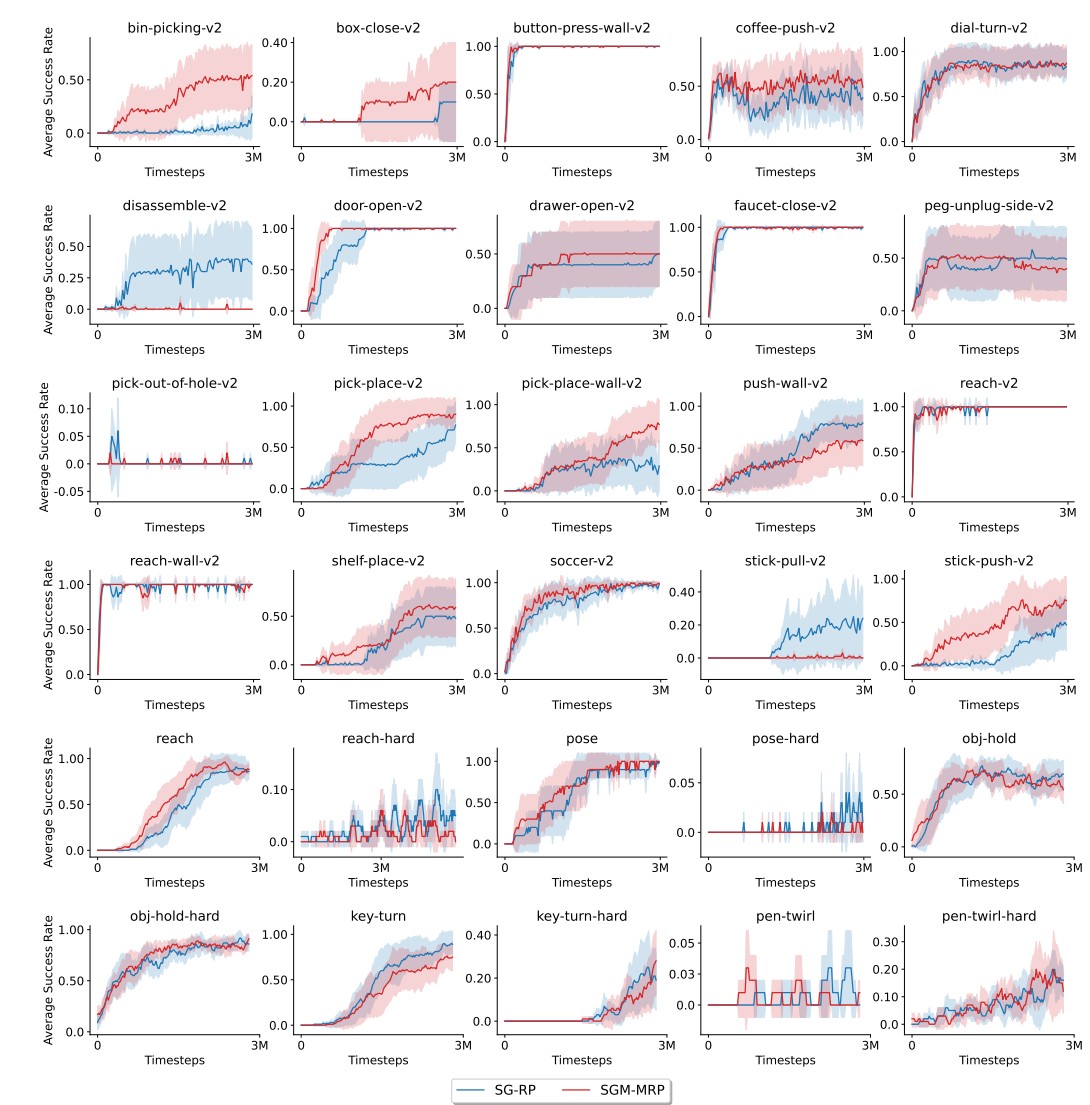

Figure 14: Learning curves in 30 additional environments from MetaWorld and MyoSuite.

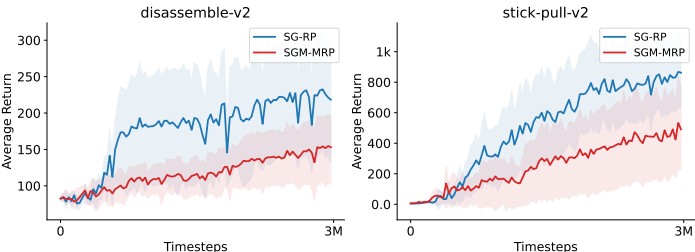

Figure 15: Learning curves in two MetaWorld environments where mixture policies fail to achieve meaningful successful rate.

0 otherwise. For `ShapedAcrobot`, we set the reward to be $-\cos(\theta_1) - \cos(\theta_2 + \theta_1) - 1.0$, where $\theta_1$ and $\theta_2$ are the first two dimensions of the state. For `ShapedMountainCar`, we set the reward to be $x - 0.6$, where $x$ is first dimension of the state. For `Acrobot` and `MountainCar`, we adapt the discrete version in Gym to the continuous action case as it is done in Neumann et al. (2022). All

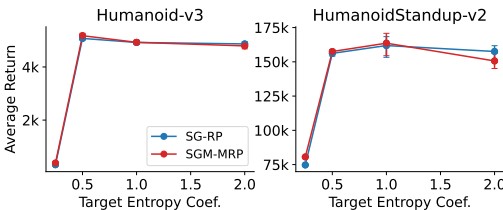

Figure 16: Performance with different target entropy coefficients for mixture policies and base policies in the remaining two MuJoCo environments. Results are averaged over 10 runs.

environments use a discount factor of 0.99. The episode cut-offs for the Pendulum, Acrobot, and MountainCar are 200, 1000, and 1000, respectively.

### E.6 ADDITIONAL PLOTS FOR CLASSIC CONTROL ENVIRONMENTS

**Results comparing SGM-MRP and SG-RP in the unshaped-reward setting.** Figure 17 shows learning and sensitivity curves. Compared to those in Figure 6, the improvement of SGM-MRP over SG-RP is much smaller (note the range of y-axis in the sensitivity curves).

**Results comparing different estimators for mixture policies.** From Figure 18, we can see that MRP is consistently the best estimator for mixture policies. Note that SGM-HalfRP has poor performance in MountainCar environments, potentially due to its higher variance compared to others.

**Limitations of fixed weighting policies.** We discuss the drawbacks of HalfRP and Gumbel RP estimators in Section 6. Here, we discuss the limitation of the other alternative, using a fixed weighting policy. First, restricting the weighting scheme reduces the flexibility of the policy class, which may be undesirable. Second, mixture policies with fixed weights often require more significant parameter updates when transitioning between distributions. For instance, when all modes have collapsed to a single mode, it is more challenging for fixed-weight mixture policies to introduce a new mode far from the current mode, as the component locations are constrained near the existing mode due to the non-zero fixed weights. In contrast, mixture policies with learnable weights can focus on a specific mode while keeping other components positioned far away, as their negligible weights minimize their impact on the resulting distribution. In scenarios where the mixture policy needs to introduce a new mode, a learnable-weight policy can simply adjust the mixing weight of a component that is already near the desired mode, enabling more efficient adaptation.

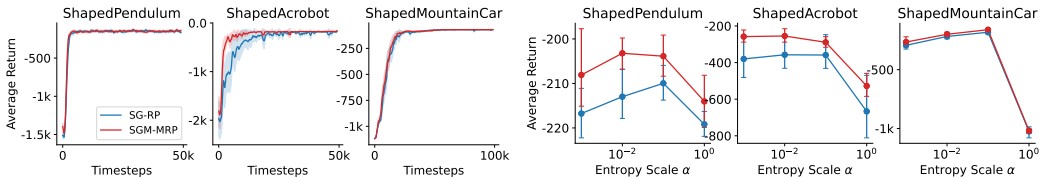

Figure 17: Learning and sensitivity curves for classic control environments with shaped rewards.

**Visualization of state visitation.** To quantify exploration coverage and visualize state visitation, we collect visited states in 30 reruns of the best hyperparameter setting for SG-RP and SGM-MRP in `MountainCar`. Specifically, we sample $10,000$ states after learning for $y \in \{0, 4000, 8000\}$ training steps. We then bin the states aggregating across different reruns and steps into $50 \times 50$ bins, with 50 bins per state dimension. Finally, we subtract the frequency of each bin for SGM-MRP from that for SG-RP and plot the difference in log scale in Figure 7 for $y = 4000$. In Figure 19, we also show the results for $y = 0$ and $y = 8000$ as well as per-algorithm visitation. The starting states are around $(-0.5, 0.0)$, and a successful trajectory would spiral away from $(-0.5, 0.0)$ and reach the dash line. In Figure 19, the difference in visitation on states far away from the starting states demonstrates the better exploration efficiency of mixture policies.

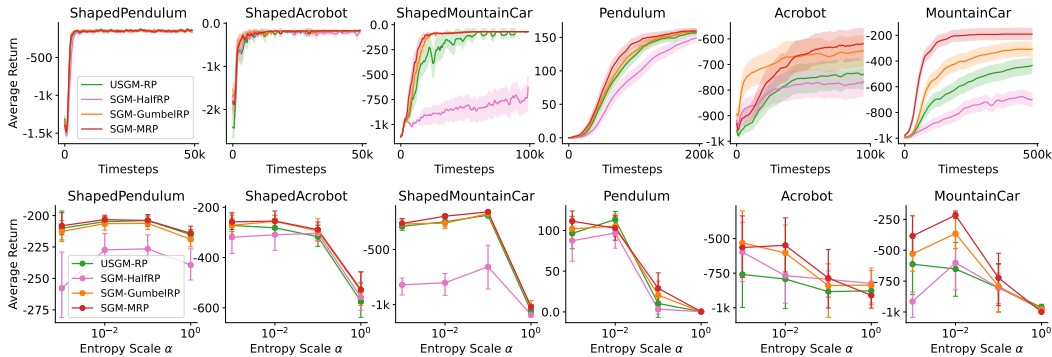

Figure 18: Learning curves of the best hyperparameter setting of different estimators for mixture policies in classic control environments.

**Visualization of the action-value estimates and policy density for base policies.**    We presented the visualization of the action-value estimates and policy density at the starting state (Figure 20) for mixture policies in Figure 8 to highlight the difference between `ShapedMountainCar` and `MountainCar`, one with shaped rewards and the other with unshaped rewards. In Figure 21, we show the same plot for base policies, SG-RP. The difference between the two types of environments is not so much different from that in Figure 8, but we can see the density is always unimodal during training, indicating less efficient exploration.

**Mixture policy statistics during training.**    We investigate three statistics of mixture policies:

- Entropy. We measure the entropy of the weighting policies to see how much mixture policies collapse to choosing only one components.
- Component separation. This metric measures the pairwise distance between the (squashed) means of all components.
- Active component separation.  This metric measures the pairwise distance between the means of components whose weight is larger than $0.01$.

From Figure 22, there are several interesting observations. First, we can see that while the entropy of the weighting policies decay as training goes on, it stays above zero. Second, in most environments, both component separation metrics maintain a relatively high value (note that the action space is $[-1, 1]$), showing that components maintain meaningful separation during training. Third, the level of entropy and component separation varies across environments. This makes sense as the rewards and the optimal entropy coefficient (see Table 3) for different environments differ from each other.

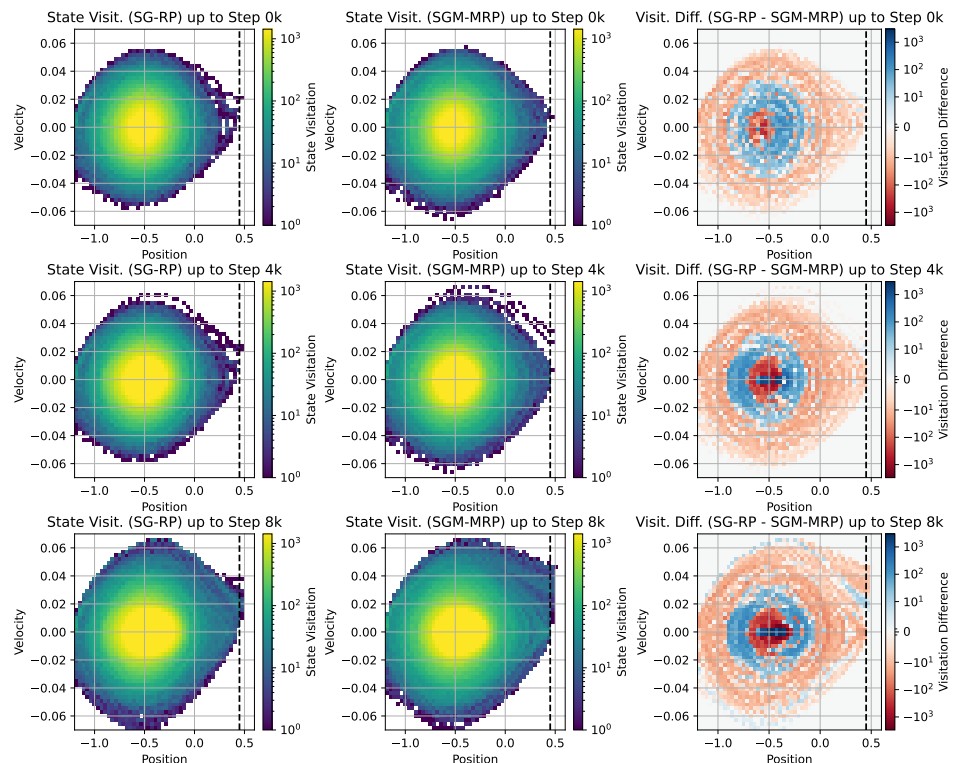

Figure 19: State visitation of SG-RP and SGM-MRP during early training in `MountainCar`.

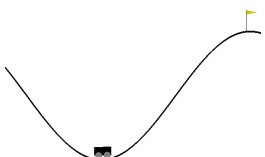

Figure 20: The starting state at the bottom in MountainCar.

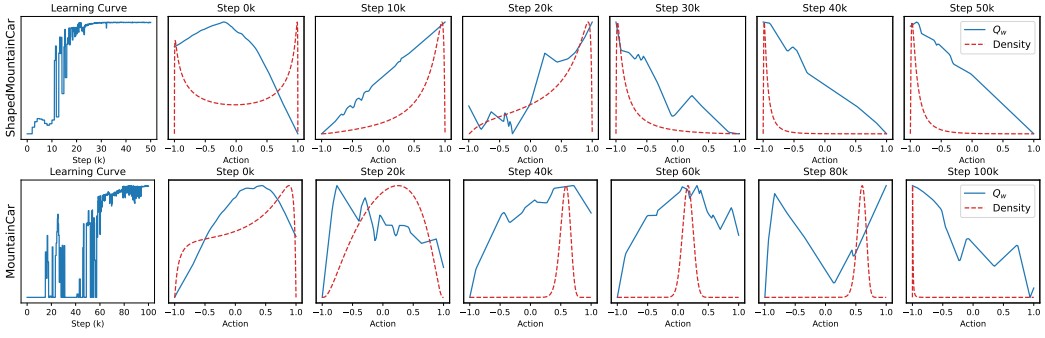

Figure 21: Learning curve, action-value estimates, and policy density at a starting state from a sample run of SG-RP in two MountainCar variants. The y-axes differ across plots and are not shown with ticks to highlight the shape of the curves rather than their exact values. The observations are similar to those in Figure 8 with the exception that the density is always unimodal except for the starting step. It indicates that SG-RP explore less efficiently, potentially explaining its worse performance in `MountainCar`.

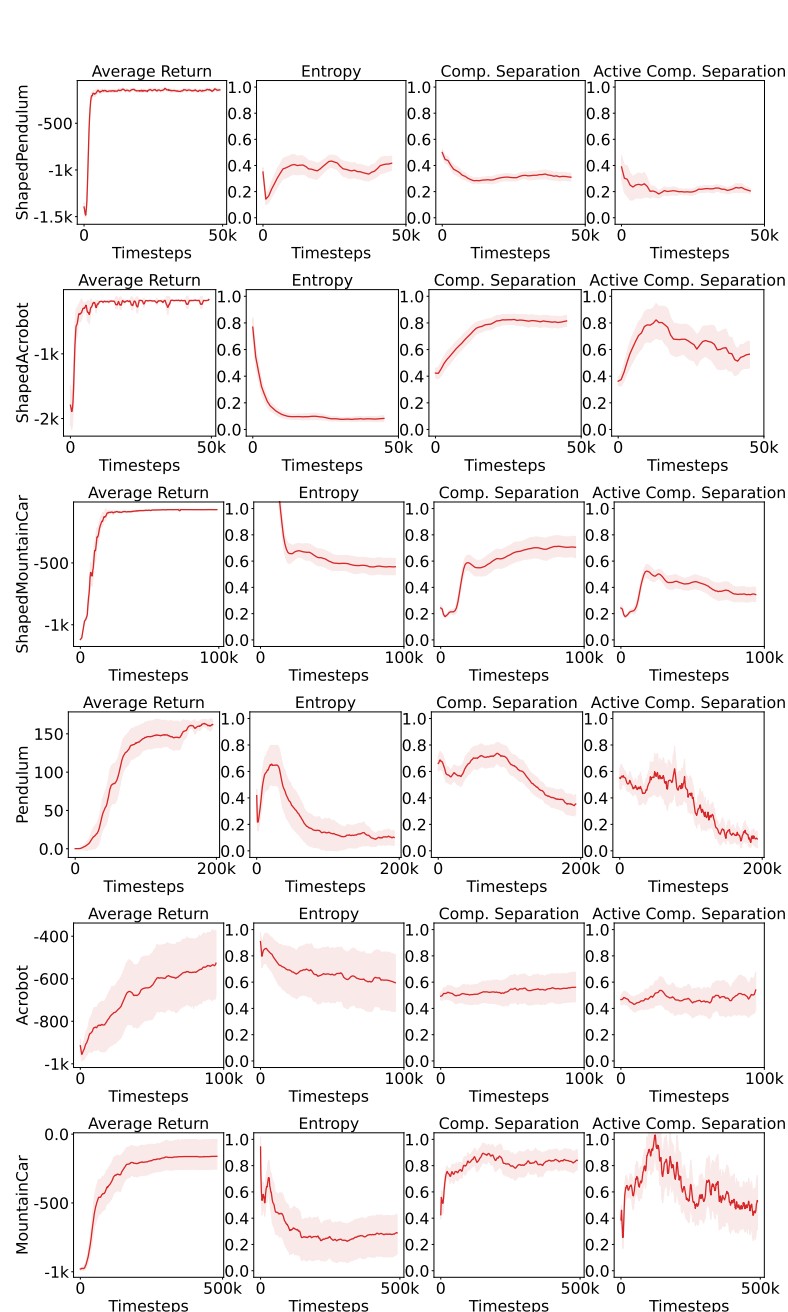

Figure 22: Mixture policy statistics during training in classic control environments. Results are averaged across 30 runs. The shaded areas plot the 95% bootstrap CIs.

# F ADDITIONAL EXPERIMENTS

In this section, we provide additional experiments that further study different aspects when using mixture policies.

## F.1 COMPARISON BETWEEN THE RP AND THE LR ESTIMATORS

In this section, we compare the MRP estimator with the LR estimator for optimizing mixture policies. To reduce variance of the LR estimator, we employ a baseline calculated using the average of the action values of 30 sampled actions for a given state. We conduct experiments on seven Gym MuJoCo environments, following the same experimental setup as Section 5.

**The LR estimator could be unstable in high-dimensional environments.** Figure 23 contrasts the performance distribution of SGM-LR to that of SGM-MRP. We can see that SGM-LR generally performs worse than SGM-MRP. Despite using a baseline to reduce variance, SGM-LR remains unstable, with divergent runs in several environments, particularly those with higher state and action dimensions (see Table 2 in Section E.1). This instability is evident in environments like `Humanoid` and `HumanoidStandup`, where SGM-LR exhibits a significant number of divergent runs (shown in black), while SGM-MRP consistently achieves high returns. Additionally, even for non-divergent runs, SGM-LR often fails to learning meaningful policies, resulting in lower return distributions (as in `Walker2d`, `Ant`, and `Swimmer`). The variance in gradient estimates remains a significant issue for likelihood-ratio-based methods, especially in high-dimensional environments. Overall, these results demonstrate that RP-based estimators provide greater stability than the LR estimator. This makes them a more effective choice for training mixture policies.

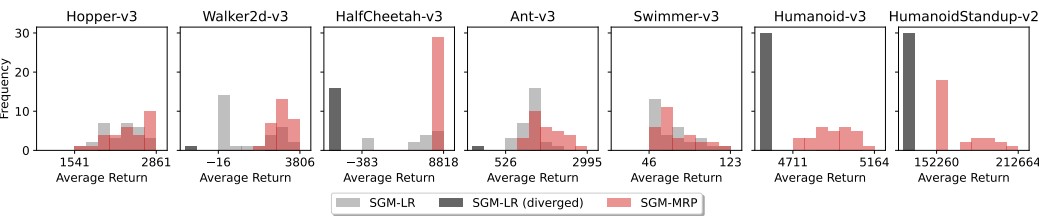

Figure 23: Comparative distribution (30 runs) of average return for squashed Gaussian mixture (SGM) policies in Gym MuJoCo environments, comparing the LR and the ERP estimators.

## F.2 ADDITIONAL ROBOTIC ENVIRONMENTS WITH UNSHAPED REWARDS

Other than the classic control environments in Section 5.3, we also investigate robotic environments with unshaped rewards. Specifically, we test `FetchReach` and `FetchSlide` from Plappert et al. (2018) as well as 10 in-hand manipulation environments from the ShadowHand Huang et al. (2021). However, both mixture policies and base policies fail without any learning progress in in-hand manipulation environments as they are too difficult. On the other hand, mixture policies appear to improve performance in the Fetch environments (Figure 24).

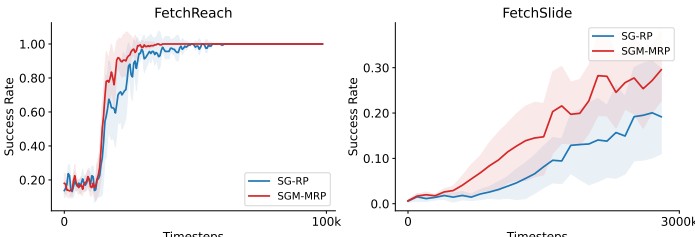

Figure 24: Learning curves in two robotic Fetch environments from Plappert et al. (2018). The shaded areas plot the 95% bootstrap CIs across 8 runs.

### F.3 EFFECT OF THE NUMBER OF COMPONENTS

We use mixture policies with five components in all experiments in the main text. Here, we study the effect of this choice in classic control environments with unshaped rewards, where differences between the base and the mixture policies are more prominent. We use the same experiment protocol as in Section 5.3 and test mixture policies with two and eight components using the MRP gradient estimator. Specifically, we sweep the same hyperparameters for each variant of mixture policies.

Figure 25 shows the sensitivity to the entropy scale. We can see that while the results are noisy and not quite consistent across environments, mixture policies with various numbers of components generally outperform the base policy. Despite significant noise in the results, we can see that mixture policies with various numbers of components are similarly effective.

We further test mixture policies with different number of components in high-dimensional continuous control environments. We use the same environments in Figure 3 where the performance gap between mixture policies and base policies is more obvious. The results are shown in Figure 26. While performance varies with $N$, the overall conclusion is similar to that in classic control domains: while $N = 2$ always slightly underperforms $N = 5$, mixture policies are similarly effective with different numbers of components.

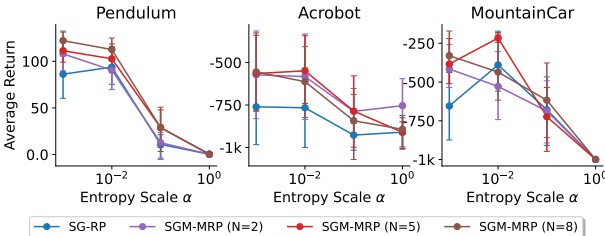

Figure 25: Sensitivity curves for mixture policies with different number of components. The error bars plot the $95\%$ bootstrap CIs across 10 runs.

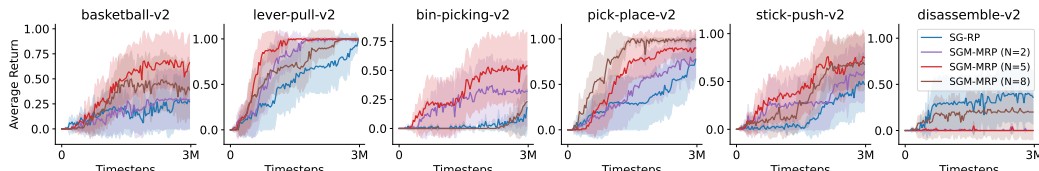

Figure 26: Learning curves for mixture policies with different number of components in selected MetaWorld tasks where the performance gap between mixture policies and base policies is most pronounced. Shaded areas show 95% bootstrap CIs over 10 runs.

**Guidance on selecting the number of components.** While our theoretical analysis in Sections 3 and 4 has not revealed potential downsides of using more components to increase the flexibility of the policy, there are practical considerations for using a restricted number of them. First, using a large number of components can increase the computational and memory cost (e.g., if $|\mathcal{A}| \times N$ is close to the hidden dimension size). Second, from the above empirical results, increasing $N$ from 5 to 8 does not consistently bring benefits, whereas decreasing $N$ from 5 to 2 appears to consistently worsen the performance. Based on the above points, we recommend to use a default of $N = 5$ and tune upwards or slightly downwards if needed.

### F.4 EFFECT OF USING A HEAVY-TAILED BASE POLICY

In principle, the base policy can be any policy and even be different across different components. Here, we consider Cauchy policy (Bedi et al., 2024) as the base policy, which is heavy-tailed and promotes persistent exploration. Our hypothesis is that *using mixture policies would also provide benefits when the base policy is Cauchy policy.* We adopt the same experiment protocol as in Section 5.3 and test Cauchy and Cauchy mixture (CM) policies with reparameterization gradient estimators.

Figure 27 shows the learning curves of the best hyperparameter setting (rerun for 200 seeds) and the sensitivity to the entropy scale. In `Pendulum` and `MountainCar`, Cauchy mixture policies significantly outperform base Cauchy policies. Notice that, in `Pendulum`, Cauchy-based policies perform significantly worse than the corresponding Gaussian-based policies. This might potentially be due to two reasons: 1) Cauchy-based policies are generally not suitable for this environment, or 2) the preset search range of $\alpha$ is too large for Cauchy-based policies. In summary, we conclude that even when the base policy is heavy-tailed, using mixture policies may still provide benefits in environments with unshaped rewards.

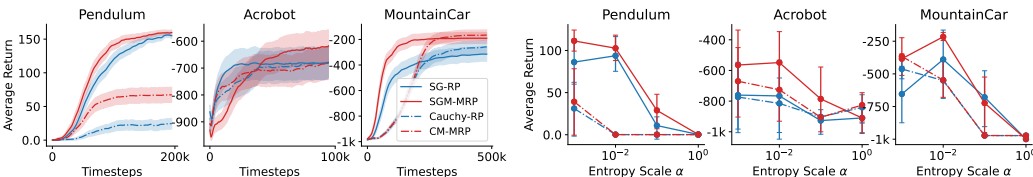

Figure 27: Learning and sensitivity curves for Cauchy policies and Cauchy Mixture (CM) policies in classic control environments with unshaped rewards.

### F.5 EFFECT OF THE TEMPERATURE PARAMETER OF GUMBELRP ESTIMATOR

In this section, we provide additional experiments that investigate the effect of the temperature parameter $\tau$ in GumbelRP estimator. We use two settings in this study: multimodal bandits and continuous control. In the bandit setting, we test GumbelRP with wide range of temperatures. Other hyperparameters are consistent with the best hyperparameters selected based on SGM-GumbelRP ($\tau = 1.0$). For continuous control, we test two additional temperature values in `Hopper-v3`. From Figure 28, we can see that the GumbelRP estimator is not sensitive to its temperature parameter in our experiments.

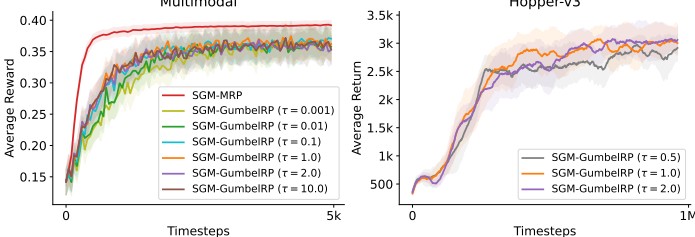

Figure 28: Sensitivity analysis of the GumbelRP estimator to the temperature parameter. The shaded areas plot the 95% bootstrap CIs across different runs. For `Multimodal`, the results are averaged over 100 bandits (10 runs each for SGM-MRP; 1 run each for SGM-GumbelRP). For `Hopper-v3`, the results are averaged over 10 runs.

## G USE OF LARGE LANGUAGE MODELS

We used large language models (LLMs) during the preparation of this paper in limited ways. Specifically: (1) LLMs were used to help polish and improve the clarity of writing. This included rephrasing sentences, tightening grammar, and suggesting more concise wording. All technical content, arguments, and contributions were conceived, derived, and validated by the authors; (2) LLMs were occasionally used to generate or refine scripts for processing experimental results and producing figures. All generated code was reviewed and, where necessary, modified by the authors to ensure correctness. LLMs were not used for research ideation, literature retrieval, or the creation of original technical content. Their role was limited to communication polish and auxiliary support.

