# OpenReview forum: "Revisiting Mixture Policies in Entropy-Regularized Actor-Critic"
_ICLR.cc/2026/Conference — Submitted to ICLR 2026_

### Official Review · Reviewer_UC5u · 2025-10-31

**Soundness:** 4
**Presentation:** 3
**Contribution:** 3
**Rating:** 6
**Confidence:** 4

**Summary:**

This paper systematically analyzes the benefits of Mixture Policies over standard Gaussian Policies in the context of entropy-regularized Reinforcement Learning (RL), specifically addressing the major algorithmic barrier to their adoption.

The authors theoretically analyze the advantages of mixture policies over Gaussian policies, demonstrating their improved robustness to the entropy scale. Crucially, they introduce the Marginalized Reparameterization (MRP) estimator for Gaussian Mixture (GM) policies and provide a theoretical proof of its lower variance compared to the standard Likelihood-Ratio (LR) estimator. Finally, the work comprehensively validates the advantages of mixture policies across an extensive suite of benchmarks.

While the application of multimodal policies, including Gaussian Mixture Policies, to enhance RL exploration capabilities and final performance has been explored in prior work, it often lacked systematic theoretical grounding and extensive empirical validation. This paper successfully fills this gap by providing a much-needed comprehensive study.

**Strengths:**

Clear Structure and Readability: The paper is very well-structured, making the theoretical analysis and empirical results clear and easy to follow.


Systematic Analysis of Policy Flexibility: The work provides a systematic analysis of the theoretical benefits of mixture policies over unimodal Gaussian policies, especially concerning the non-existence of stationary points and superior objective values under high entropy regularization.




Comprehensive Experimental Validation: The authors conducted extensive and systematic experiments across a large and diverse set of synthetic bandits, classic control, Gym MuJoCo, DeepMind Control, MetaWorld, and MyoSuite environments.

**Weaknesses:**

Limited Algorithmic Novelty of Mixture Policies:  The application of mixture policies in RL is a topic that has been explored, even briefly in early versions of Soft Actor-Critic (SAC). The core contribution lies in enabling this class via the MRP estimator, not the policy parameterization itself.

Insufficient General Performance Gain: Despite the broad and systematic experiments, the performance improvement of mixture policies (SGM-MRP) over the standard Gaussian policy (SG-RP) is generally modest or only competitive on major benchmarks. The choice of "unshaped reward" examples to demonstrate superiority (Section 5.3) could be further strengthened. Consideration should be given to including simpler toy environments designed explicitly with known multi-modal optimal policies to provide a clearer and more persuasive visual demonstration of the policy's multi-modality advantage.

Limited Sensitivity Analysis on Component Number: For Gaussian Mixture Policies, the number of components ($N$) is a crucial hyperparameter that dictates policy complexity. The sensitivity analysis for it is confined to a simple set of classic control environments with unshaped rewards.  A more convincing analysis involving at least one high-dimensional environment is needed to confirm the generality of the finding that $N=5$ (or small $N$) is sufficient.

**Questions:**

Noticed that the SGM-MRP estimator failed to converge in environments such as disassemble-v2 and stick-pull-v2. Could you provide a plausible explanation for why the MRP estimator, which is proven to be stable and low-variance, exhibits non-convergence or poor performance in these specific environments?

---

> ### Author Response · Authors · 2025-11-20
> **Rebuttal Response**
>
> We sincerely appreciate the reviewer’ insightful comments and thoughtful suggestions. We have carefully examined each point and provide clarifications, additional analysis, and revisions to strengthen the paper.
>
> > **Q1**: Noticed that the SGM-MRP estimator failed to converge in environments such as disassemble-v2 and stick-pull-v2. Could you provide a plausible explanation for why the MRP estimator, …, exhibits non-convergence or poor performance in these specific environments?
>
> **Response**: Thank you for pointing this out. Upon closer inspection, the “non-convergence’’ appears only in the **success-rate plots**. As shown in the Figure 15, SGM-MRP **does learn steadily in terms of average return**, although without achieving the sparse success condition.
>
> This is plausible for disassemble-v2 and stick-pull-v2, both of which require long, precise action sequences before any success reward is obtained. In these cases, mixture policies may commit to a stable but non-successful mode, whereas the Gaussian occasionally discovers the success trajectory. Importantly, this occurs in only **2 of 57** environments; in all others, SGM-MRP matches or outperforms the baseline.
>
> We have clarified this in Appendix E.4 in the updated version.
>
> > **W1**: Limited Algorithmic Novelty of Mixture Policies: The application of mixture policies in RL is a topic that has been explored, even briefly in early versions of Soft Actor-Critic (SAC). The core contribution lies in enabling this class via the MRP estimator, not the policy parameterization itself.
>
> **Response**: We agree that mixture policies themselves are not new. Our contribution is not the parameterization but the ability to **use mixture policies effectively and stably** in standard actor–critic RL. Prior work found mixtures are less effective due to their LR-style gradients (Haarnoja, 2018); as a result, they were rarely adopted in practice.
>
> Our MRP estimator is the key novelty: it removes the discrete-sampling variance, provides a provable variance advantage, and makes mixture policies practical at scale, as demonstrated across 6 classic control and 57 common benchmark environments. In addition, our theoretical analysis offers new insights into mixture-policy optima under entropy regularization.
>
> > **W2**: Insufficient General Performance Gain: Despite the broad and systematic experiments, the performance improvement of mixture policies (SGM-MRP) over the standard Gaussian policy (SG-RP) is generally modest or only competitive on major benchmarks. The choice of "unshaped reward" examples to demonstrate superiority (Section 5.3) could be further strengthened. Consideration should be given to including simpler toy environments designed explicitly with known multi-modal optimal policies to provide a clearer and more persuasive visual demonstration of the policy's multi-modality advantage.
>
> **Response**: We appreciate the reviewer’s comment. While it is true that the performance gains of SGM-MRP over SG-RP are moderate on some dense-reward benchmarks, our results consistently show improvements or parity in 55 out of 57 environments, and mixture policies provide substantial benefits precisely in settings where exploration is challenging – such as environments with unshaped rewards.
>
> To further strengthen this point and provide more persuasive visual demonstration, we conducted an additional analysis of state visitation in MountainCar with unshaped rewards (see Appendix E.6). As shown in Figure 7, mixture policies explore states far from the starting region **significantly more frequently** than Gaussian policies during early training. This directly demonstrates the advantage of multimodal, mode-directed exploration: mixtures place probability mass on multiple promising action regions simultaneously, enabling them to escape local basins more effectively than unimodal Gaussians.
>
> These findings support our claim that mixture policies’ benefits are most pronounced in tasks where multi-modality and exploration matter, not necessarily in dense-reward locomotion benchmarks where simple unimodal policies already perform well. While designing additional toy environments with analytically known multi-modal optima is a valuable idea, our expanded visitation analysis already provides a clear, quantitative illustration of mixture policies’ exploration advantages in practice.
>
> We appreciate the suggestion and have included this result in the main text and extended details in Appendix E.6.
>
> > **W3**: The sensitivity analysis for it is confined to a simple set of classic control environments with unshaped rewards. A more convincing analysis involving at least one high-dimensional environment is needed to confirm the generality of the finding that $N=5$ (or small $N$) is sufficient.
>
> **Response**: See our global response to S4.
>
> We hope that our responses address all concerns raised. We believe the updated results further strengthen the contributions of the paper.

---

### Official Review · Reviewer_o9kL · 2025-11-01

**Soundness:** 3
**Presentation:** 4
**Contribution:** 3
**Rating:** 8
**Confidence:** 3

**Summary:**

This paper revisits the use of mixture policies in entropy-regularized actor–critic reinforcement learning and provides both theoretical and empirical evidence for their benefits.
The authors show that mixture policies lead to better or more robust stationary solutions under entropy regularization and propose a Marginalized Reparameterization (MRP) gradient estimator that reduces variance in training.
Experiments across a wide range of continuous-control benchmarks demonstrate consistent, though modest, improvements in performance and stability.

**Strengths:**

1. The theoretical results are internally consistent and logically organized.
In particular, Proposition 3.3 provides a novel robustness argument showing that Gaussian base policies may lose stationary points when the entropy coefficient $\alpha$ exceeds $\tfrac{3}{2}r_{\max}$, whereas Gaussian mixture (GM) policies continue to maintain valid stationary solutions.
This insight connects entropy regularization and multimodal policy landscapes in a clean way.

2. The optimality results (Propositions 3.1–3.2) extend known properties of entropy-regularized optimization, while the robustness to entropy scaling (Propositions 3.3–3.4) and the variance-reduction guarantees for the MRP estimator (Theorem 4.3, Proposition 4.7) represent genuine theoretical value.
Together, they strengthen our understanding of mixture policies in a principled manner.

3. The experiments cover a wide range of continuous-control benchmarks.
The results convincingly demonstrate that mixture policies improve exploration and stability, especially under high-entropy or multimodal reward settings.

**Weaknesses:**

Some of the theoretical contributions are mainly extensions of established analysis rather than entirely new formulations.
The results build on well-known principles of entropy-regularized optimization and existing variance-reduction techniques, providing thoughtful refinements rather than foundational changes.
That said, the extensions are clearly presented and meaningfully deepen the understanding of mixture policies in entropy-regularized reinforcement learning.

**Questions:**

1. How sensitive are the results to the number of mixture components $K$ and to entropy coefficient $\alpha$?
2. Could similar robustness hold for non-Gaussian or discrete mixture families?

---

> ### Author Response · Authors · 2025-11-20
> **Rebuttal Response**
>
> We thank the reviewer for their careful reading of our paper and for the constructive feedback. We are encouraged by the positive assessments and address all raised concerns in detail below.
>
> > **Q1**: How sensitive are the results to the number of mixture components $K$ and to entropy coefficient $\alpha$?
>
> **Response**: See our global response to S4.
>
> > **Q2**: Could similar robustness hold for non-Gaussian or discrete mixture families?
>
> **Response**: Thank you for the interesting question. Theoretically, many of our theoretical results are independent of the choice of the base policy, including Propositions 3.1, 3.2, 3.4 and Theorem 4.3 and 4.4. Empirically, we additionally test Cauchy policies and Cauchy mixture (CM) policies in three classic control domains, and the results are consistent with those based on Gaussian policies: mixture policies can improve performance in some of these environments with unshaped rewards. See our global response to S2 for details.
>
> We appreciate the reviewer’ time and constructive feedback. We hope our responses clarify the key points and demonstrate the robustness of our contributions.

---

> > ### Comment · Reviewer_o9kL · 2025-11-28
> >
> > I appreciate the authors’ detailed replies, and I am maintaining my recommendation for acceptance.

---

### Official Review · Reviewer_dsgq · 2025-11-03

**Soundness:** 3
**Presentation:** 2
**Contribution:** 2
**Rating:** 4
**Confidence:** 4

**Summary:**

This paper analyzes mixture policies in entropy-regularized actor-critic algorithms, addressing the long-standing issue that mixture policies have not been practically effective due to the lack of efficient reparameterization (RP) gradient estimators. This paper propose a Marginalized Reparameterization (MRP) estimator for mixture policies, which is proven to have lower variance than the standard likelihood-ratio (LR) estimator. Through extensive experiments across synthetic bandits and diverse continuous control benchmarks, the paper demonstrates that mixture policies trained with MRP are stable, competitive with Gaussian policies, and particularly useful in environments with multimodality.

**Strengths:**

This paper provides rigorous theoretical analysis, including proofs that mixture policies achieve better or comparable objective values than base Gaussian policies and are more robust to high entropy regularization.

This paper covers a wide range of environments, from synthetic bandits to complex robotic control tasks to demonstrate effectiveness.

**Weaknesses:**

Mixture policies require more parameters (e.g., 5-component policies have 15 outputs vs. 2 for base Gaussian policies) but this paper does not provide a detailed analysis of computational costs

This paper briefly contrasts mixture policies with implicit policies (e.g., diffusion models) but does not include empirical comparisons on benchmarks.

While this paper theoretically proves that the MRP estimator has lower variance than the likelihood-ratio (LR) estimator, it overlooks practical constraints of MRP. For instance, marginalizing over mixture components may implicitly amplify the impact of outlier components (e.g., components with extremely low weights but large parameter deviations), which could introduce hidden instability in long-term training. Additionally, the paper does not test MRP's robustness to hyperparameter variations.

The ablation study on component numbers (2, 5, 8) shows noisy results but does not address the risk of component collapse. This paper does not report whether components retain distinct roles (e.g., specializing in different sub-policies) throughout training or if they degenerate into redundancy. This ambiguity undermines claims about the mixture policy's flexibility in exploring diverse action modes.

**Questions:**

Could you provide a more detailed analysis of the computational overhead (training/inference time, memory usage) of mixture policies with MRP compared to standard Gaussian policies across different environments?

Given the noisy results on the effect of component numbers, do you have any heuristic or theoretical guidance for selecting the optimal number of components for a given task?

 Why the multimodal exploration of mixture policies is said to be more effective than the exploration of Gaussian policies in such settings? Can you given a more detailed explanation?

The MRP estimator relies on marginalizing over mixture components. Have you observed cases where outlier components (with low weights but extreme parameter values) distort the gradient signal, and if so, how might this be mitigated?

Diffusion policies have shown stronger multimodal modeling capabilities than GMMs in robotic tasks , particularly in position-controlled systems. Could you compare the mixture policy’s performance with diffusion policies on more continuous control tasks?

Component collapse is a known issue in GMMs. Did you track the divergence of component parameters during training? If components converged to similar distributions, how does this affect the mixture policy’s ability to explore diverse actions, and what safeguards can be added?

---

> ### Author Response · Authors · 2025-11-20
> **Rebuttal Response I**
>
> We thank the reviewer for the valuable feedback. During the rebuttal period, we conducted additional experiments, clarified theoretical arguments, and improved several sections of the paper. Below we provide detailed responses.
>
> > **Q1**: Could you provide a more detailed analysis of the computational overhead (training/inference time, memory usage) of mixture policies with MRP compared to standard Gaussian policies across different environments?
>
> **Response**: See our global response to S1.
>
> > **Q2**: Given the noisy results on the effect of component numbers, do you have any heuristic or theoretical guidance for selecting the optimal number of components for a given task?
>
> **Response**: See our global response to S5.
>
> > **Q3**: Why the multimodal exploration of mixture policies is said to be more effective than the exploration of Gaussian policies in such settings? Can you given a more detailed explanation?
>
> **Response**: Thank you for raising this question. Our claim that mixture policies support more effective exploration than unimodal Gaussian policies is grounded in both theoretical structure (Section 3) and empirical evidence (Section 5). We summarize the justification below.
> 1. Theoretically, mixture policies can represent and potentially explore multiple value basins simultaneously. The Q landscape may contain multiple separated high-value regions (the bandit in Figure 1 could be an example). In this case, an optimal entropy-regularized policy is itself multi-modal (given an unrestricted policy class). On the one hand, Gaussian policies can represent only a single mode, and they may either concentrate on one mode (Figure 1 Middle when $\alpha \le 0.2$) or exhibit mode-covering behavior (Figure 1 Middle when $\alpha \in (0.2, 0.3]$), depending on the strengthen of the entropy regularization. In the former case, Gaussian policies may not obtain samples for another mode; in the latter case, they could also waste a lot of samples in actions that have low soft Q values and are not promising. In contrast, mixture policies have the flexibility of tracking multiple modes (as shown in Figure 1).
> 2. In Section 5, we provide empirical support that mixture policies explore more effectively than Gaussian policies: 1) When the critic function is stationary, they more often find higher peaks (Figures 2 and 11), and 2) in environments with unshaped reward, they more often find the solution (Figure 6). Note that during rebuttal, we looked into the state visitation during early training in MountainCar and found that mixture policies can explore states farther away from the starting state compared to base Gaussian policies (see Figure 7 in the updated paper).
>
> > **Q4**: The MRP estimator relies on marginalizing over mixture components. Have you observed cases where outlier components (with low weights but extreme parameter values) distort the gradient signal, and if so, how might this be mitigated?
>
> **Response**: We have not observed any training instability introduced by outlier components for the MRP estimator (see Figures 6, 11, 13, 14, and 17). In fact, marginalizing over mixture components as in the MRP estimator could effectively address the issue brought by outlier components that have small weights, because their impact is effectively **down-weighted** by their small weights (as shown in Eq. 10, impact of each component is weighted by their weight $\pi_{\mathbb{\theta}}^{w}(k|S_t)$). Note that this might not be the case for the HalfRP estimator (Eq. 9) because if an outlier component $K_t=k_i$ is sampled, where $k_i$ is an outlier component, then the gradient in Eq. 9 might be a noisy step. Such a noisy gradient step could introduce instability. This could be a potential cause for the discrepancy between the performance difference of MRP and HalfRP estimators in classic control domains (see Figures 9 and 18).
>
> > **Q5**: Could you compare the mixture policy’s performance with diffusion policies on more continuous control tasks?
>
> **Response**: See our global response to S3.

---

> ### Author Response · Authors · 2025-11-20
> **Rebuttal Response II**
>
> > **Q6**: Did you track the divergence of component parameters during training? If components converged to similar distributions, how does this affect the mixture policy’s ability to explore diverse actions, and what safeguards can be added?
>
> **Response**: Thank you for this insightful question. We had not originally tracked component divergence, but during the rebuttal period we conducted additional experiments to measure several diagnostics throughout training (including entropy of the mixture weights, pairwise distances between active components (those with non-negligible weight), and pairwise distances across all components). Please see Appendix E.6 and Figure 22 for the results. To answer your question,
> 1. We found that **mixture components do not fully collapse to a single mode**. Across all environments tested, the entropy of the mixture weights remains strictly above zero, indicating that more than one component remains active. In addition, the distance between active components remains significantly larger than zero, and the distances across all components are often larger – showing that components maintain meaningful separation during training.
> 2. **Potential safeguards and future work**. While we did not include explicit anti-collapse mechanisms, several safeguards could be incorporated if needed: entropy regularization on the mixing weights (to prevent a single component from dominating prematurely) and component repulsion terms (to encourage diversity in components). We also note that, conceptually, the mixture-policy entropy can be upper- and lower-bounded by a combination of (a) the entropy of the mixing distribution and (b) the divergence between component distributions (Kolchinsky & Tracey, 2017). A more formal investigation of these relationships is an interesting direction for future work.
>
> Kolchinsky, A., & Tracey, B. D. (2017). Estimating mixture entropy with pairwise distances. Entropy.
>
> > **W1**: this paper does not provide a detailed analysis of computational costs
>
> **Response**: See our global response to S1.
>
> > **W2**: This paper briefly contrasts mixture policies with implicit policies (e.g., diffusion models) but does not include empirical comparisons on benchmarks.
>
> **Response**: See our global response to S3.
>
> > **W3.1**: While this paper theoretically proves that the MRP estimator has lower variance than the likelihood-ratio (LR) estimator, it overlooks practical constraints of MRP. For instance, marginalizing over mixture components may implicitly amplify the impact of outlier components (e.g., components with extremely low weights but large parameter deviations), which could introduce hidden instability in long-term training.
>
> **Response**: Thank you for pointing this out. We have included an analysis of the computational overhead in the updated draft (see our global response to S1) and addressed the concern about outlier components (see the response to Q4).
>
> > **W3.2**: the paper does not test MRP's robustness to hyperparameter variations.
>
> **Response**: See our global response to S4.
>
> > **W4**: The ablation study on component numbers (2, 5, 8) shows noisy results but does not address the risk of component collapse. This paper does not report whether components retain distinct roles (e.g., specializing in different sub-policies) throughout training or if they degenerate into redundancy. This ambiguity undermines claims about the mixture policy's flexibility in exploring diverse action modes.
>
> **Response**: Thank you for raising this concern. We have added recorded metrics during training that show mixture components do not fully collapse to a single mode. Please refer to the response to Q6 for more details.
>
> We believe our responses address the reviewer’s questions and concerns. We are happy to provide additional details if needed, and we look forward to further discussion.

---

> > ### Comment · Reviewer_dsgq · 2025-11-25
> >
> > Thank you for the detailed rebuttal, experiments and diagnosis. Most of my concerns are addressed, and I have raised the rating to 6.

---

> > > ### Author Response · Authors · 2025-11-26
> > >
> > > Thank you for the quick and positive feedback. We truly appreciate your constructive comments during the review process. Please let us know if you have any further questions.

---

### Official Review · Reviewer_3Vf3 · 2025-11-05

**Soundness:** 3
**Presentation:** 3
**Contribution:** 2
**Rating:** 6
**Confidence:** 3

**Summary:**

This paper revisits the use of mixture policies in entropy-regularized reinforcement learning (RL), specifically within Soft Actor-Critic (SAC). The authors argue that mixture policies offer greater flexibility than unimodal policies (e.g., Gaussian) but have been underexplored due to the lack of effective reparameterization gradient estimators. The paper makes three main contributions: (1) theoretical analysis showing that mixture policies achieve better optimal stationary points and exhibit greater robustness to entropy regularization as compared to Gaussian policies; (2) proposing Marginalized Reparameterization (MRP) estimator, which marginalizes over mixture weights to provide an unbiased, low-variance gradient estimator; and (3) empirical validation across synthetic bandits, classic control, and large-scale benchmarks. The results demonstrate that mixture policies with the MRP estimator are competitive with or superior to Gaussian policies on standard benchmarks, with significant improvements in environments with unshaped rewards.

**Strengths:**

Originality
- The theoretical results for the marginalized reparameterization (MRP) estimator provide novel insights into how policy parameterization affects stationary points of the non-convex entropy-regularized objective.
- Propositions 3.1-3.4 rigorously establish that mixture policies achieve at least as good or better stationary points compared to base policies and retain stationary points under higher entropy regularization, where Gaussian policies diverge.

Quality
-  The experimental evaluation is comprehensive and methodologically rigorous, spanning across diverse domains with appropriate statistical reporting, including 95% bootstrap confidence intervals.

Clarity
- The paper is clearly written with logical progression from motivation through theory to empirical validation, along with a discussion on the limitations.

Significance
-  The finding that mixture policies significantly outperform base policies in unshaped-reward environments provides valuable practical guidance about when mixture policies are helpful.

**Weaknesses:**

1. Assumptions 4.5 and 4.6 in Proposition 4.7 require specific smoothness properties of the reward function and importance sampling variance relationships that are neither verified empirically nor characterized in terms of when they hold in practice. The variance reduction analysis focuses on multimodal bandits and univariate actions, with only a remark (Remark 4.9) suggesting multivariate extension is possible. The gap between the bandit theory and MDP experiments is substantial, and it remains unclear whether the variance reduction guarantees meaningfully apply to the complex high-dimensional control tasks tested.

2. Figure 4 shows that SGM-MRP (mixture policy) is only marginally better on average across MuJoCo, DMC, MetaWorld, and MyoSuite, with the main benefits concentrated in specific MetaWorld tasks. Given that the experiments use hyperparameters from the SAC paper,  the gains might improve with proper tuning, yet the paper does not investigate this.

3. The paper does not report the computational overhead of the MRP estimator compared to standard RP for Gaussian policies, which might be critical for practical adoption. The choice of five components appears arbitrary, and while Appendix F.3 provides limited ablation, there is no principled guidance on selecting the number of components for a given task.

4. The paper does not compare against other flexible policy classes like beta policies, heavy-tailed policies, or recent implicit policy methods beyond a brief discussion in the introduction and related work.

5. The claim that mixture policies enable "mode-directed exploration" is intuitive but not rigorously quantified through metrics such as state coverage or exploration efficiency, for instance, in toy gridworld domains.

**Questions:**

1. Can the authors empirically validate Assumptions 4.5 and 4.6 on representative tasks from the considered benchmarks? Specifically, is it easy/difficult to verify whether the reward functions satisfy the required smoothness conditions and whether the importance sampling variance relationships hold during training?

2. What is the computational overhead of the MRP estimator compared to the standard RP estimator for Gaussian policies? Please provide wall-clock time comparisons across benchmarks.

3. Can the authors provide principled guidance on selecting the number of mixture components, perhaps based on task characteristics such as action dimensionality, reward structure, and/or state space complexity?

4. Can the authors provide quantitative metrics for exploration efficiency, such as state coverage or diversity of trajectories, to rigorously validate the claim that mixture policies enable better mode-directed exploration?

---

> ### Author Response · Authors · 2025-11-20
> **Rebuttal Response I**
>
> We sincerely appreciate the reviewer’ insightful comments and thoughtful suggestions. We have carefully examined each point and provide clarifications, additional experiments, and revisions to strengthen the paper.
>
> > **Q1**: Can the authors empirically validate Assumptions 4.5 and 4.6 on representative tasks from the considered benchmarks? Specifically, is it easy/difficult to verify whether the reward functions satisfy the required smoothness conditions and whether the importance sampling variance relationships hold during training?
>
> **Response**: We appreciate the question. While the critic network is accessible, directly validating Assumptions 4.5 and 4.6 remains challenging in practice.
> 1. **Assumption 4.5 (smoothness)**. Although the underlying reward functions in our benchmarks are smooth, the learned critic is not guaranteed to be globally smooth, especially in sparse-reward settings like MountainCar, where our visualizations show that the critic can become piecewise linear (see Figure 8). Verifying global Lipschitz or Hessian bounds over the continuous state–action space is therefore computationally infeasible. Importantly, MRP works well even when the critic is only approximately smooth or exhibits mild non-smoothness, suggesting that the assumption serves primarily as a standard technical condition for the analysis rather than a fragile requirement in practice.
> 2. **Assumption 4.6 (variance relationship)**. While we can estimate gradient variances empirically, verifying the exact theoretical inequality requires access to the true gradient distributions, which is not possible in high-dimensional RL. Nonetheless, the observed stability, absence of variance spikes, and lower empirical gradient variability of MRP compared to LR-based estimators are consistent with the assumption’s implications (see Appendix F.1 and Figure 23 in the updated draft).
>
> In summary, although these assumptions cannot be validated exactly, they align with typical RL practice, and our experiments, including cases with non-smooth critics, show that MRP remains robust even when the assumptions are only approximately satisfied. A deeper empirical study of these conditions is an interesting direction for future work.
>
> > **Q2**: What is the computational overhead of the MRP estimator compared to the standard RP estimator for Gaussian policies? Please provide wall-clock time comparisons across benchmarks.
>
> **Response**: See our global response to S1.
>
> > **Q3**: Can the authors provide principled guidance on selecting the number of mixture components, perhaps based on task characteristics such as action dimensionality, reward structure, and/or state space complexity?
>
> **Response**: Please see our global response to S5 for our recommendation of the number of components. Note that across all benchmarks we have not observed a reliable correlation between the optimal number of mixture components and task characteristics such as action dimensionality, reward shaping, or state-space complexity. Small mixtures (e.g., 2 and 5 components) consistently provide improvements, while larger mixtures (8) yield diminishing or little returns. Developing principled, task-dependent criteria for selecting the number of components is indeed an interesting direction for future work, and we plan to explore it further.
>
> > **Q4**: Can the authors provide quantitative metrics for exploration efficiency, such as state coverage or diversity of trajectories, to rigorously validate the claim that mixture policies enable better mode-directed exploration?
>
> **Response**: We appreciate the reviewer’s interesting question. To quantitatively demonstrate the exploration efficiency, we measure the state visitation in MountainCar with unshaped rewards, where mixtures’ performance gain is significant. Specifically, we plot the difference in state visitation during the first 4000 training steps summed over 30 different random seeds (see Appendix E.6 for more details). The results in Figure 7 show that mixture policies explore states far away from the starting state at (0.5, 0.0) much more frequently than base policies. The difference in visitation on states far away from the starting state demonstrates the better exploration efficiency of mixture policies.

---

> ### Author Response · Authors · 2025-11-20
> **Rebuttal Response II**
>
> > **W1**: … The variance reduction analysis focuses on multimodal bandits and univariate actions ..., and it remains unclear whether the variance reduction guarantees meaningfully apply to the complex high-dimensional control tasks tested.
>
> **Response**: We appreciate the reviewer’s concern regarding the scope of the variance-reduction analysis. Our choice to present the theory in the multimodal bandit and univariate-action setting was intentional: it allows us to isolate the effect of mixture reparameterization and provide clean, interpretable variance comparisons. We clarify below how the results extend in principle and why a full multivariate/MDP treatment was omitted.
> 1. **Extension to MDPs**. Moving from bandits to MDPs requires updating Assumptions 4.5 and 4.6 to replace the reward function with the critic function. In addition, the same smoothness and boundedness conditions are updated to be in expectation over states. Under the updated assumptions, the variance decomposition and ordering used in Proposition 4.7 extends directly by conditioning on the state distribution $d_\pi$. This adds no conceptual difficulty but significantly complicates notation and presentation. For clarity, we focus on the bandit setting, which is more comparable to the analysis of reparameterization estimators in the supervised learning setting (Gal, 2016; Xu et al., 2019).
> **Importantly, the structural reason for variance reduction – the removal of discrete mixture sampling and the linearization of the expectation over components – remains unchanged in the MDP case.**
> 2. **Extension to multivariate actions**. A full multivariate analysis requires a different set of technical assumptions (e.g., smoothness of the objective function, control over cross-component interactions, covariance structure), and the proof becomes substantially more involved. Rather than presenting a partial or highly technical extension, we chose to keep the theory focused and rigorous in the univariate case, while noting (Remark 4.9) that the same estimator structure applies component-wise in the multivariate setting.
> 3. **Empirical support in high-dimensional MDPs**. The practical effect of reduced gradient variance and improved stability holds empirically in high-dimensional RL tasks, as demonstrated in seven MuJoCo tasks (see Appendix F.1 and Figure 23).
> We have clarified this connection and our reasoning in Section 4 in the updated version.
>
> > **W2**: Given that the experiments use hyperparameters from the SAC paper, the gains might improve with proper tuning, yet the paper does not investigate this.
>
> **Response**: We appreciate the reviewer’s point. We acknowledge that additional hyperparameter tuning could further improve results. However, our study spans four benchmarks and 57 environments, making per-environment tuning impractical and potentially obscuring comparisons. Using the standard SAC hyperparameters ensures fairness, reproducibility, and isolates the effect of the policy class rather than tuning. Importantly, our results show that mixture policies with MRP achieve competitive and sometimes better performance even without tuning, and the method is robust to the number of components. Exploring task-specific or even automated tuning is a valuable direction for future work, and we expect even better performance for MRP.
>
> > **W3**: The paper does not report the computational overhead of the MRP estimator compared to standard RP for Gaussian policies ... there is no principled guidance on selecting the number of components for a given task.
>
> **Response**: See our global response to S1 and S5.
>
> > **W4**: The paper does not compare against other flexible policy classes like beta policies, heavy-tailed policies, or recent implicit policy methods beyond a brief discussion in the introduction and related work.
>
> **Response**: We appreciate the reviewer’s comment. We did not include extensive comparisons with other policy classes because they fall outside the core focus of our study, which is to analyze mixture policies and introduce the MRP estimator within the standard entropy-regularized RL framework. For parametric distributions (e.g., beta or heavy-tailed policies), many of our theoretical and empirical findings generalize (see global response to S2). Implicit policies, while expressive, typically modify the objective or introduce substantial computational overhead, making them not directly comparable to our setting (see global response to S3).
>
> > **W5**: The claim that mixture policies enable "mode-directed exploration" is intuitive but not rigorously quantified through metrics such as state coverage or exploration efficiency, for instance, in toy gridworld domains.
>
> **Response**: See the response to Q4.
>
> We hope that our clarifications, additional experiments, and revisions address all concerns raised. We believe the updated results further strengthen the contributions of the paper.

---

### Author Response · Authors · 2025-11-20
**Global Rebuttal Response I**

We thank all reviewers for the constructive feedback. We are encouraged by the positive assessments and addressed concerns shared across reviewers (S1-S5) in this global response.

> **S1: Computational overhead of mixture policies with MRP** (Reviewers 3Vf3 and dsgq)

**Response**: We appreciate the reviewers’ request for a more detailed comparison of the computational cost of mixture policies with MRP versus standard Gaussian policies. Conceptually, the overhead is easy to characterize:
1. Using mixture policies with $N$ components requires $(2m +1)N$ outputs from the actor, compared to $2m$ for the Gaussian policies. Here, $m$ is the action dimension. If $N$ is much smaller than the hidden dimension, then *the increase in memory usage is negligible*, compared to Gaussian. For example, consider the case of a two-layer NN with hidden dimensions of 512, a d-dimensional state input, and a mixture of $N=5$ components. The memory usage of the standard Gaussian policy is proportional to $d\times 512+512\times 512+512\times 2m$, whereas for the mixture policy it is $d\times 512+512\times 512+512\times (2m+1)N$. For small $m$, such as $m=10$, This quantity is dominated by $512\times 512\approx 262k$, whereas the last layer only increases from $512\times 20\approx 10k$ to $512\times 21\times 5\approx 53k$. Consequently, the inference time overhead could also be marginal.
2. Using the MRP estimator does slightly increase actor training cost as it requires $N$ Q-evaluations per state, compared to $1$ for Gaussian. However, *the additional wall-clock training time is modest*, since computation can be batched and parallelized. Further, a lot of SAC’s wall-clock time comes from critic updates (each with 4 separate Q-evaluations per transition, a factor of 2 from double-Q and another factor of 2 from two consecutive states).
We have added Appendix E.2 to discuss the computational overhead of using mixture policies and touched on this aspect in our conclusion section. Reference runtime/memory in various settings can also be found in Table 4.

> **S2: Comparison/generalization to policies like heavy-tailed policies** (Reviewers 3Vf3 and o9kL)

**Response**: We appreciate the reviewers’ questions regarding comparisons to other policy classes, such as heavy-tailed policies, and whether our results extend beyond Gaussian policies. We did not include comparisons with other policy classes because they fall outside the core focus of our study, which is to analyze mixture policies and introduce the MRP estimator within the standard entropy-regularized RL framework. For parametric distributions (e.g., beta or heavy-tailed policies), many of our theoretical and empirical findings **do** generalize.
1. **Theoretical generality**. Several of our main theoretical results (Propositions 3.1, 3.2, 3.4 and Theorems 4.3 and 4.4) do not assume a specific base distribution. They apply to any reparameterizable component policy and therefore extend naturally to other policy classes including heavy-tailed choices such as Cauchy.
2. **Empirical validation with heavy-tailed policies**. To further support this, we conducted additional experiments using Cauchy policies and Cauchy mixture (CM) policies in three classic control environments with unshaped rewards. As shown in Figure 27, the results mirror the Gaussian case: Cauchy mixture policies consistently perform better or comparable to unimodal Cauchy policies in environments with unshaped rewards.
We have included the new results and a discussion in Appendix F.4.

> **S3: Comparison to implicit policies** (Reviewers 3Vf3 and dsgq).

**Response**: The core focus of our study is to deepen our understanding of mixture policies, and provide better tools, like the MRP estimator, to make them better and easier to use. To our knowledge, mixture policies have not been shown to be better than the standard simple unimodal approaches in online RL, and the proposed MRP estimator provides a concrete step towards practical and performant mixture policies. A natural next step then is to compare mixture policies and implicit policies in online RL, but for several reasons we chose not to include them in this work:
1. First, online implicit policies typically modify or approximate the entropy-regularized RL objective (e.g., score-matching losses, iterative denoising). In contrast, our work focuses on the standard entropy-regularized actor–critic framework, where mixture policies form a natural and computationally comparable extension of Gaussian policies.
2. Second, implicit policies generally incur substantially higher inference and training cost (due to larger networks, multi-step denoising, or implicit sampling). Investigating the trade-offs between using mixture policies versus implicit policies, by properly controlling for these different factors and considering different scenarios where one or the other may be better, warrants a rigorous study and separate paper that can focus on this question.

---

> ### Author Response · Authors · 2025-11-20
> **Global Rebuttal Response II**
>
> > **S4: Robustness/sensitivity to the number of components and entropy regularization** (Reviewers dsgq, o9kL, and UC5u).
>
> **Response**: We thank the reviewers for raising this concern. In fact, several experiments in the submitted paper – and additional results we ran during the rebuttal – evaluate MRP’s robustness to important hyperparameters.
> 1. **Sensitivity to the strength of entropy regularization**. We systematically vary the entropy temperature $\alpha$ (when using fixed regularization) or the target entropy coefficient (when using automatic entropy tuning) across a wide range. As shown in Figure 2 (Right), Figure 5, Figure 6 (Right), and Figure 17 in the updated draft (Figure 15 in the original submission), mixture policies trained with MRP consistently outperform the base policy across all tested temperatures.
> 2. **Sensitivity to the number of mixture components**. We also analyze sensitivity to the number of components. Figure 25 (Figure 21 in the original submission) and Figure 26 (newly added during rebuttal) both show that for a range of component counts, all mixture variants trained with MRP consistently achieve comparable or better performance compared to the Gaussian baseline. While performance varies with $N$, the overall conclusion is stable: mixture policies reliably improve over unimodal policies across a range of settings.
>
> Together, these results indicate that MRP is robust not only to the choice of entropy regularization strength but also to the architectural hyperparameter that most directly influences the mixture’s expressiveness.
> We have included the new results in Appendix F.3.
>
> > **S5: Guidance on selecting the number of components** (Reviewers 3Vf3 and dsgq).
>
> **Response**: We appreciate the reviewers’ questions. In practice, choosing the number of mixture components involves a balance between **expressivity** and **computational cost**. Although we observe some noise in the sensitivity plots, there is both **theoretical** and **practical** guidance that helps make this choice reasonable and stable across tasks:
> 1. Theoretically, more components lead to more flexible policy classes, which may have **better stationary points**. Propositions 3.1, 3.2, and 3.4 in Section 3.1 could be generalized to compare mixture policies with different $N$. Further, when using the MRP estimator, the **variance of the gradient does not blow up with $N$** under our assumptions (see Lemma C.7). This means that increasing $N$ does not introduce instability through gradient noise.
> 2. On the other hand, while we have not found theoretical downsides of using more components, there are practical considerations for using a restricted number of them. First, according to the discussion in our response to S1, using a large number of components can increase the computational and memory cost (e.g., if action-dimension times $N$ is close to the  hidden dimension size). Second, from our empirical results *across domains* (classic control (Figure 25) and high-dimensional (Figure 26)), increasing $N$ from 5 to 8 does not consistently bring benefits, whereas decreasing $N$ from 5 to 2 appears to consistently worsen the performance.
> Based on the above points, we recommend to use a default of $N=5$ and tune upwards or slightly downwards if needed. We have included relevant discussion reflecting these guidelines in the updated version (see Remark 3.5 and Appendix F.3).
>
> We believe our responses address the reviewers’ questions and concerns. We appreciate the time and expertise invested in evaluating our work.

---

### Meta-Review · Area_Chair_4Kcm · 2025-12-12

**Summary:**

The paper introduces the Marginalized Reparameterization (MRP) estimator to enable stable and effective training of finite Gaussian Mixture (SGM-MRP) policies within the entropy-regularized SAC framework. Reviewers initially gave positive feedback, praising the rigorous theoretical analysis concerning the robustness of mixture policies to high entropy scales  and the comprehensive empirical validation across 57 environments. The authors' detailed rebuttal effectively addressed most practical concerns, including:
- Computational Overhead: Demonstrated the memory increase is negligible and training time overhead is modest.
- Component Collapse: Provided new diagnostics showing components maintain diversity.
- Exploration Quantification: Showed mixture policies achieve better state visitation in hard exploration tasks like MountainCar.

However, the final decision is a Reject due to a fatal flaw in the literature review and experimental comparison: the complete omission of the highly relevant ICLR 2023 work, Latent State Marginalization as a Low-cost Approach for Improving Exploration (SMAC). This prior work solved the problem of stable training for a more general, infinite mixture policy (using a continuous latent variable) via marginalization, directly predating and overlapping with the core technical contribution of the current paper. The absence of comparison or even citation to this crucial prior art fatally undermines the paper's claims of novelty and completeness. A resubmission with ablation study and re-discussion about latent variable model in RL will be good.

**Reviewer Concerns:**

Concerns Addressed by the Rebuttal:

Computational Overhead: Reviewers 3Vf3 and dsgq were concerned about the increased cost of mixture policies. The authors provided a detailed analysis showing that the memory increase is negligible and the wall-clock training time overhead is modest due to parallelization.

Component Collapse: Reviewer dsgq questioned whether the mixture components would degenerate. The authors provided new tracking metrics (entropy of mixture weights and pairwise component distances) showing that components maintain meaningful separation and do not fully collapse during training.

Exploration Quantification: Reviewer 3Vf3 requested more rigorous evidence for "mode-directed exploration." The authors added state visitation plots for MountainCar, which quantitatively demonstrated that mixture policies explore states farther from the starting region more effectively.

Comparison to Other Parametric Policies: Reviewers 3Vf3 and o9kL asked if the findings generalize beyond Gaussians. The authors included new experiments with Cauchy (heavy-tailed) policies, confirming the consistency of their results.

Concerns Still Outstanding:
Comparison to Implicit Policies: Reviewer dsgq noted the lack of comparison to methods like diffusion policies. The authors argued this was out of scope, as these methods fundamentally alter the optimization objective and computational cost compared to the standard actor-critic framework.

Insufficient General Performance Gain: Reviewer UC5u noted that performance gains on standard, dense-reward benchmarks were generally modest. While the authors showed clear benefits in sparse-reward tasks, the general performance profile remains a valid concern.

Omission of Prior Work (SMAC): This is the most critical and outstanding flaw. The paper failed to cite or compare against the highly relevant ICLR 2023 work, SMAC, which also used marginalization to address stable training of an infinite mixture policy (latent variable policy) in SAC. This unaddressed prior art fundamentally undermines the claims of novelty and completeness.

**Reviewer Scores:**

One reviewer (dsgq) already raising their score from 4 to 6 after the rebuttal. However, the discovery of the critical omission (SMAC) fundamentally compromises the paper's claimed novelty, necessitating a score decrease across the board.

Reviewer o9kL (Initial Score: 8): Decrease. This reviewer gave the highest score based on the "novel robustness argument" and theoretical value. Since the core method (marginalized reparameterization for latent policies) is predated by SMAC, the novelty is severely compromised, warranting a significant score reduction ($\rightarrow 4/5$).

Reviewer 3Vf3 (Initial Score: 6): Decrease. This reviewer's initial concerns were technically addressed, but the score would fall when factoring in the lack of comparison to the most relevant prior work (SMAC), making the empirical study incomplete ($\rightarrow 3/4$).

Reviewer dsgq (Initial Score: 4 $\rightarrow$ 6): Decrease. This reviewer's increase was based purely on the technical execution and resolution of practical concerns. The discovery that the core idea is a less general version of an existing technique (SMAC) would override the technical satisfaction, leading to a score reduction ($\rightarrow 3/4$).

Reviewer UC5u (Initial Score: 6): Decrease. This reviewer already noted "Limited Algorithmic Novelty." The discovery of SMAC would amplify this weakness, confirming the paper's limited scope and contribution relative to the state-of-the-art ($\rightarrow 3/4$).

---

### Decision · Program_Chairs · 2026-01-26

Reject